# An Integrated Workflow for Parametrization of Fracture Network Geometry in Digital Outcrop Models

Stefano Casiraghi<sup>1</sup>, Gabriele Benedetti<sup>1</sup>, Daniela Bertacchi<sup>2</sup>, Silvia Mittempergher<sup>1</sup>, Federico Agliardi<sup>1</sup>, Bruno Monopoli<sup>3</sup>, Fabio La Valle<sup>4</sup>, Mattia Martinelli<sup>4</sup>, Francesco Bigoni<sup>4</sup>, Cristian Albertini<sup>4</sup>, Andrea Bistacchi<sup>1</sup>

10 Correspondence to: Casiraghi Stefano (s.casiraghi21@campus.unimib.it)

**Abstract.** Mesoscale fractures, with lengths between meters and tens of meters, cannot be effectively characterized in the subsurface, due to limitations of borehole and geophysical datasets. On the other hand, large quantitative structural datasets can be collected on outcrops by combining direct observations and remote sensing (digital outcrop models - DOMs). These data can be used to constrain geometrical models of subsurface fracture networks with the outcrop analogue approach.

In this contribution we present a workflow that leverages DOMs with at least two perpendicular faces and combines multiple types of input data (point cloud, textured surfaces and orthophoto DOMs), to collect a suite of statistical parameters to be used as input in current stochastic 3D DFN (Discrete Fracture Network) models.

Orientation data are collected with a semi-automatic procedure applied to point cloud DOMs of the vertical side of the outcrop to extract 2D polygonal facets. Fracture sets are defined with a clustering procedure and different orientation distributions are fitted and tested with goodness-of-fit tests.

Fracture traces are digitized on textured surface or orthophoto DOMs. Topological parameters are calculated on the digitized fracture network on horizontal and vertical orthomosaics, also considering relationships between fractures and bedding. Trace length and height distributions are estimated with an innovative approach, accounting for the censoring bias with survival/reliability analysis.  $P_{21}$  (ratio between total fracture length and sampling area) is measured from traces digitized on the large horizontal outcrop, also allowing for the Representative Elementary Area (REA) to be assessed. Even if the height/length ratio cannot be measured on an outcrop by any means, we attempt to relate heights and lengths under the assumption that the two datasets are correlated, with the longest fractures being also the tallest. We discuss the applicability of our workflow on a large high-quality fractured limestone outcrop in the Murge Plateau near Altamura (Puglia, Italy).

<sup>&</sup>lt;sup>1</sup>Dipartimento di Scienze dell'Ambiente e della Terra, Università degli Studi di Milano-Bicocca, Milan, 20126, Italy

<sup>&</sup>lt;sup>2</sup>Dipartimento di Matematica e Applicazioni, Università degli Studi di Milano-Bicocca, Milan, 20125, Italy

<sup>&</sup>lt;sup>3</sup>LTS s.r.l., Treviso, 31020, Italy

<sup>&</sup>lt;sup>4</sup>Eni S.p.A, Global Natural Resources, San Donato Milanese, Italy

#### 1 Introduction

Fractures exert a fundamental control on the mechanical and hydraulic properties of rock masses, and their relevance extends to multiple applications, including reservoirs of every kind of geofluid (March et al., 2017; Wallace et al., 2021; Wang et al., 2022; Forstner et al., 2025), nuclear waste repositories (Follin et al., 2014; Hadgu et al., 2017), geology engineering (Eberhardt et al., 2004; Agliardi et al., 2017; Franzosi et al., 2023a, b) and contaminant transport (Medici et al., 2024; Cherubini, 2008). In all these applications, fracture patterns hold great importance as they influence the direction, magnitude, and heterogeneity of fluid flow, the storage volume of reservoirs (Davy et al., 2013; Wang et al., 2022), and rock mass strength.

Fracture networks are complex geological objects composed of all the fractures in a rock mass. Here, the term "fracture" will be used as a general term including both opening-mode or shear fractures (joints, faults, etc.), filled or not (veins, joints, etc.). Broadening the meaning of "fracture" by including other kind of discontinuities, such as deformation/compaction bands, foliations, bedding, pressure solution seams and stylolites, etc., may be useful in some research field or application, such as engineering rock mechanics, geomorphology or hydrogeology (Schultz, 2019; Eppes et al., 2024). Fractures can be classified in sets, i.e. populations of cogenetic discontinuities related to the same deformation phases, kinematics (e.g. joint, normal fault), filling (e.g. quartz vein) and orientation, within statistical variability (Hancock, 1985; Laubach et al., 2019).

| Parameter                                  | Fracture network | Fracture set | <b>DOM - Facets</b> | DOM - Traces |
|--------------------------------------------|------------------|--------------|---------------------|--------------|
| Number of sets                             | *                |              | *                   | *            |
| Orientation                                |                  | *            | *                   |              |
| Topology                                   | *                | *            |                     | *            |
| Size (length/height)                       |                  | *            |                     | *            |
| H/L ratio                                  |                  | *            |                     | *            |
| Density/Intensity (1)                      | *                | *            |                     | *            |
| Aperture                                   |                  | *            |                     |              |
| Spatial organization                       | *                | *            |                     | *            |
| Representative Elementary Volume, Area (2) | *                | *            |                     | *            |
| Roughness                                  |                  | *            | * (3)               | * (4)        |
| Kinematics                                 |                  | *            |                     |              |
| Deformation Mechanism                      |                  | *            |                     |              |
| Filling                                    |                  | *            |                     |              |

Table 1 Summary of the fracture properties needed to quantitatively characterize a fracture network. (1) The  $P_{xx}$  system introduced by Dershowitz and Herda (1992) is generally used for density and intensity. (2) The representative elementary volume (REV) can be different for each property and the overall REV of the fracture network can be seen as a combination of REVs for individual properties (e.g. Martinelli et al., 2020). (3) (Candela et al., 2012) (4) (Bistacchi et al., 2011)

The quantitative characterization of fracture networks requires the determination of several geometrical and topological attributes of fractures and their statistical distributions (Table 1). Some of these attributes apply to the individual fracture set (e.g., orientation, length/height distribution) others to the whole fracture network (e.g., topology).

Several factors negatively impact our ability to quantify these parameters, both in the subsurface and in outcrops (e.g. Healy et al., 2017; Laubach et al., 2019; Martinelli et al., 2020):

1) Fractures in the subsurface (e.g. in reservoirs) can only be partially characterized at the mesoscale (meters to tens of meters) using direct techniques. Boreholes provide local information (limited to1D traces in a 3D volume) about the orientation distribution, aperture, fracture abundance (P<sub>10</sub>, Dershowitz and Herda, 1992) and, if the borehole is properly oriented with respect to the average orientation of a fracture set, 1D spatial arrangement. In contrast, length and height distributions, connectivity and the REA cannot be measured.

- 2) Geophysical methods can provide continuous 3D information, but with important limitations since (i) fractures are not always associated to a contrast in physical properties that can be imaged with geophysical techniques, and (ii) in any case the spatial resolution of these datasets is limited. For instance, in good quality industrial 3D seismics, fractures smaller than about 200-300m cannot be detected, and, in order to be directly observed, these fractures should be characterized by a displacement that results in a contrast of seismic impedance across the discontinuity. Summing up, only macro-scale faults can be reasonably imaged in seismics, and this induces a biased estimate of volumetric fracture metrics (e.g. Laubach et al., 2019).
- At the outcrop scale four major biases must be taken into account (Baecher and Lanney, 1978): orientation bias, truncation bias, censoring bias and size bias. The orientation bias stems from the nature of the intersections between the fracture plane and the outcrop surface and to the choice of the sampling dimensionality (e.g., lines, areas or volumes). It influences the representativity of field measurements, and results in downsampling of certain fracture sets with respect to others. The truncation bias imposes a lower boundary to the measured fracture trace length, and it is defined by the smallest feature 70 that is possible to detect. The censoring bias is due to the finite nature of outcrops since the full length of the longest fractures is limited by the outcrop size, and in any case the length of fractures ending outside of the outcrop is not known exactly. The size bias states that larger fractures (i.e. fracture surfaces with a larger area) have a greater probability to intersect the outcrop surface and to be sampled. Another bias, related to layered media, is the under-sampling of fractures shorter than the bed thickness (Ortega and Marrett, 2000). This bias changes the shape of the length distribution, given 75 that only the fracture high enough to abut or crosscut the bedding interface can be systematically sampled. To these major biases it is important to add that the morphological and weathering conditions of the outcrop strongly influence the calculation of parameters like topology, density and intensity. In addition to objective biases related to outcrop geometry or sampling methods, subjective biases introduced by the interpreter should also be considered (Andrews et al., 2019). In the specific context of automatic feature extraction, it is also important to account for biases inherent to the algorithms 80 themselves, including the potential for extracting artifact features.

4) A complete 3D description of the fracture state is only possible in the lab at the centimetric to decametric scale, using non-destructive imaging techniques such as X-Ray Computer Tomography (Agliardi et al., 2014, 2017), which allow measuring volumetric parameters such as  $P_{33}$  (fracture porosity, i.e. fracture volume per unit volume),  $P_{32}$  (volumetric fracture intensity, i.e. fracture area per unit volume) and  $P_{30}$  (volumetric fracture density, i.e. fracture number per unit volume; Dershowitz and Herda, 1992).





The impossibility to directly map or image fractures in the subsurface lead to using continuum representations based on some form of upscaling or homogenization, such as the dual porosity model (Warren and Root, 1963). Alternatively, the Discrete Fracture Network (DFN) approach allows generating stochastic simulations where fractures are simplified as planar polygons in 3D or segments in 2D. In the standard and most widespread approaches to stochastic 3D DFNs, the geometrical properties of each fracture are drawn from parametric length and orientation distributions, and fracture height is generally controlled by a fixed height/length ratio. The simulator generates fractures until a target fracture intensity  $P_{32}$  (Dershowitz and Herda, 1992) is reached in the simulation volume (e.g. Move – https://www.petex.com/pe-geology/move-suite/, Petrel – https://www.slb.com/products-and-services/delivering-digital-at-scale/software/petrel-subsurface-software/petrel, FracMan – https://www.wsp.com/en-gl/services/fracman, DFNworks – https://dfnworks.lanl.gov/). Fractures are randomly distributed in the simulation volume according to a Poisson point process, therefore connectivity or any other form of spatial organization cannot be reproduced in these models. More sophisticated approaches have been developed in the last years to try and solve this fundamental limitation (Bonneau et al., 2013; Davy et al., 2013; Bonneau et al., 2016; Shakiba et al., 2024), but a satisfactory solution has yet to be found, especially in 3D.

Due to the beforementioned limitations in subsurface datasets, input properties for generating stochastic DFNs are often obtained from representative analogues exposed in outcrops that can be characterized, compensating for the information gap at the reservoir scale. The outcrop analogue approach assumes that the detailed information gathered at selected, high-quality rock outcrops can be considered representative of the fracture network properties of deep rock masses that underwent a geological and tectonic evolution that is at least partly comparable. The applicability of an outcrop as an analogue should be evaluated carefully, and some assumptions should be eventually made (Forstner and Laubach, 2022).

This approach relies on the availability of extensive datasets to characterize statistical distributions of the fracture network. In this regard, field survey, intended as physically inspecting and collecting data from outcrops, is a fundamental step in the process of fracture network characterization, because features such as kinematics, roughness, relative chronology and mineralization/filling can only be gathered during fieldwork. At the same time, even if it is possible, manually collecting massive amounts of data is time consuming on horizontal outcrops, and very difficult in vertical outcrops, where the accessibility is limited (data can only be collected in the portion of the outcrop reachable by the geologist) and depending on the conditions, safety is not guaranteed (e.g. rocks falling from the top of the cliff). To solve this problem, Digital Outcrop Models (DOMs) - high-resolution 3D photorealistic representations of natural outcrops (Bellian et al., 2005; Bistacchi et al., 2022b) have been successfully employed to collect large quantitative structural datasets, overcoming the limitations of classical

survey techniques (Sturzenegger and Stead, 2009; Gigli and Casagli, 2011; Sturzenegger et al., 2011; Riquelme et al., 2014, 2015; Bistacchi et al., 2020; Martinelli et al., 2020; Bistacchi et al., 2022a; Storti et al., 2022).

Depending on the outcrop morphological expression, data can be collected from DOMs using either facets - 2D planes interpolated on the DOM, or traces - polylines that are usually digitized in a GIS environment, but sometimes also on a 3D DOM (Bistacchi et al., 2022b). These two types of data carry different but complementary information; however, the methodologies developed in previous contributions by different authors are often based on only one of these kind of data, limiting the number of parameters that can be obtained (Ortega et al., 2006; Boro et al., 2014; Martinelli et al., 2020; Smeraglia et al., 2021).



The scope of this paper is to present a workflow based on statistically rigorous methodologies to characterize a fracture network from the geometrical point of view. The result of such workflow provides a suite of parametrical distributions to be used as input in current stochastic 3D DFN models. The parameters considered here are: The orientation distribution, the length/height distributions, the topological parameters, the fracture areal intensity ( $P_{21}$ ) and the H/L ratio. We aim at integrating 2D and 3D data sources (point clouds, orthomosaics, DEMs), vertical and horizontal outcrops and facets and traces data to achieve a 3D geometrical parametrization of the fracture network (Sect. 9 and following). The methodologies proposed to estimate each parameter can be applied independently, subject to the type and quality of the outcrop.

The first part of the paper is dedicated to best practices about data acquisition (both ground-based and UAV-based), preprocessing, reconstruction and quality assessment of a photogrammetric model (Sect. 3). Then two separate processing pipelines are presented, depending on the DOM type: (i) semi-automated fracture orientation analysis carried out on point cloud DOMs (Sect. 4.4); and (ii) fracture trace analysis carried out on orthomosaics, allowing to measure topological relationships, length and/or height distributions,  $P_{21}$ , and to estimate (subject to assumptions) the H/L ratio distribution (Sect. 5 to 8).

We tested our workflow at an abandoned quarry of the Altamura Limestone Fm. (Puglia, Italy), where both a horizontal pavement and vertical walls provide the opportunity to fully characterize the fracture network in 3D.

Figure 1 Flow chart of the presented workflow. Numbered boxes define the sections of this contribution in which the respective steps of the workflow are addressed.

#### 2 Selecting an outcrop: the Altamura Limestone at Pontrelli quarry



Outcrops for quantitative fracture survey needs to be carefully selected, in order to satisfy some requirements: a) representativity of the structural and lithological properties of the larger rock volume of interest (e.g. lithological characteristics, structural setting, etc.); b) size large enough to be representative for the structures to be investigated; c) continuous unimpeded exposure; d) optimal orientation with respect to the main fracture sets, to minimize orientation biases (Terzaghi, 1965; Zhang et al., 2002). In this context, it is important to select outcrops that present at least two exposed perpendicular sides (e.g., a vertical cliff and an exposed pavements), natural or artificial (e.g., quarry site), for a full 3D characterization of the fracture network metrics.

Here we consider an abandoned quarry (cava Pontrelli) carved into the fractured limestones of the Apulian platform, in the Murge Plateau near Altamura (Puglia, Italy), in the forebulge of the Southern Apennines fold and thrust belt(Panza et al., 2019). The quarry provides 18.000 m<sup>2</sup> of horizontal pavement and vertical walls with a cumulative width of up to 500 m and up to 6 m in height, where fractures are beautifully exposed thanks to the careful maintenance of the site (Figure 2A, B, C) that is carried out because of thousands of dinosaur footprints that were discovered by Nicosia et al. (1999). The outcrop, well

known and described in previous papers(Panza et al., 2015, 2016, 2019) has been recognized as a suitable analogue for reservoirs in related areas (Zambrano et al., 2016).

The quarry is carved into the shallow marine intertidal limestones of the Calcare di Altamura Formation (Coniacian to Early Campanian, Panza et al., 2016). Limestones are well-stratified light-brown mudstones, with wackestone-packstone layers at the bottom of the beds and sometimes algal laminites in the upper parts. Strata are 20 – 60 cm thick and are organized in thickening upward cycles, some meters thick, bounded at the top by major surfaces of subaerial exposure. Bed interfaces often consist of stylolites having teeth both perpendicular to the folded bedding and tangential (slickolites). The outcrop shows three main fracture sets and a set of "major" structures, that are actually major at the outcrop scale, but negligeable at the regional scale (Figure 2B). Set 1 is the most persistent, it is NW-SE striking and, based on abutting relationships, predates all the other sets (Table 2). It presents both joints and meso-faults with a vertical displacement up to a few cm. Set 2 is also striking NW-SE on average, but with a wider scatter, and it also includes both joints and strike-slip meso-faults. However, structures belonging to Set 2 always abut on those belonging to Set 1, showing that Set 2 is younger (Table 2). Set 3 is NE-SW striking and includes fractures that abut on those belonging to both Set 1 and 2. The trace of these fractures, that are limited by older structures, are relatively short, but are responsible of most of the connectivity of the network (Table 2). Aside from the geometrical characteristics, veins are absent in all of the three fracture sets, as well as fibres on small faults (Set 1 and Set 2). Even though this outcrop is top-quality in terms of fracture parameterization, due to the areal extension, cleanliness of the pavement, and the association between horizontal and vertical outcrops, some limitations must be still evaluated.

The pavement (Figure 1A, B) presents some no data zones – where no data can be collected at all, due to debris patches or the presence of strong concentrations of non-natural features produced by quarrying activities. Other zones distributed across the pavement are partially affected by non-natural, quarrying-related fractures, but with a careful analysis it is still possible to detect Set 1, while Set 2 and especially Set 3, being composed of smaller fractures, are more difficult to interpret and separate from the ones related to quarrying. Regarding the quarry walls, here we present data on the NW wall (Figure 2C), that is less disturbed by quarrying activities and favourably oriented with respect to Set 1 and 2, while Set 3 is sub-parallel to the wall. The wall is around 6m tall, and according to the stratigraphic analysis proposed by Panza et al., (2016), includes a bed package developed above the quarry pavement, which is one major subaerial exposure surface, while other prominent subaerial exposure surfaces are not detected inside the wall.

Figure 2 (A) Aerial view of the Pontrelli quarry, Altamura, Italy. The quarry pavement is highlighted in pink, while the analysed quarry wall is highlighted in green. (B) Orthomosaic of the quarry pavement, with digitized fractures and interpretation boundary. (C) Orthomosaic of the interpreted vertical wall, with digitized fractures and interpretation boundary. (D) Field data collected along the quarry walls. The stars represent the medoid of each cluster. Each medoid is colorized with the color of the set according to the legend. (E) Rose diagram of orientation data collected from fracture traces on the orthophoto of the pavement (B). (F) Rose diagram of orientation data collected from fracture facets on the digital outcrop shown in (C).

| Fracture set | Structures & kinematics | Average strike | Relative chronology                       |
|--------------|-------------------------|----------------|-------------------------------------------|
| Set 1        | Joints and meso-faults  | NW-SE          | Abutted by Set 2 & 3                      |
| Set 2        | Joints and meso-faults  | NW-SE          | Joints abut on Set 1 & abutted by Set 3.  |
| Set 2        | Joints and meso-faults  | NW-SE          | Faults crosscut Set 1 & abutted by Set 3. |
| Set 3        | Joints                  | SW-NE          | Abut on Set 1 and 2                       |

Table 2 Summary of fracture sets characteristics at the Pontrelli quarry.

#### 3 Digital Outcrop Model reconstruction and pre-processing

Once the best exposures have been selected, we must also take care of collecting the best input data in order to create a high-quality DOM, that will greatly facilitate the workflow downstream. This topic was covered extensively by Bistacchi et al. (2022b) and here we just summarize the main requirements in the next paragraphs, always considering the Cava Pontrelli case study.

#### 3.1 Photogrammetric acquisition






Horizontal pavement DOMs have been acquired with a DJI Mavic 3E drone flown with an autonomous flight application (DJI Fly app). The photos were shot perpendicular to the outcrop, with a 70% overlap, both between photos pertaining to a single strip and between adjacent strips. As discussed in Bistacchi et al. (2022b), flights at different altitudes were collected to avoid large-scale distortion in the photogrammetric model, and the minimum altitude of 8m allowed collecting images with a ground resolution of 4 mm/pixel. Georeferencing of these DOMs is based on GPS data collected by the drone and recorded in EXIF data of each photo.

Vertical cliff DOMs have been collected with a Nikon Z7 full-frame mirrorless camera mounted on a tripod with a graduated head, adopting a multiple fan scheme (Bistacchi et al., 2022b), in which every shooting station is evenly spaced by 10° of interstation vision angle, measured targeting a certain point on the outcrop and moving parallel to the outcrop by a distance corresponding to 10°. From each camera locations several photos were shot with a fan pattern, trying to cover the whole outcrop and using different focal lengths, and some shooting stations were collected from a larger distance. This shooting

- scheme allows (i) avoiding large-scale distortion in the photogrammetric model and (ii) results in an optimal reconstruction of rough outcrop faces, characterized by facets that form a high angle with respect to the main viewpoint. Noteworthy, this kind of survey could be also carried out with a drone, flying and shooting manually, replicating the ground-based multiple fan scheme, but only high-end cinema-grade drones have cameras that can come close to the quality of a high-end full frame DLSR or mirrorless camera, with significantly higher costs, hence where possible we prefer to use the ground-based technique. The resulting photogrammetric model has a resolution of approximately 2 mm/pixel.
- Georeferencing of the terrestrial surveys was simply performed by marking on the outcrop the location of the mirrorless camera shooting stations before carrying out the drone survey. These points were then retrieved from the drone dataset with an accuracy of better than 4 mm (allowed by the high resolution) and used to co-locate the terrestrial dataset in an accurate and perfectly consistent way.

#### 3.2 Point cloud DOM and Textured surface DOM

Regardless of the technique used to acquire the data, DOMs can be rendered, depending on the outcrop morphology and the scope of the work, as point cloud DOMs (PC-DOMs) or textured surface DOMs (TS-DOMs) (Bistacchi et al., 2022b). PC-DOMs, as the name suggests, are dense sets of points, where each point is characterized by XYZ coordinates and an RGB value, and they are the main output of SFM/MVS photogrammetric reconstructions or laser scanning acquisitions. PC-DOMs are particularly suitable to carry on structural interpretations, using specific tools (Thiele et al., 2018), on outcrops where fractures appear as facets of different size and orientation (as in Figure 3). TS-DOMs are derived from PC-DOMs by generating a polygonal mesh from the point cloud and texturing images onto its surface (Tavani et al., 2014; Bistacchi et al., 2015). In this case the geological and structural interpretation can be carried out in 3D or, as we do in this contribution, with a standard 2D Geographical Information System (GIS) environment (e.g., OGIS).

#### 3.3 Quality of the photogrammetric model

- Defining an absolute quality criterion for a point cloud obtained from a photogrammetric survey is not easy, as different kinds of applications have different requirements. In our application scenario, absolute precision is of lesser priority with respect to the relative accuracy within a local reference frame. This can be evaluated in early stages of the photogrammetric processing considering the image reprojection error, measured in pixels, as it directly impacts the relative accuracy of the photogrammetric model as a fraction of its ground resolution (expressed in mm/pixel).
- A fundamental requirement in a DOM aimed at structural analysis is that it must be completely free from artifacts (doubled surfaces, distortion, doming), and that noise (isolated points outside the outcrop surface) should be as low as possible. A typical artifact resulting from a low-quality acquisition scheme, that does not include fans or photos collected at variable altitude as discussed above, is the presence of doubled "surfaces", consisting in layers of duplicated points that do not define univocally the outcrop surface. Bistacchi et al. (2022b), suggested that the best solution is to use a high-quality acquisition scheme, since a posteriori solutions do not work or are hugely time-consuming.

We believe that the most important parameter to evaluate the quality of a photogrammetric model for applications in structural geology is the point cloud surface density (SD). By defining a kernel - a sphere of radius R moving in such a way as to being cantered on each point - the point surface density SD can be calculated as the ratio between the number N of points falling in the kernel and the area  $\pi R^2$  of the largest circle inscribed in the sphere with radius R:

$$245 SD = \frac{N}{\pi R^2} (1)$$

As an example, in Figure 3, two PC-DOMs of the same vertical outcrop are compared, collected in two different ways to obtain a different *SD*. The PC-DOM in Figure 3C is reconstructed from more than 400 photos collected as discussed above (fans scheme, with high end camera, Nikon Z7). On the other hand, the PC-DOM in Figure 3D is collected with a smaller dataset (150 photos) collected with a lower quality camera (DJI Mini 3 Pro). The mean *SD* of the PC-DOM shown in Figure 3D is two orders of magnitude higher than the PC-DOM of Figure 3C (298826 vs. 5249 points/m<sup>-2</sup>), resulting in a much sharper point cloud, from which it is possible to extract more easily, much more structural information.

In conclusion a good PC-DOM must be free of artifacts, have low noise, and have a high SD on all surfaces of interest, including facets that form a high angle with the outcrop mean plane, which can be properly imaged only if a multiple fans scheme is used.

Figure 3 Comparison between two different PC-DOMs of the same vertical outcrop. (A) and (B) frequency distribution of the SD measured with a kernel of 0.049m. The parameters are obtained by fitting a Gaussian model. (C) PC-DOM reconstructed from photos collected with a high-resolution full-frame mirrorless camera. (D) PC-DOM reconstructed from photos collected with a commercial drone and a simplified acquisition scheme. Point size has been magnified five times, for visualization reasons.

#### 4 PC-DOM: semi-automated analysis of fracture orientation

The goals of orientation analysis are to measure the attitude of each fracture facet that can be mapped on the DOM and to classify it within a fracture set (i.e. a statistically defined fracture cluster), amongst those identified in the preliminary field survey, or emerging from the clustering analysis (Section 4.3), and finally to obtain statistically validated orientation statistics for each fracture set.

Fractures in PC-DOMs are mainly represented by point patches that are the morphological expression, on the outcrop surface, of fracture surfaces, exposed due to natural (e.g. erosion or rockfall) or anthropic (e.g. excavation) events. Here we propose a semi-automatic workflow to map these patches, based on a first step of manual mapping of a subset of fracture planes on the PC-DOM. This allows selecting different sets of structures, characterizing their orientation statistics, and assigning them to sets defined in the field. Based on dip and dip direction ranges, the PC-DOM is manually segmented into as many parts as the number of recognized fracture sets. The automatic step consists in the automatic interpolation of 2D planar features from the segmented point cloud, eventually allowing to greatly increase the number of facets included in the analysis, with important benefits for the statistical analysis (Figure 5).

Our workflow can be carried out in CloudCompare (<a href="https://www.danielgm.net/cc/">https://www.danielgm.net/cc/</a>), the most used open-source software for point cloud processing (Dewez et al., 2016; Thiele et al., 2018) or in PZero, a new 3D geomodelling application where we are also developing new tools for DOM analysis (<a href="github.com/gecos-lab/PZero">github.com/gecos-lab/PZero</a>).

#### 4.1 Orientation parameters for fracture sets

Orientation data are usually recorded in geology using polar coordinates, either as dip and dip direction (dip azimuth) or dip and strike for planar features, or as plunge and trend for axes. In general, any orientation can be represented as a unit vector within a three-dimensional spatial framework (e.g. Mardia and Jupp, 2000), and polar coordinates can be converted into director cosines in a dextral cartesian reference with:

$$L = \sin(dir)\cos(dip) \tag{2}$$

$$285 \quad M = \cos(dir)\cos(dip) \tag{3}$$

$$N = -\sin(dip) \tag{4}$$

where L is the component in direction East, M is directed towards the North, and N is directed upwards (Borradaile, 2003).

The unit normal vector ( $\vec{v} = L, M, N$ ) is calculated on every point of the point cloud by fitting a plane to the points that fall within a sphere of specified radius, cantered at the point itself. The larger the radius the smoother the normal vectors will result, with the drawback of longer computational times.

With a few exceptions (e.g. bedding with polarity, flow directions), the orientation of geological structures and particularly of deformational features like fractures shows a symmetry where the sense does not bear any geological meaning. In mathematical terms this means that two vectors  $\vec{v}$  and  $-\vec{v}$  are equivalent in this kind of analysis. This symmetry implies that different conventions can be adopted for the sense of normal vectors, i.e. the geological convention where normal vectors always point downwards or the photogrammetric convention where they point out of the outcrop.

#### 4.2 Manual orientation mapping

The first step of the workflow is carried out by manually mapping facets, and particularly their attitude, in the PC-DOM with the Compass plugin in CloudCompare (Thiele et al., 2018) or with the facet mapping tool in PZero. These tools behave in the same way and mimic the process of manually collecting attitude data in the field with a geologist compass-clinometer. The fundamental goal of this step of the workflow is to sample each set to define its dip and dip direction range, that will be used in the manual segmentation step. Therefore, we suggest carrying out the mapping with an initial random sampling and then avoiding oversampling the most represented sets (that are generally those favoured by the outcrop orientation bias).

Dip and dip direction values are obtained by fitting a local plane on points selected with a spherical kernel. The kernel radius is defined on-the-fly during mapping by the user, based on the dimension of the fracture plane. A too small kernel will result in measurements affected by the roughness of the fracture plane, while a too big kernel will include points pertaining to other surfaces, biasing the orientation value. Orientation data collected in this way are plotted in stereoplots and compared with data collected in the field (Figure 2D), in order to assess whether all field-defined sets are also represented in the digital dataset.

#### 4.3 K-medoid clustering

- The precise identification of fracture clusters is fundamental in the following automated segmentation step, where each fracture set corresponds to a cluster of orientation data that can be uniquely defined with a measure of location a series of measures to locate the fracture cluster in the parameter space (e.g., mean L, M, N), and a measure of concentration or dispersion (Borradaile, 2003). Clustering analysis provides a quantitative answer to both the number of clusters the dataset is composed of, and the parameters of each cluster, given an assumption on the type of distribution.
- We apply the K-medoid method to a dataset organized as a table, with *n* rows corresponding to individual orientation data and three columns corresponding to the three director cosines. K-medoids is a partitional method (Kaufman and Rousseeuw, 1987) aimed at classifying the data into  $k_m$  groups, where  $k_m$  is the number of fracture set defined in the field, eventually adjusted by the visual inspection of the plotted data. Each group must contain at least one object, and each object must belong to only one group. Partitional methods try to find a suitable partition by separating objects close to each other from objects far away from each other, and how the proximity between objects is calculated determines the specificity of the method. Considering K-medoids in a 3D parameter space (L, M, N), the location parameter is defined by a medoid, i.e. the point belonging to the cluster that minimizes the average distance in the 3-dimensional space between all the other data belonging to the cluster and the medoid itself (Kaufman and Rousseeuw, 1987, 2005). K-medoids therefore measure distance in an isotropic way in the 3-dimensional space of the dataset. When compared to the more popular K-means approach, K-medoids are more robust and less affected by outliers (Kaufman and Rousseeuw, 1987).

One of the problems that arises with K-medoids and other similar partitional methods lies in the definition of the approach itself, as the number  $k_m$  of cluster is imposed by the user, and this can lead to an underestimation or overestimation of the real number of clusters (Kaufman and Rousseeuw, 2005). However, in our application this is not a problem, since the number of

fracture sets is iteratively defined starting from an initial guess defined in the field and the clustering algorithm is applied as a validation of that hypothesis, eventually adjusting the number of sets to account for clusters that only surfaced during the statistical analysis.

A second drawback is that in the standard implementation the initial guess for the medoids is chosen randomly, and when different fracture sets show a partial superposition, the clustering algorithm could yield inconsistent and unreliable results. To address this issue, the initial guess can be defined by the interpreter by manually positioning the initial guess for the medoid. Finally, to avoid the sense ambiguity of orientation unit vectors discussed above, particularly critical for clusters of subhorizontal vectors that can be mirrored across the stereoplot equator, we have developed a solution based on mirroring all input data. For each input vector  $\overrightarrow{v_i}$  we create another vector  $-\overrightarrow{v_i}$  with parallel direction and opposite sense (i.e. pointing in the upper hemisphere (Figure 4). Then we perform the K-medoids clustering on this duplicated dataset, extracting  $2k_m$  clusters both in the upper and lower hemisphere. Given the symmetry imposed by duplicating the data with mirroring, also the resulting  $2k_m$  medoids will be symmetrical, with each medoid in the lower hemisphere having a perfectly symmetrical pair in the upper hemisphere and vice versa, then to conclude the analysis we extract just the  $k_m$  clusters with the medoid pointing downwards, in the lower hemisphere (Figure 4).

Figure 4 (A) 3D stereoplot of two hypothetical fracture set collected on the field. Light blue and magenta sets are the same set but recognized as two different sets, due to the geological sign convention. (B) Every set is doubled and mirrored with respect to the center of the sphere (B). The clustering algorithm is applied on double the number of the set and only the centrotypes that follow the geological sign convention are kept (B).

#### 4.4 Manual segmentation of PC-DOM and semi-automatic planar feature extraction

Based on orientation statistics and K-medoids clustering, it is now possible to segment the whole point cloud into subsets with normal unit vectors falling within the statistical variability of different fracture sets.

The subsets are composed of isolated clusters of points, with the same orientation, each representing a portion of a separated fracture plane. This greatly improves the automatic extraction of 2D polygonal features during the following steps because every patch of points is isolated from the others, nullifying the risk of merging adjacent clusters into one bigger 2D polygon and avoiding the possibility of generating planes with an averaged orientation between two different point clusters, pertaining to two different fracture sets with different orientations. Another advantage of the manual segmentation is that it is possible to specifically calibrate the algorithm for each fracture set.

The segmented point cloud subsets are then fed to the CloudCompare FACETS plugin (Dewez et al., 2016), and specifically to its fast-marching algorithm (Sethian, 1999), to interpolate a planar polygonal feature per every point patch that matches the calibrated algorithm parameters. In this context, manual mapping and orientation analysis act as a calibration step in preparation for the final automatic extraction of planar features, that is greatly simplified and results in clean results thanks to the segmentation step, that avoids generating spurious facets.

Figure 5 Scheme of the semi-automatic workflow for segmenting the point cloud presented in Section 4. Point cloud colored based on dip direction with a HSV 380 colour scale. (A) Manual data collection on PC-DOM. (B) Manually collected orientation data during the preliminary orientation analysis. Number of data: Set 1 = 351, Set 1 = 256, S = 87, Set 3a = 74, Set 3b = 42 (C) Manual segmentation of the PC-DOM. (D) automatic feature detection with FACETS plugin. (E) Final result of the semi-automatic extraction workflow. Each fracture set is individually shown with contour lines.

Fast marching algorithms are a class of methods developed to track propagating interfaces into a bi-dimensional or three-dimensional space (Sethian, 1999). In the FACETS plugin (Dewez et al., 2016), the fast-marching algorithm is employed to create polygonal planar surfaces interpolating subsets of the point cloud. The algorithm is based on four parameters that need to be calibrated for optimal results. In particular:

• Octree level defines the level of systematic recursive subdivision of the point cloud three-dimensional space, defined by its bounding box, in this case, a cube. Every level involves subdividing the box into 8 sub-cubes, that allow optimizing the definition of the smallest feature we want to detect (i.e. the scale of analysis), with the computational time increasing with the level. No specific strategy exists to calibrate the octree level. As a starting guess we should chose a value that results in cubes with a dimension comparable to the smaller fracture facet we want to detect. From this value, it is possible

to decrease or increase the octree level by one level, visually checking the results. In our experience, increasing too much the octree level does not increase the quality of the analysis but will result in an over-segmentation of the facets and possibly in an increase of noise.

• Maximum distance defines a generalization criterion to merge adjacent features. For instance, if maximum distance is set at 68%, at least 68% of the points associated to a facet must have a distance to the facet mean plane that is lower than the standard deviation of the distances from the mean plane fitted from the points defining the facet. In geological terms, this parameter controls the maximum roughness accepted for a plane to be fitted. The maximum distance parameter can be calculated by manually isolating a certain patch of points, representing a fracture plane. The distance from the mean plane of every point can be manually calculated by fitting a mean plane to the point patch. The result is given in the form of a scalar field associated to the point cloud. The mean distance is calculated by fitting a Gaussian model to the frequency histogram of the previously calculated distances.

- Minimum points per facet defines the minimum number of points needed to define a facet. The higher the octree level, the smaller this parameter should be, as the dimension of the smallest feature detected decreases and so the related number of points. This parameter can be considered a threshold between what we consider as noise, and what we consider as a proper feature. The minimum points per facet parameter must be tuned according to the average surface density SD of the PC-DOM. The higher the surface density the higher will be number of points in the smaller element produced by the octree subdivision, therefore the higher this parameter can be set.
  - *Maximum edge length* is related to the length of the boundary of the facet. Small values of this parameter impose concave and compact boundaries, while larger values allow for elongated and/or convex boundaries. There is no general rule for the calculation of this parameter, which must be empirically calibrated on a case-by-case basis.
- Calibrating all these parameters on the whole point cloud is taxing in terms of computational time, therefore we suggest selecting at least 30 representative facets (Fisher, 1992), in terms of dimension and roughness, for every fracture set, calibrating the algorithm parameters on these facets, and then use this calibration to process the whole PC-DOM. When working on a single facet, the octree level must be set to 0, as an isolated facet is considered as a point cloud on its own (Figure 6).
  - The result of the feature extraction algorithm is a set of 2D polygonal facets resulting from the interpolation of point patches that met the criteria defined in the calibration, from which orientation parameters will be obtained (Figure 7E).
- It is important to remember that facets can be interpolated only on fully exposed planar structures, therefore they represent only the part of the fracture plane that shows a morphological expression. Moreover, based on the calibrated parameters there is the possibility that the interpolation of an exposed surface will result in a combination of facets, and not a single one. All of this to say that data like faces height and surface extension can be useful but should be handled with care if trying to obtain volumetric parameters ( $P_{30}$  or  $P_{32}$ ) or height distributions.

Figure 6 Example of the parameter calibration on a single facet. The max distance is calculated as the mean of the frequency distribution of the distances from the mean plane. (A) Example of a planar feature extraction when the max edge length parameter is too low. (B) When the max edge length is too high the planar feature results in a non-representative polygon. When the parameters are correctly calibrated the output planar feature precisely follows the point cloud border and no point is excluded from the interpolation for a too low max distance value.

#### 4.5 Orientation parameters calculation

In structural studies of fracture networks, is common practice to assume that each fracture set shows an orientation distribution described by the unimodal circular symmetric Von Mises-Fisher distribution on a sphere (Fisher, 1953) - a specific declination of the more general Von Mises distribution (Mardia and Jupp, 2000) that, following a common practice in geological applications, will be called "Fisher distribution" in the following.

Even if it is sometimes reasonable to assume that fracture sets follow a distribution with circular symmetry, with the exception of particular situations like radial dikes and fractures formed in a flat layer before the onset of folding (e.g. Mandl, 2005), a statistical test is needed, particularly if the final goal is to use the results of orientation analysis in downstream simulations. Fisher and Best (1984) proposed a goodness-of-fit test for the Fisher model, starting from a previous graphical test developed by the same authors (Lewis and Fisher, 1982). The poles of the family of n planes that need to be tested ( $I_j$ ,  $D_j$ ) with mean dip and direction ( $\bar{I}$ ,  $\bar{D}$ ), are rotated to obtain vectors with new coordinates ( $I'_i$ ,  $D'_i$ ), with mean dip and dip direction (0,0). The

original vectors are than rotated a second time to obtain a new set of vectors  $(I''_j, D''_j)$ . On these rotated values, the following derived datasets are tested:

• 
$$S_E \equiv \{c_i' = 1 - \cos D_i', 1 \le j \le n\}$$
 (5)

$$\bullet \quad S_U \equiv \{I_j', 1 \le j \le n\} \tag{6}$$

$$\bullet \quad S_N \equiv \left\{ Z_i = D_j^{\prime\prime} \sqrt{\sin D_j^{\prime\prime}}, 1 \le j \le n \right\}$$
 (7)

 $S_E$  is tested with the Kolmogorov-Smirnov test (Stephens, 1974) against an exponential distribution  $E(^1/_k)$  to check the underlying colatitude distribution (exponentiality test). The Kuiper test (Stephens, 1974) is applied to test  $S_U$  against a uniform distribution  $U(0,2\pi)$  to check the assumption of rotational symmetry around the mean vector (circularity test). The goodness-of-fit of  $S_N$  to a normal distribution  $N(0,\sigma^2)$  is tested with the Kolmogorov-Smirnov test (Stephens, 1974) to check against the correlation between colatitude and longitude (normality test) (Figure 7).

Overall, this procedure provides a quantitative way to assess if the dataset can be fit with a Fisher model, allowing the calculation of the mean dip and dip direction of the cluster and the concentration parameter k.

Figure 7 Example of the goodness-of-fit test for Set 2 and Set 3a of the Pontrelli quarry. The hypothesis of sphericity of the data is rejected based on all the tests.

If the Goodness-of-fit test is rejected, it is possible to fit a more general distribution. The Kent distribution is a natural extension of the Fisher distribution as it represents the analogue of a general bivariate normal distribution on a sphere (Kent, 1982). It includes the additional parameter  $\beta$  that describe the ovalness of the distribution, allowing to fit more elliptical clusters (Kent, 1982). No goodness-of-fit tests exist to check if the data follow the Kent distribution. Noteworthy, the Kent model results in

orientation distributions that are similar to those off the Bingham-5-parameter distribution (Kent, 1982), but has a different mathematical formula.

#### 5 Enhanced interpretation on orthomosaic and DEM

- Collecting orientation data directly from outcrops that lack a noticeable 3D morphological expression (i.e. the facets discussed above) is not possible. At the same time, measuring fracture size and intensity or density (length, height, *P*<sub>21</sub>, etc.) from PC-DOMs is not reliable because facets do not correspond to complete fracture surfaces. For these reasons, we digitize fracture traces on TS-DOMs, obtained by projecting and merging the images collected during the photogrammetric survey onto a polygonal mesh or a DEM (in turn interpolated from the PC-DOM). The data extracted from fracture traces are different and complementary to those provided b fracture traces, and only combining both kinds of information we can extract the most complete datasets from a DOM. The digitization of fracture traces on the vertical TS-DOM is done considering also the corresponding PC-DOM. By integrating TS-DOM and PC-DOM data, each digitized fracture trace can be associated with a best-fit plane derived from the point cloud. This approach enables the assignment of fracture traces to specific fracture sets. Fractures on the vertical wall that could not be reliably linked to a fracture plane were excluded from the digitization process.

  The dataset deriving from the interpretation of TS-DOMs is composed of an interpretation boundary (closed polygon) and a series of factors traces (a led lines) attributed to different fracture sets. The interpretation boundary (closed polygon) and a
- series of fracture traces (polylines) attributed to different fracture sets. The interpretation boundary limits the portion of the outcrop where fractures can be detected and digitized. It can include holes to isolate parts of the outcrop, covered by debris or vegetation, that are large enough to hinder the interpretation.
- In this case study we were able to obtain orthophotos (see Section 3) of both the sub-horizontal pavement and of sub-vertical walls (Figure 2), and this allowed carrying out the fracture trace digitalization in a 2D GIS environment.
  - The availability of both RGB images and DEM for the sub-horizontal pavement (from which we can derive slope, aspect and hillshade), can allow following structures that may be challenging to detect in RGB images only, due to alteration of the pavement surface, lack of colour contrast or zones damaged by quarrying activities, where longer fractures can still be digitized but are difficult to detect.
- Every fracture set is saved in a dedicated file and every characteristic pertaining to a specific set is recorded into an attribute table field. For example, both set 1 and set 2 include meso-faults and joints, this information is stored in an integer field coded as 1 for faults and 0 for joints. At the same time, set 3 is characterized by two main average orientations (Set 3a and Set 3b), but the fractures can be associated by their average length and abutting relationships with other sets. Fractures of Set 3a and 3b are than separated in a specific field.
- Precise termination (snapping in GIS jargon) of abutting fracture traces is managed automatically, defining a threshold distance quantified in pixels. In the following sections we discuss how we characterize topological relationships, length/height distribution, H/L ratio and P<sub>21</sub> from digitized fracture traces.

#### 6 Fracture network topology






Due to the limitations imposed by observing them in outcrop, the topology of fracture networks, which are actually composed of fracture surfaces embedded in a 3D rock volume, is most of the times characterized in 2D, from fractures traces limited and/or connected by nodes (Dershowitz and Einstein, 1988; Barton et al., 1989; Renshaw, 1996; Manzocchi, 2002; Sanderson and Nixon, 2015; Sanderson et al., 2019). Even under this limitation, topology is a fundamental component of fracture network analysis because it is directly related to connectivity, as demonstrated by Sanderson and Nixon (2015).

Topological relationships are also instrumental in calculating unbiased length and height distributions, because topology allows identifying censored fractures by means of B nodes (Benedetti et al., 2025), and this also cascades into the estimation of the H/L ratio.

### 6.1 Standard topological analysis

From a topological point of view, a fracture network can be seen as a connected set of branches (fracture traces) and nodes (terminations and intersections), delimited by an interpretation boundary (e.g. defined by the natural limits of an outcrop). Six main nodes categories can be defined in a fracture network (Benedetti et al., 2025; Forstner and Laubach, 2022; Nyberg et al., 2018, Figure 8):

- I nodes: fracture trace true tip points;
- Y nodes: abutting relationship;
- X nodes: crosscutting relationship;
- V nodes: perfect coincidence of two tip points belonging to two different fractures these are theoretically possible, but hard to recognize at the interpretation scale;
- B nodes: boundary nodes, where a fracture trace terminates at the interpretation boundary.
- C nodes: Contingent nodes that can be enabled or not, generating different fracture network configurations, depending on configuration rules defined according to the study objectives and sometimes micro-scale observations (Forstner & Laubach, 2022).

The nature of I, Y, X, V and C nodes is related to the processes that generate the fractures in the first place, but an additional consideration pertains to B nodes (Nyberg et al., 2018, Benedetti et al., 2025), which result from the interaction between the fracture network and the size and shape of the outcrop. This interaction leads to the formation of false tip lines (false I nodes  $\rightarrow$  B nodes) and the censoring of fracture traces. To prevent an underestimation of network connectivity, it is fundamental to exclude B nodes from the calculation of the relative proportions of I, Y, and X nodes.

Nodes classification is based on their topological value (Sanderson et al., 2019), representing the number of branches connected to each node. Specifically, I nodes have a topological value of 1, V nodes have a value of 2, Y nodes have a value of 3, and X nodes have a value of 4. B nodes can be categorized as nodes with a topological value of 3, but one branch originates from the fracture trace while the others come from the interpretation boundary. C nodes assume a different topological value depending

on the chosen configuration. If they are enabled, the topological value will be equal to 2 (V node), if they are not enabled, topological value will be equal to 1, and one C node generates two I nodes. This choice heavily impacts length and height distribution calculation as it is controlling the results of the topological analysis. Therefore, the decision about connecting or not fractures through C nodes should be made before running the topological classification.

FracAbility (Benedetti et al., 2025) is an easy-to-use original Python library developed for the quantitative statistical processing of fracture networks. Taking as input a set of polylines for each fracture set and a polygon representing the interpretation boundary, it is possible to obtain:

- IYX ternary diagram: the standard classification when it comes to topology (Figure 8A). In this diagram, the normalized distance from each vertex represents the relative frequency of I, Y and X nodes.
- Connectivity Index (CI): the mean number of connections per line, rendered as contour lines on the IYX ternary diagram (Manzocchi, 2002). As a reference, CI = 3.57 was defined as the critical CI for a constant length uniformly clustered system (Manzocchi, 2002).
- Backbone: the largest cluster of connected fractures in the network.




These results represent different aspects of the degree of connectivity of the fracture network. The ternary diagram and the connectivity index give information about the average connectivity of the network. Only when a fracture abuts on another one it is possible to reduce the number of I nodes in the network, thus moving towards the X and Y nodes vertices in the diagram means increasing, on average, the possibility to form large, connected clusters within the network. However, the presence of clusters of connected fractures with a high connectivity index cannot be unravelled by the simple node count. Backbone extraction addresses this problem by highlighting the most extensively connected cluster. Under the assumption that all fractures are open, it also provides a graphical solution to the percolation threshold problem. Specifically, if the backbone spans two opposite sides of the interpretation boundary, it indicates the presence of a giant connected component, allowing for the establishment of a continuous flow (Haridy et al., 2020). As illustrated in Figure 9, the backbone is characterized by a notable increase in the CI value.

Figure 8 Topological relationship of the Pontrelli quarry fracture network. (A) Associated IYX ternary diagram, the red dot represents the connectivity index (CI).

Figure 9 Backbone of the Pontrelli quarry fracture network. The backbone connects two sides of the pavement, indicating the presence of a giant connected component. (A) XYI ternary diagram of the backbone zone, the CI increase form 1.4 for the whole network to 3.5 along the backbone zone.

## 555 6.2 Directional Topology


Topological parameters presented in the previous section give a general picture of the fracture network as a whole and are calculated considering the fracture network as a single entity, not considering the geological classification of fractures in different sets (Figure 10A). It is thus impossible to retrieve information about a specific fracture set, for example, how many I, Y and X nodes a certain set have, or how a set is related to another one in terms of crosscutting and abutting relationships. This kind of information can be obtained using what we call "directional topology." In standard topological analysis, nodes store only the topological value. In contrast, in directional topology nodes also contain information about the fracture set (in

the case of I-nodes) or sets (in the case of Y- and X-nodes) from which the connected branches originate. This enables a more detailed topological characterization: I-nodes are classified by set, X-nodes are described by the intersecting sets (e.g., an X-node between Set 1 and Set 2), and for Y-nodes, it is possible to determine whether they are generated by Set 1 abutting on Set 2 or vice versa, by counting the number of branches (Fig. 10).

To address this issue, when splitting fractures into branches to calculate topological values, FracAbility stores into the node attributes the set to which every branch belongs and the associated directionality (Figure 10B).





Figure 10 (A) Topological relationships based on the topological value for a fracture network composed of two fracture sets. Highlighted in red, the branches necessary to define one specific node. (B) Topological relationship calculated taking into account the branch origin. X nodes are identified by the presence of two branches for every fracture set, Y nodes are identified by three connected branches. Of the three branches if only one pertain to a specific set it means that it is abutting on the other. I nodes are classified depending on the origin of the connected branch.

The usefulness of directional topology is not only limited to a more advanced description of the topological relationships within the fracture network, but can be also employed to define a quantitative parametrization of relative chronology between fracture sets, and of the stratabound versus non-stratabound nature of fractures.

In a hypothetical case where a fracture network is composed of two fracture sets, without censored terminations, and one of the two sets consistently abuts on the other, the abutting set will only show Y nodes. Thus, dividing the number of nodes by the number of fractures will yield exactly 2. In a real scenario, where the fracture network also includes censored fractures and B nodes, this result will be less than 2 because some of the Y-nodes are masked by censoring. To shield this relationship from

censoring, it is necessary not only to subtract the number of censored fracture traces from the total, but also the number of Ynodes that represent the termination of a censored fracture trace from the total Y-nodes, defining the following relationship:

Fracture Binding Index = 
$$\frac{n_{Y \, nodes} - n_{Y \, censored}}{2 \, (n_{frac} - n_{frac, \, censored})} * 100 = 100\%$$
 (8)

where  $n_{Y\,nodes}$  is the number of Y nodes of the abutting set,  $n_{Y\,censored}$  is the number of Y nodes associated to a censored fracture (i.e. trace with one B node and one Y node),  $n_{fractures}$  is the total number of fractures of the abutting set and  $n_{frac.\,censored}$  is the number of censored fractures of the abutting set. The Fracture Binding Index (FBI) ranges from 0, when no fractures from the abutting set abuts on the other set, to 100% when every fracture is abutting on another fracture set. 100% of abutting nodes is an asymptotic value, difficult to reach in a natural context, but nonetheless revealing a tendency in this direction would be interesting.

FBI can assume a different meaning depending on the context in which it is applied. In general FBI represents a quantitative way to assess relative chronology. In fact, considering two fracture sets, the one with the higher FBI is interpreted as being younger than the other one since it is consistently abutting. Moreover in vertical outcrops, if we consider the topological relationships between one fracture set and the bedding, FBI represents a quantitative parameter for the quantification of the tendency of a fracture set to be bounded by bedding surfaces.


Figure 11 (A) Directional topology applied to Set 1 fractures on Pontrelli vertical wall (B) Standard topological classification of Pontrelli vertical wall. The complete topological characterization is given by the combination of A and B visualizations.

Figure 11 shows the application of directional topology to the Pontrelli vertical wall, considering Set 1 and the bedding (S). Applying the directional topology analysis we obtain:

$$610 - n_{Y nodes} = 100$$

$$- n_{Y censored} = 28$$

$$- n_{frac.} = 93$$

$$- n_{frac. censored} = 40$$



$$FBI_{1-B} = \frac{100-28}{2(93-40)} * 100 = 67\%$$

Therefore Set 1 is stratabound at 67%.

Considering now, for example, the abutting relationships of Set 3 on Set 1 as mapped in the pavement:

- 
$$n_{Y nodes (3-1)} = 1161$$

-
$$n_{Y \ censored \ (3)} = 4$$

- 
$$n_{frac. (3)} = 1863$$

- 
$$n_{frac.\ censored\ (3)} = 12$$

$$FBI_{3-1} = \frac{1161-4}{2(1863-12)} * 100 = 31.2\%$$

On the contrary:





- 
$$n_{Y nodes (1-3)} = 183$$

- 
$$n_{Y \ censored \ (3)} = 1$$

- 
$$n_{frac. (3)} = 2003$$

- 
$$n_{frac.\ censored\ (3)} = 175$$

$$FBI_{1-3} = \frac{183-1}{2(2003-175)} * 100 = 4.9\%$$

Therefore Set 3 abuts with a FBI of 31.2% on Set 1, and on the other hand Set 1 abuts with only a marginal FBI of 4.9% on Set 3, which is considered just an effect of a few digitization errors or local deformational effects.

### 7 Trace length/height distribution

Defining an unbiased trace length distribution has always been one of the main challenges in rock mass and fracture network characterization. When calculating or estimating trace length parameters it is possible to distinguish between distribution-dependent (assume a specific probability distribution) and distribution-free methods (population parameters not linked to any specific probability distribution, Mauldon, 1998). On one hand, distribution-free methods do not rely on any specific assumption about the underlying distribution, but provide an unbiased estimator of only the mean trace length by an indirect correlation (i.e. length is not physically measured, Warburton, 1980; Pahl, 1981; Kulatilake and Wu, 1984; Mauldon, 1998; Zhang and Einstein, 1998; Mauldon et al., 2001; Rohrbaugh Jr. et al., 2002). Not making assumptions about the shape and mathematical form of the distribution could be seen as an advantage, but actually the non-parametrical nature of these

approaches implies that it is not possible to obtain statistical parameters of the population such as the standard deviation without imposing further assumptions (Pahl 1981). This makes distribution-free methods unsuitable for modern modelling applications, such as stochastic generation of fracture network, where a fully specified distribution is required. On the other hand, distribution-dependent methods make assumptions on the shape of the underlying trace length distribution, thus constraining their results. Because of this, it is necessary to test how well the chosen distribution fits the data. In the past this was a strong limitation, due to the biases discussed in Section 1.







Digital outcrops and the increasing computational power make it possible to overcome some problems that previous authors could only consider theoretically from a mathematical and stereological point of view. On one hand, these new techniques facilitate the acquisition of massive datasets on large sampling windows and successfully tackle the different biases that can be present on an outcrop. On the other hand, the increased computational power makes it possible to calculate the solution to mathematical problems that previously could not be solved due to the lack of a closed form solution (Baecher, 1980).

The orientation bias can be treated by applying areal sampling on outcrops with perpendicular faces. All the fracture sets perpendicular or sub-perpendicular to the horizontal plane are detected on the pavement. If present, fracture sets parallel to the horizontal pavement can be measured on the perpendicular vertical wall, eliminating the issue of under-sampling fracture sets with unfavourable orientations.

The size bias applies to 1D sampling methodologies (scanlines) where longer fractures have a higher probability of being sampled, but this bias does not apply to areal sampling strategies where everything inside the interpretation boundary is sampled. Even fractures much longer than the interpretation boundary are sampled and classified as censored fractures (see below). Areal sampling alone, however, does not account for the possibility of fractures parallel to the outcrop mean plane, and for the under-sampling of fractures shorter than the bed thickness (Ortega and Marrett, 2000). The association between the vertical and horizontal side of the outcrop can partially solve this bias. On the vertical side it is possible to check the presence/absence of a fracture set parallel to the horizontal outcrop surface and the relationship between fractures and the bedding interface. The problem remains for fracture sets parallel to the vertical outcrop mean plane, as the orientation bias hinders the trace mapping. Regarding our case study, we observed that the number of Set 1 fractures on the vertical outcrop roughly matches the number of fractures in the adjacent part of the horizontal outcrop. For Set 2 fractures, we can measure them on the vertical side but they are hidden by artificial fractures related to quarrying activities on the horizontal side. Set 3 is almost parallel to the vertical outcrop configuring the situation in which this bias cannot be evaluated.

Working with DOMs, the truncation bias applies to small fractures that can be truncated by limited DOM resolution. In our case, the resolution of the TS-DOM is around 4 mm/pixel and the smallest digitized fracture is 57 cm. Although the possibility remains that some fractures were missed during the digitization process — including potentially fractures smaller than the identified truncation threshold — the order of magnitude difference between the resolution of the DOM and the smallest recognized fracture is expected to mitigate truncation bias at a fixed scale.

Consequently, the only remaining bias to be treated is the censoring bias, which occurs naturally due to the finite nature of the outcrop or due to the presence of vegetation or debris. In statistics, censoring is a condition where the value of a measurement

is partially known. This occurs when some of the data are subject to limitations or restrictions, preventing us from observing the complete information. Censoring can happen for various reasons, and it is a common scenario in statistical analysis (Kaplan and Meier, 1958; Leung et al., 1997; Lawless, 2011). In our case, the fractures that touch the interpretation boundary are objects whose length is partially measured, therefore affected by random censoring (Benedetti et al., 2025).

We thus apply the theory and approaches described in Benedetti et al., (2025) to obtain an unbiased statistical model from censoring of both the length and height distributions. As discussed in Benedetti et al., (2025), this analysis solves the problems related to single censoring, double censoring, and those related to "holes" within the interpretation boundary, as depicted in Figure 12.


Figure 12 Different cases of censoring in a natural outcrop. The presence of information gaps affects trace length measurements.

Double censored fractures are considered a single censored fractures with one of the end nodes coinciding with the interpretation boundary. Fractures that look coplanar across an information gap are considered two separate censored fractures.

Survival analysis parameter estimation is based on optimization algorithms, like the Maximum Likelihood Estimation (MLE), where censoring is taken into account by calculating the likelihood of a censored measurement by using the survival function

instead of the probability density function (Benedetti et al., 2025). MLE is a parametric approach that needs a testing phase to validate its results. Working with censored data, without a specified distribution, leads to a situation where none of the standard non-parametric goodness-of-fit test can be applied (Benedetti et al., 2025).

Using the survival analysis approach different hypothesis (statistical models) can be estimated with the censored data. For this case study we propose to fit the following statistical models: Lognormal, General Gamma, Weibull, Exponential, Gamma, Logistic and Normal. Several statistical distances are calculated between the available empirical data and the fitted model to show which of the proposed models is more representative of the data. We chose to use the same distances as (Benedetti et al, 2025) thus using: Kolmogorov-Smirnov distance ( $DC_n$ ) (maximum distance), the Koziol-Green distance ( $\Psi_n^2$ ) (sum of squared distances) and the Anderson-Darling distance ( $AC_n^2$ ) (weighted sum of squared distances), with respect to a uniform distribution U(0,1) (2) Akaike information criterion (Benedetti et al., 2025).

#### 7.1 Distance from U(0,1)



The probability integral transformation theorem (Fisher, 1930) is a fundamental concept in probability and statistics, whose primary application is to transform the values of a random variable into a random uniform variable. The perfect model for fitting a dataset will follow a uniform distribution between 0 and 1, meaning a perfect correspondence between the empirical data and the theoretical distribution, and other models that are close to the uniform U(0,1) with a small deviation will be suitable to describe the data (Figure 13). Therefore, the purpose of the probability integral transformation is to normalize distributions in order to be able to compare deviations on a common ground.

Table 3 show the rankings based on the different normalized distances for Set 1 fractures. The lognormal distribution and the general gamma distribution rank respectively first and second in all the 3 rankings. In contrast, the logistic and the normal distribution are not suitable for our data. With intermediate rankings, we can appreciate the different meaning of the various distances; for example, the Weibull distribution shows a smaller maximum distance  $(DC_n)$  with respect to the exponential and the gamma distribution.

Figure 13 PIT visualization for Set 1 fractures.

| Name          | $DC_n$ rank | $\Psi_n^2$ rank | $AC_n^2$ rank | Mean rank |
|---------------|-------------|-----------------|---------------|-----------|
| Lognormal     | 1           | 1               | 1             | 1         |
| General Gamma | 2           | 2               | 2             | 2         |
| Weibull       | 3           | 4               | 3             | 3.33      |
| Exponential   | 4           | 3               | 5             | 4         |
| Gamma         | 5           | 5               | 4             | 4.67      |
| Logistic      | 6           | 6               | 6             | 6         |
| Normal        | 7           | 7               | 7             | 7         |

Table 3 Ranking based on the Kolmogorov-Smirnoff distance, Koziol-Green distance and Anderson-Darling distance for Set 1 trace length data.

#### 720 7.2 Akaike information criterion

The Akaike information criterion (AIC), is a criterion to rank models from the best to worst based on the empirical data (Akaike, 1974; Burnham and Anderson, 2004). AIC is designed to identifying the so called MAICE (Minimum information theoretic criterion (AIC) Estimate) (Akaike, 1974), as the model that give the minimum of AIC, defined as:

$$AIC = 2k - 2ln(L_{max}) \tag{9}$$

Where:

- *k*: number of model parameters
- $L_{max}$ : maximized value of the likelihood function





Therefore the Akaike Information Criterion favours parsimony, preferring models with fewer parameters that still adequately explain the data (Akaike, 1974).

Associated to AIC, there is another important parameter that gives the probability that a certain model is the best model for a given dataset, the Akaike weights  $(w_i)$  (Burnham and Anderson, 2004). These sum to 1 and have to be interpreted as a weight of evidence, meaning that the higher the value, the higher the probability that a certain model, in the pool of the selected models, is the best model for our data (Benedetti et al., 2025).

The main advantage of this method is that it takes as input the maximized value of the likelihood function, de facto ranking models and their relative parameters taking into account censored data.

| Rank | Distribution name | AIC         | $w_i$        |
|------|-------------------|-------------|--------------|
| 1    | Lognormal         | 7514.617939 | 0.9912701876 |
| 2    | General Gamma     | 7524.082426 | 0.0087298124 |
| 3    | Gamma             | 7636.568978 | 0            |
| 4    | Weibull           | 7653.734152 | 0            |
| 5    | Exponential       | 7655.821815 | 0            |
| 6    | Logistic          | 8804.536054 | 0            |
| 7    | Normal            | 9246.922169 | 0            |

Table 4 Ranking based on the Akaike information criterion for Set 1 trace length data.

Just looking at AIC values (Tab.4), and at the various distances measured in the previous section, it seems that even if the lognormal distribution wins, the three-parameters general gamma distribution is still a valid model for our data. Akaike weights clarify this situation showing that the lognormal model is much more powerful at describing our data with respect to the gamma model. At the same time, all the other models have Akaike weight equal to 0 meaning that with respect to the lognormal and the General gamma models, they are completely unsuitable.

# 8 Fracture areal intensity $(P_{21})$

Fracture areal intensity is defined as the ratio between the total sum of fracture trace length and the sampling area (Dershowitz and Herda, 1992; Mauldon et al., 2001). This parameter is very important since its volumetric equivalent  $P_{32}$  (total fracture area in unit volume) is used as a stopping criterion in stochastic DFN modelling, meaning that the stochastic generation of fractures will stop when the target intensity is reached, and  $P_{32}$  can be obtained form  $P_{21}$  via a calibration procedure (Staub et

al., 2002). Given the heterogeneous distribution of fractures in natural outcrops, the characterization of this value cannot be separated from the concept of Representative Elementary Volume (REV) or Area (REA) (Bear, 1975). In outcrop studies REA is the area above which a certain parameter value becomes independent from the position and scan area size with which it is calculated and thus the value can be used to constrain wider models.

To perform this analysis limiting the orientation bias we cover the outcrop surface with hexagonal grids of increasing edge length, ranging from 1m to 26m in the Pontrelli quarry case study. Only whole hexagons are considered and data are plotted using the graphical boxplot method proposed by Tukey, 1977 (Figure 14A). The lower threshold of REA can be defined as the minimum hexagon area where no significant difference is detected between the mean and standard deviation of  $P_{21}$  obtained at that area and at the next step.

To quantitatively measure the significance of this difference, statistical techniques like ANOVA, used to compare the mean of different populations, can be used in theory (Stahle and Wold, 1989; Moder, 2010). However, ANOVA is based on three assumptions: (i) hypothesis of normality, (ii) homogeneity of variances, (iii) independence between samples (Moder, 2010). In our case, P<sub>21</sub> samples collected with smaller scan areas are clearly asymmetrical (from 1m to 5m), while P<sub>21</sub> samples collected with larger scan areas tend to be more symmetrical (Figure 14A). Consequently, variance is inhomogeneous through the dataset, leading to an increase of type 1 errors (Moder, 2010). This problem is enhanced by the fact that the sample size is unequal and decays as the scan area edge length increases due to the finite size of the outcrop (the larger the outcrop, the smaller the number of scan areas). For these reasons, ANOVA and similar tests cannot be applied, and we decided to adopt a qualitative approach based on the difference between the interquartile range (deltaIQR) of two subsequential P<sub>21</sub> samples. With this approach, REA is reached when deltaIQR stabilizes around 0 (Figure 14B). To account for "far out" data, that are not included in the IQR, we also consider the range between the whiskers calculated as the difference between the upper whisker length (O3 +1.5IOR) and the lower whisker length (O1 - 1.5IOR) (Figure 14C).

In both cases, the REA correspond to a plateau that in our case study between 5m and 12m of scan-area edge length. Above 12m, the representativity is compromised by the too small sample size (

Figure 14 Representative Elementary Area analysis on Pontrelli quarry Set 1. (A) Boxplot of  $P_{21}$  data collected with increasing scan area size. Red dashed line:  $P_{21}$  calculated on the whole outcrop, with the interpretation boundary as scan area. The green box identifies REA range. Small number under the boxplot: sample numerosity. (B) Delta between IQR of two subsequential  $P_{21}$  samples. (C) Range between upper and lower whiskers for each  $P_{21}$  sample.

# 9 Height/Length ratio





The H/L ratio between height of fractures, measured along the dip direction, and their length, measured along strike, is the object of an extensive literature and is believed to span from 1:2 (Odling, 1997; Panza et al., 2015; Giuffrida et al., 2020), to 1:4 (Panza et al., 2015) or even 1:5 (Boro et al., 2014; Smeraglia et al., 2021), depending and different mechanical hypotheses, but could be probably even more variable when the possible combinations of crosscutting/abutting relationships between fracture sets and with bedding are considered.

H/L ratio is applied, in association with the length distribution, in most commercial and open-source 3D stochastic DFNs to model the geometry of the discontinuities, often represented as rectangular or elliptical surfaces. Therefore, the H/L ratio is directly correlated to the shape of the fracture planes generated by the DFN and controls the switch in dimensionality between 2D and 3D models. Unfortunately, the H/L ratio cannot be directly measured in outcrops due to the impossibility to map the full extension of fracture surfaces (but only their traces or partial facets).

In the studied outcrop, the availability of both height and length data allows us to make at least some realistic and transparent assumption on the H/L ratio based on a correlation of length and height distributions.

Our assumption is that traces mapped on the pavement (i.e. lengths) and on the wall (i.e. heights) can be associated in ordered pairs from the shortest to the longest. Making some assumption of this kind is unavoidable since there is no way to directly

observe the correspondence between horizontal and vertical traces. We want to stress that this criterion is not unique, and other relationships can be established between length and height data (e.g., random association), but this criterion seems reasonable from a fracture mechanics point of view.

To test our hypothesis, a hundred values of length and height are randomly sampled from the statistical distributions of length and height, ordered from smallest to largest and associated in pairs, and the H/L ratio is eventually obtained with linear regression (Figure 16A).





In our case study the height of Set 1 fractures should be limited by the height of the bed package, but the random sampling of the height distribution, that is a lognormal distribution, also generates, although with decreasing probability, fractures that are much higher. This is why in Figure 15 we limited the linear regression to height values smaller than 6m (height of bed package, Panza et al., 2016). One thousand realizations are made to account for the variability of the random sampling and the arithmetic mean of the regression line is taken as the representative H/L ratio (Figure 15B).

Figure 15 H/L ratio calculated for Set 1 fractures. (A) Example of one realization, where hundred values are sampled from the height and length distributions. r: Pearson correlation coefficient. (B) Frequency histogram of H/L ratios calculated from thousand realizations.

## 10 Summary of the results: fracture network characteristics at the Pontrelli quarry

In this section we describe the fracture network of the Pontrelli quarry and the results and limitations of the parametrization obtained with our workflow. Field observations show that Set 1 is the most prominent set in the outcrop, its fracture traces are homogeneously distributed across the pavement, and it can be detected even in areas damaged by quarrying. From field and DOM evidence, all other fracture sets abut or crosscut Set 1, therefore pinpointing this set as the older one. This conclusion is also supported by the topological, length and height analyses. Set 1 shows a lognormal length distribution with the largest average and maximum length, in agreement with a condition in which fractures are free to grow, in absence of a mechanical

compartmentalization defined by previously developed fracture sets, bedding aside (Ackermann and Schlische, 1997). Set 1 also shows a negligible percentage of abutting relationships with Set 2 (2.7%) and Set 3 (4.9%). This marginal number of unlikely relationships, in a geological context in which Set 2 and Set 3 postdate Set 1, can be explained considering a limited reactivation of Set 1 fractures in more recent tectonic phases. On the contrary, both Set 2 and Set 3 exhibit a significant number of abutting relationships against Set 1, respectively 21.41% and 31.2%.

The favourable orientation of the vertical wall supports the collection of a consistent statistical sample of height measurements, enabling reliable fitting of the height distribution. However, the height of the vertical wall is barely sufficient to get a complete observation window on a bed package. This limits the representativity of the assumption on the stratabound nature of Set 1. Our interpretation relies on the results from the directional topology analysis between Set 1 and the bedding. Since more than 67% of the observed Set 1 fractures abut on bedding surfaces, we believe that it should be vertically confined by the height of the bed package. In relation to classical height pattern classification schemes (Hooker et al., 2013), Set 1 falls between the perfectly bed bounded and the top bounded classes, given that even if the majority of the fracture about on the bedding, some fractures (33% of the non-censored fractures) end between two bedding surfaces. The H/L ratio for set 1 is calculated as discussed above from the trace height and length distributions, under the assumption that height and length values are associated in pairs from smallest to largest. This assumption is supported by the strong linear correlation between the two sets of values (Figure 15) and the resulting mean H/L ratio is 0.345.

| Set 1               |                     |               |               |          |        |        |         |
|---------------------|---------------------|---------------|---------------|----------|--------|--------|---------|
|                     | Test result         | Mean dip dir. | Mean dip      | Fisher K | Kent K | Kent β | n. data |
| Orientation         | rejected            | 213.94        | 86.3          | 36.59    | 39.59  | 5.42   | 1475    |
|                     | Set 2               | Set 3         | S (Bedding)   |          |        |        |         |
| FBI                 | 2.70%               | 4.9%          | 67%           |          |        |        |         |
|                     | Dist. Type          | Mean          | Standard dev. | n. data  |        |        |         |
| Length distribution | Lognormal           | 4.49          | 6.03          | 2014     |        |        |         |
| Height distribution | Lognormal           | 2.09          | 1.69          | 93       |        |        |         |
|                     | Value               |               |               |          |        |        |         |
| H/L ratio           | 0.345               |               |               |          |        |        |         |
|                     |                     | REA           |               |          |        |        |         |
| P <sub>21</sub>     | 0.7 m <sup>-1</sup> | 5-12 m        |               |          |        |        |         |

Table 5 Result summary for Set 1 fractures.

Set 2 mean length is intermediate between Set 1 and Set 3 and the topological relationships exhibit a higher occurrence of Y nodes against Set 1 with respect to Set 3, in good agreement with Set 2 being the second older set. The orientation of the vertical wall still allows to collect both faces and traces of Set 2 fractures, albeit the trace height dataset is less numerous than for Set 1, resulting in a less constrained height distribution. This is also reflected in the H/L ratio calculation, which should be applied more cautiously to stochastic modelling.

Unlike Set 1, Set 2 fracture traces in the northern part of the outcrop are masked by the non-systematic fractures generated by quarrying activities. Faces and traces data collected on the wall prove however that set 2 fractures are developed also in this sector of the quarry. The incomplete sampling of fracture traces across the pavement defines a strong bias that hinders the calculation of the  $P_{21}$  for Set 2, since without a sufficiently wide sampling area the REA calculation is not representative. Therefore, if we would model this fracture set with a stochastic approach, some assumption on the REA should be introduced. Most of Set 1 and set 2 fractures on the vertical wall abut on bedding surfaces. In particular, almost all the Set 2 fractures abut on the bedding surfaces, identifying it a perfectly bed bound fracture set in the height classification scheme of Hooker et al. (2013). This implies that bedding surfaces, in this context, actually have a control on the vertical development of the fractures. This is particularly true for the "high order" bedding surfaces that limit the bed package, where almost 50% of Set 1 fractures abut.

| Set 2               |             |               |               |          |        |        |         |
|---------------------|-------------|---------------|---------------|----------|--------|--------|---------|
|                     | Test result | Mean dip dir. | Mean dip      | Fisher K | Kent K | Kent β | n. data |
| Orientation         | rejected    | 75.37         | 89.31         | 26.47    | 27.91  | 3.1    | 1933    |
|                     | Set 1       | Set 3         | S (Bedding)   |          |        |        |         |
| FBI                 | 21.41%      | 9.69%         | 97.91%        |          |        |        |         |
|                     | Dist. Type  | Mean          | Standard dev. | n. data  |        |        |         |
| Length distribution | Lognormal   | 2.094         | 2.2           | 913      |        |        |         |
| Height distribution | Lognormal   | 1.3           | 0.78          | 32       |        |        |         |
| H/L ratio           | 0.359       |               |               |          |        |        |         |
|                     | Value       | REA           |               |          |        |        |         |
| P <sub>21</sub>     | \           | \             |               |          |        |        |         |

Table 6 Result summary for Set 2 fractures.




Set 3 shows the highest number of abutting relationships against the other sets and the smallest average length, in agreement with Set 3 being the younger in this outcrop.

Given the unfavourable orientation of the vertical wall (sub-parallel to Set 3 average attitude), only orientation data from facets can be collected and the height distribution cannot be characterized. This implies that also the H/L ratio cannot be obtained. Finally, regarding  $P_{21}$  and REA, the same limitations as for Set 2 apply. Therefore, in case we would model this fracture set with a stochastic approach, many relevant assumptions must be introduced even if in general the quality of the outcrop is high.

| Set 3                |             |               |               |          |        |        |         |
|----------------------|-------------|---------------|---------------|----------|--------|--------|---------|
|                      | Test result | Mean dip dir. | Mean dip      | Fisher K | Kent K | Kent β | n. data |
| Orientation (Set 3a) | rejected    | 157.46        | 78.75         | 132.83   | 133.52 | 4.45   | 1269    |
| Orientation (Set 3b) | rejected    | 124.08        | 76.8          | 91.83    | 92.63  | 4.2    | 2123    |
|                      | Set 1       | Set 3         | S (Bedding)   |          |        |        |         |
| FBI                  | 31.2%       | 21.19%        | \             |          |        |        |         |
|                      | Dist. Type  | Mean          | Standard dev. | n. data  |        |        |         |
| Length distribution  | Lognormal   | 0.74          | 0.75          | 1863     |        |        |         |
| Height distribution  | \           | \             | \             | \        |        |        |         |
| H/L ratio            | \           |               |               |          |        |        |         |
|                      | Value       | REA           |               |          |        |        |         |
| P <sub>21</sub>      | \           | \             |               |          |        |        |         |

Table 7 Result summary for Set 3 fractures.

#### 11 Discussion



This contribution is focused on the geometrical characterization of fracture networks and in particular on the input parameters necessary to generate stochastic DFN models. The main goal of the paper is to provide quantitative methodologies that limit the user choices as much as possible through the implementation of statistical tests (e.g. orientation distribution). If statistical tests are not viable due to the violation of the underlying assumptions, other statistical parameters ( $P_{21}$  REA) or statistical distances from a non-parametrical estimator (length and height distribution) are provided. The presented methodologies are based on data collected from DOMs, both point clouds and orthomosaics. In the context of upscaling geometrical parameters, DOMs are a convenient framework when it comes to collecting data on wide outcrops, decreasing the time for the acquisition process, allowing data collection in areas inaccessible due to practical or safety reasons, and opening to the possibility of implementing automatic feature extraction methods or automatic classification methods (topology). For a complete

characterization of the fracture network, especially when targeted to fluid flow simulations, the geometrical parameters included in this contribution have to be integrated with further analysis, to characterize filling, mineralization and other characteristics of the network (e.g., microscale connectivity) that can be assessed with other type of techniques and at a smaller scale (Forstner et al., 2025). Our approach is based on a combination of field surveys and DOM surveys, the latter used to obtain large datasets to support statistical analysis, the former to guide the digital mapping, filtering the noise given by external factors (e.g. quarrying operations) and assigning every fracture to a specific set through geological observations (particularly kinematics and relative chronology).

In quarries some fractures are generated during excavation. It is thus of the utmost importance, for both genetic reconstructions and analogue modelling, to exclude fractures that are related to anthropogenic surface processes. In our case study, the measured fracture sets are in accordance with the existing literature on the area (Sec. 2). In the outcrop pavement are present no data zones, characterized by debris accumulations, where no fracture set can be detected. Other parts of the pavement are affected by quarrying activities, resulting in zones "saturated" by fractures with random orientations or distributed following a radial pattern (related to explosions). In these areas only Set 1 is clearly detectable, given the constant spacing and orientation, an average length higher than the other fractures and the centimetrical displacement. Set 2 and Set 3 are drowned by these artificial fractures and even if present it is difficult to reliably isolate and digitized them.

The high quality of the outcrop, with adjacent horizontal and vertical surfaces, was instrumental in testing techniques that represent, in our opinion, a step forward in collecting rich quantitative datasets and developing rigorous statistical treatments for many geometrical parameters of a fracture network (Table 1). On the other hand, we must also recall that for some parameters there are still limitations in data collection and analysis. Both these points are discussed in the following subsections.

### 11.1 Combined analysis of fracture traces and faces






The integration of facets and traces (collected both on horizontal and vertical outcrops) allows a complete characterization of the parameters listed in Table 1, while other approaches rely on the analysis of facets or traces only (e.g. Ortega et al., 2006; Boro et al., 2014; Martinelli et al., 2020; Smeraglia et al., 2021).

Orientation data have been collected on the vertical wall PC-DOM, where dip and dip direction of true 3D planes can be measured by fitting a mean plane to planar patches of the point cloud.

TS-DOMs enable the digitalization of fracture traces and interpretation boundaries on both horizontal and vertical outcrops at a fixed scale, corresponding to the resolution at which they were collected. Nonetheless, the proposed methodologies can be applied to different scales, from thin sections to satellite images, provided that data are organized as digitized fracture traces combined with the interpretation boundary. The integration between fracture traces and the interpretation boundary is fundamental to avoid underestimating the connectivity index by misinterpreting B-nodes as I-nodes and provides a fundamental input to identify censored fractures.

 $P_{21}$  is calculated on the pavement TS-DOM where the huge areal extension ( $\approx 18.000 \text{ m2}$ ) enables to define a sufficient number of scan areas to detect the REA lower threshold.

# 905 11.2 Orientation analysis





The methodologies we suggest for orientation analysis are aimed at reducing subjective choices of the interpreter and at the same time exploiting semi-automatic data collection to increase the volume of data that support statistical analysis. This was achieved by:

- Introducing cluster analysis, in addition to classical structural observation, to segment fracture sets following a statistical criterion.
  - Calibrating an automatic feature extraction algorithm (FACETS) to maximize the data that can be extracted from PC-DOMs, avoiding the generations of artifacts (e.g. planes with an intermediate orientation) as sometimes happens in workflows tested by previous authors (e.g. Menegoni et al., 2019; Panara et al., 2024).
- Rather than assuming circular symmetry and fitting a Fisher distribution without prior statistical verification, our approach
   explicitly tests the fitted orientation distributions using goodness-of-fit tests. This provides a statistically grounded assessment of fracture set orientation parameters.

Our rigorous analysis revealed that, in our case study, no fracture set follows a Fisher distribution (Tab.5, 6, 7), in contrast to the conclusions of previous studies on the same outcrop (Panza et al., 2016; Zambrano et al., 2016). At the same time, Kent distribution parameters indicate a low ovalness for all fracture sets, even if Set 3a and Set 3b seem to be almost elliptical clusters (Fig 5e). The reason is that sets 2, 3a, and 3b are strongly asymmetrical and set 1 is multimodal, as highlighted by contour plots in figure 5.

The applicability, and thus the quality of the results produced by the automatic feature extraction algorithm, strongly depend on the ability to distinguish and characterize each fracture set within the network. In this study, reliable results were obtained by clearly distinguishing fracture sets through the integration of field data, DOM-derived data, and clustering analysis. In more geologically complex settings, where fracture sets are less well defined, caution is advised both when applying the clustering algorithm—since the number of sets must be specified a priori—and when using the automatic feature extraction algorithm.

In the context of generating analogue fracture sets with stochastic modelling (e.g. in DFN), where only the Fisher distribution can be modelled by standard software, these parameters should be handled with care. Using a Fisher distribution with a K parameter small enough to fit all the planes in the cluster (i.e. with a large spherical variance) will result in artifacts along the minimum axis of the elliptical distribution, or in not properly represented tails in case of asymmetrical distributions. On the contrary, using a K parameter that is too large (i.e. with a small spherical variance) will result in an underrepresentation of the oval tails of the data. All these problems will result in an incorrect modelling of connectivity, that is positively correlated to orientation dispersion (e.g. Smith et al., 2013).

A possible workaround for this problem, using the available software, could be to split the oval clusters and fit multiple Fisher distributions to model their parts, possibly validating the results generating synthetic clusters and comparing them with the

natural ones. More in general, more advanced distributions such as the, Kent, Bingham-5-parameters, Bingham-8-parameters or mixed Bingham can be adopted to fit asymmetrical or multimodal clusters (Kurz et al., 2014; Gilitschenski et al., 2016; Yamaji, 2016), and we feel that adopting at least the Kent distribution in stochastic modelling applications would be a significant improvement along the path of creating realistic stochastic fracture networks.

### **11.3 Topology**

In the ongoing discussion on topological analysis in fracture studies, some authors (e.g. Sanderson and Nixon, 2015) proposed to consider the connectivity of fracture branches, instead of full fracture traces, due to a supposed uncertainty in unravelling crosscutting, abutting, or splay relationships when branches form a small angle. Other authors (Forstner and Laubach, 2022) suggest considering also contingent nodes (C nodes) that would allow merging individual small branches to form larger traces, based on the considered scale and/or diagenetic consideration. In this context, we like to recall that (i) linear traces or branches represent the intersection of fracture surfaces with the outcrop surface (e.g. Sanderson and Nixon, 2015), and that (ii) most of the time we are actually interested in fracture surfaces rather than in their traces. In other words, branch connectivity parameters and length distribution can be useful information to characterize 2D connectivity of lines, but we would like to stress that considering branch length data as if they were full trace lengths would dramatically underestimate the dimensions of the underlying fracture surfaces, overestimate fracture density (since branches are more numerous that full traces) and would not allow performing directional topology analysis on a per-set basis, as discussed above. For these reasons we insist in considering fracture traces, relying on geological information collected in the field to solve the ambiguity highlighted by Sanderson and Nixon (2015).

Directional topology adds a further improvement to the fracture network characterization by assigning every node to specific fracture set(s). This allows quantifying crosscutting and abutting relationships between different fracture sets and understanding how the different fracture sets contribute to the overall connectivity of the fracture network. At the same time, it is possible to derive parameters like the FBI that express the relative chronology and relationships with bedding surfaces in a more quantitative way. The necessity of excluding censored fracture traces from the FBI calculation might result in either an overestimation of this index, in case the longest traces tend to have I-nodes, or an underestimation in case they tend to terminate with Y-nodes. Unfortunately, this effect is difficult to assess at the moment, and we leave a more detailed analysis for future studies.

Extracting the backbone of the trace network as in Figure 9, i.e. the largest connected cluster of the network, seems a really interesting result, and is possible thanks to the code we developed (https://github.com/gecos-lab/FracAbility). However, we would like to raise some cautionary note about these results. In fact in our outcrop the backbone is located along one of the major structures present in the pavement but, at the same time, it has been detected in a zone of the pavement that is particularly clean and free from quarrying-induced fractures, and it is bound by zones where the possibility to detect smaller fractures belonging to Set 2 and 3 is more limited. Therefore, there is no way of knowing if the shape and position of the backbone are due only to the presence of a major structure, or if they would change if other areas of the outcrop were not disturbed. In

conclusion, the backbone detected by our analysis probably represents a subset of what it could be, if a complete undisturbed dataset was available. This also means that the connectivity index of the whole fracture network is probably underestimated, even in such a high-quality outcrop.

### 11.4 Length and height distributions





Of the four major biases that hinder the definition of a correct length and height distribution, censoring is the only one that cannot be addressed by changing sampling strategy (as for size and orientation bias) or improving data acquisition techniques and checking the quality of the input data by quantifying the resolution of the TS-DOM with respect to the smallest fracture that can be detected (truncation bias, Section 7). This has led to excluding censored data (Bisdom et al., 2014) or considering censored fractures as complete ones (Panara et al., 2024; Smeraglia et al., 2021), resulting in statistical models that always underestimate length (Benedetti et al., 2025). At the same time, non-parametric approaches do not provide a fully specified parametric distribution (Mauldon et al., 2001), that is a fundamental input data in applications such as stochastic models, and also to evaluate the meaning of data in general (i.e. knowing the mean without any hint on standard deviation is meaningless). The censoring correction obtained by Benedetti et al., (2025) with survival analysis allows to use the full extent of the dataset collected from TS-DOM, and obtaining an unbiased statistical distribution.

In this contribution we leverage the availability of both pavement and vertical wall exposures, obtaining an H/L ratio specific for our outcrop. This is a significant improvement with respect to previous approaches, where the H/L ratio is generally assumed based e.g. on theoretical mechanical considerations, without any comparison with empirical data (Odling et al., 1999; Schultz and Fossen, 2002, and references therein).

The approach we propose, however, is not completely immune from a-priori assumptions. In fact, even in ideal outcrops, it is impossible to actually measure H/L of a single fracture surface because either we see the direct connection of the fracture surface in the vertical and pavement exposures – but in this case both traces are censored, or we see complete traces – but in this case we cannot see the connection. Our assumption is that height and length statistics must be correlated according to size rank, as discussed above in Section 9, and we believe that this is a reasonable assumption based e.g. on mechanical considerations (Odling et al., 1999) and results of our linear regression (Figure 15) but it is nevertheless important to state this transparently.

## 11.5 Fracture intensity and representative elementary area

Areal fracture intensity is often estimated using methods based on scan lines, scan areas or circular scan line (Rohrbaugh Jr. et al., 2002; Zeeb et al., 2013). These methods provide a minimum scan area size for a representative estimation of  $P_{21}$  based on the mean fracture trace length. In this contribution we proposed a different approach, based on the concept of Representative Elementary Area to try to quantify the range of scan area size in which fracture intensity can be mediated to ensure a proper continuum-equivalent description (Bear, 1975).

As a partial correction to the approach in Martinelli et al. (2020), we recently noticed that the finite nature of outcrops determines limits in the collection of  $P_{21}$  data, such as the progressive decrease in scan areas numerosity as the scan area is increased, and the non-independence of  $P_{21}$  samples collected at increasing scan area sizes. For these reasons a quantitative approach based on formal statistical tests cannot be safely applied. Adopting a more qualitative approach will result in a less significant result, which partially depends on the interpreter choice, however, it does not require such stringent assumptions as the tests used by Martinelli et al., (2020). We also recognize that adopting a more qualitative approach may introduce subjectivity in the selection of window size, but still having an order of magnitude for the REA (and hence for REV) is important in modelling studies. Indeed, in addition to formal statistical reasons, defining the REA has important practical applications. For instance, the choice of the optimal cell size in reservoir-scale models stems from a combination of several factors, that are the geological characteristics of the area, lithostratigraphic heterogeneities, mean spacing between wells and the available computational power. In general, however, there is a lower limit to the resolution of the model - around 50 m of cell size, dictated by computational power. This is an important piece of information because it outlines the minimum size that an analogue outcrop should have to capture all the variability within a hypothetical cell. From this point of view, the Pontrelli quarry provides a pavement two to three times larger than the minimum cell size, granting a sufficient area to calculate representative statistics. At the same time, the  $P_{21}$  REA, determined for Set 1 fractures between 5m and 12m, is approximately five to ten times smaller than the minimum cell size, allowing a safe application of continuum-equivalent upscaling techniques.

### 12 Conclusions








In conclusion, this paper presented a series of quantitative methodologies to characterize fracture network geometry from Digital Outcrop Models (DOMs). The ideal conditions for applying our methodologies involve an outcrop that enables the collection of a statistically significant and complete dataset (depending on the scope of the work). This requires favorable orientation of the outcrop faces relative to the fracture set orientation, overall surface cleanliness (minimal debris, vegetation, or damaged zones), sufficient size to ensure adequate sampling, and the presence of at least two perpendicular exposures (horizontal and vertical). Although such conditions are challenging to achieve in natural settings, they should serve as guidelines for selecting a suitable outcrop. Among all the parameters required to fully characterize a fracture network, we focused on those required to generate 3D stochastic DFN models, that are: orientation parameters, topological relationships, length and height distribution parameters, H/L ratio and  $P_{21}$ :

- Orientation data are collected through a semi-automatic workflow, clustered with k-medoids, and tested for the goodness-of-fit to a Fisher distribution. Alternatively, the Kent distribution parameters are also provided. This procedure allows subjectivity to be removed from the assignment of dip/dip direction data to a specific fracture set and supports the choice of meaningful orientation parameters through the implementation of statistical tests.
- Topological relationships are calculated including the interpretation boundary, this allows to: (i) to define B nodes and exclude them from the connectivity index (CI) calculation (ii) to identify censored fractures in an

automatic way. Backbone extraction highlights the presence of large, connected clusters in the network. Crosscutting and abutting relationships between different fracture sets are quantified through directional topology.

- The approach developed to deal with censoring bias provides as a result a set of fully specified distributions (all parameters are explicit) corrected for censoring. The best model among the initial selection is defined through a graphical approach and a series of statistical distances.
- We demonstrate that estimating H/L is not possible without introducing some assumption, even for the best exposed set and in the presence of both horizontal and vertical exposures. Therefore, we opted to make our assumption as transparent as possible, and we tested it with regression analysis.
- P<sub>21</sub> REA is calculated with a qualitative approach to avoid violating the underlying assumption of more formal statistical tests.

### 13 Code and data availability

Codes and data are available at the following GitHub repositories owned by the Gecos-lab group of the University of Milano

— Bicocca (https://github.com/gecos-lab):

- FracAttitude: Python code for orientation data analysis available at https://github.com/gecos-lab/FracAttitude;
- DomStudioOrientation: Matlab code for orientation data analysis available at <a href="https://github.com/gecos-lab/DomStudioOrientation">https://github.com/gecos-lab/DomStudioOrientation</a>;
- FracAbility: Python toolbox for topology and survival analysis available at <a href="https://github.com/gecos-lab/FracAbility">https://github.com/gecos-lab/FracAbility</a>;
- FracAspect: Python code for H/L ratio calculation available at <a href="https://github.com/gecos-lab/FracAspect">https://github.com/gecos-lab/FracAspect</a>;
- FracElementary: Python code for P<sub>21</sub> and REA analysis available at <a href="https://github.com/gecos-lab/FracElementary">https://github.com/gecos-lab/FracElementary</a>.

### 14 Author contribution

AB, FA and BM conceived the first version of the orientation analysis workflow. SC, AB, GB, SM, FA and DB formulated the key concept of the paper and the final workflow structure. FLV, MM, FB and CA participated to fundamental discussion throughout the project. AB and SC collected the raw photogrammetric data. SC performed DOM interpretation to generate the dataset. SC wrote the manuscript. AB, SM and FA heavily contributed in the revision process and in defining the final structure of the paper. AB led the project and provided funding.

# 1060 15 Competing interests

The authors declare that there are no competing interests.

# 16 Acknowledgements

We thank Eni SpA for funding, reviewing and agreeing to publish the results of the study. We thank SABAP – BA (Soprintendenza Archeologia, Belle Arti e Paesaggio per la Città Metropolitana di Bari) for granting the access to the Pontrelli quarry, making this work possible. We also thank François Bonneau for the fruitful discussions, in particular regarding directional topology.

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
