# Peer review of "An Integrated Workflow for Parametrization of Fracture Network Geometry in Digital Outcrop Models"

_EGUsphere, 2025_

## Referee Comment (RC1)

The contribution proposes a 'complete and rigorous workflow' to acquire fracture information from outcrops to 'constrain models of subsurface fracture networks' by using the outcrops as analogs for the subsurface. The contribution is within the scope of the journal. The research methods are solid, the topic is of considerable practical and fundamental interest, and the MS is mostly well presented and has generally clear illustrations. The associated codes should be welcome by many. And this contribution should be widely read and used. There are a few places, however, where the text can be adjusted or added to so that the scope of the workflow is clearer and this workflow is situated relative to a really complete and rigorous description of outcrop fractures.

For example, the scope of the paper needs to be clearer. The Introduction text and the title give the impression that the work is a complete workflow for characterizing fractures in outcrop, but the workflow is actually narrower: the emphasis is extracting traces from 3D DOM representations of outcrops. This is a valuable element of outcrop fracture characterization but it is only part of the equation, and leaves out some key steps for acquiring fracture information from outcrops to constrain models of subsurface fracture networks. Possibly the authors allude to these other elements in what they call 'traditional direct geological and field survey' (line 116) but it is hard to tell because these supposed traditional methods are not adequately explained. This issue could be resolved by (1) revising the Introduction to more clearly state the scope and (2) provide at least a thumbnail sketch of the other elements in a complete outcrop fracture workflow. Much of this latter step can be accomplished by reference to the literature including perhaps a short synopsis of the steps mention in Peacock et al. (2022) and Elliott et al. (2025). I've provided further suggestions in the comments keyed to lines in the text.

The Introduction seems unfocused and could do better at setting the scope of the paper and drawing in the reader. It would be useful to revise the Introduction to end with a clear, explicit statement of claims (here we show that…). Currently the reader is not alerted to what the paper shows and so may be unmotivated to read further. Also, some of the material that is in the Conclusions is not obviously prefigured in the text or in the Introduction.

There are places in the text where the fracture trace map connectivity pattern is equated with flow pathways. This may be ok if the paper were about flow modeling on a hypothetical network where all the fractures were specified as being open. But it is a problematic step in an MS about collecting data from outcrops; the fractures in outcrop may or may not be open or partly open. So the text about connectivity and flow needs to include this caveat.

Comments keyed to lines in the text

29 The Introduction. Think about creating an opening line that points the reader to the focus of your paper and organizing the elements of the Introduction from general to specific. Line 36 is the most general and ought to be first: '*Fractures exert a fundamental control on mechanical and hydraulic properties of rock*' and '*knowledge of fracture attributes has application to many societally important engineering operations*' (cite a few references, or put all the references from lines 37-45 in a table. This long text list of applications goes off topic and some of it is repetitive.) Then cover what is important about fractures: '*The effects of fractures on strength and fluid flow depend on several factors, including mode (fault versus opening-mode), mineral fill, orientation, size, and spatial arrangement* (some references).' Then focus in on your topic: how fractures are arranged in space. You might want to consider how topic is covered in the 2018 J. Struct. Geol. review of spatial arrangements of faults and

opening-mode fractures: '*The arrangement of fractures in space and in relation to one another into networks…*' note that size and spatial arrangement are challenging or impossible to document using well data and then go to something specific about what your paper provides. Much of lines 68 to 109 seem distracting for an Introduction. These thoughts may be more effectively explored in the Discussion.

30 This definition of 'fracture' seems overly broad. Useful definitions to some extent depend on the application and how results are going to be used. The Schultz 2019 definitions might be useful in rock mechanics/excavation/engineering setting, but seem to me to be the wrong place to start if your objective is using the outcrop as an analog for the subsurface in the applications that you list next. And there are other uses for outcrop fracture characterization that might use different definitions. For example, for geomorphologic work—which uses many of the same analytic tools of topology, spatial arrangement, and aperture measurement—alternate categories are useful (see for example, the workflow paper by Eppes et al.).

So I suggest you adopt a more structural geologic definition (faults and opening-mode fractures) and mention that for other applications workers may need other terms or definitions.

Not all bedding constitutes discontinuities. And why do you say foliations are a primary feature and fractures secondary? Foliations are certainly in the same 'secondary' category as foliations. Later in the text you do not consider bedding to be 'fractures' so you don't seem to be following your own definition.

30 'where'?

35 for the 'filling' aspect these references seem inadequate and moreover, it has already been shown that using mineral deposits alone to help define sets is quite misleading for several reasons. I suggest you call out the Reviews of Geophysics 2019 paper here, which is already in your reference list (line 1185). As for the other citations, I think citing textbooks is not favored. And you already have the Hancock 1985 review in your reference list with its classic summary of fracture set rules.

46 And there are some papers that are examples of extracting fracture information from outcrop specifically for these applications. How does your work differ from or advance from these other studies?

47 'and rock strength.'

31, 35, 55, 130 A complete description of a fracture network and a method for extrapolation from outcrop to subsurface ought to have a step in it where the diagenetic state of the outcrop host rock and fractures is documented. What is the diagenetic state of the host rock and what kinds, if any, mineral deposits are in the fractures? This is not a hard step to add. There are a number of examples in the literature that describe how to do it. It's just a matter of describing (or even reporting) host rock and fracture properties; and if for some reason this cannot be done (not even one thin section?) then at least the 'complete and rigorous workflow' could mark this as a gap. The paper need not be made much longer by mentioning the need for such a step. The MS already cites one paper that makes this point (Forstner and Laubach 2022) so this is not a matter of adding a citation.

And the case has been made in the literature, and I think it is hard to dispute at this point, that key parameters such as connectivity and length are modified in essential ways by cement deposits. This information, if possible, should be included in the basic network description and the topological

formulations. For example, there is a big difference between a network of two orthogonal joints sets, where all fractures are open, and a geometrically identical arrangement where the first set is filled (veins, for example) and the second is open, and a case where all the fractures of both sets are sealed. All of these cases have been found in outcrop. It's not helpful to the modeling or analog user community to report that all with connectivity indices based solely on the trace patterns.

As a brittle structural community and creators and users of fracture outcrop analogs we can't be satisfied with methods that ignore mineral deposits when core data shows that such deposits are a fundamental attribute of most of subsurface fractures that are of interest to practical applications. Moreover, the mineral deposits in fractures are one of the few features that can reliably be measured in both outcrop and subsurface. Such observations can be a useful part of relating outcrops to specific subsurface targets (Ukar et al., 2019; Elliott et al., 2025). Host rock composition and diagenesis is also something that you need to know in comparing mechanical and fracture stratigraphy from analog to target.

50 Table and 55-60 Text

This list of fracture properties needed is incomplete.

The table footnotes should do a better job of explaining what you mean by 'static/dynamic'. Overall the table does not seem very clear, and some aspects are questionable. For example, by starring 'network' for 'topology' but not 'sets' you imply that topology can (or should) only be documented for 'networks' but this way of thinking of the issue is limiting. For example, if there is only one set, you could still define the topology (it would be the topology defined by fractures in that set). Core data suggest that 'one set' is common in several basins (see papers by Laubach and by J. Lorenz from the 1990s). If you leave this circumstance out, you may be missing just what the analog is supposed to capture. And in many cases, the network in outcrop in not what you want to describe the subsurface. Instead, it would be better to isolate part of the outcrop network for topology analysis. The literature has plenty of examples of outcrop fracture studies where the first step is figuring out what weathering and other 'near surface' fractures ought to be omitted (or at least accounted for separately in topology analysis). There is a clear example by Lorenz. The studies of fractures in central Texas related to the SSC cite are another. In both of these examples, extracting the meaningful fractures to analyze has major practical implications for the usefulness of the analog study.

On the static versus dynamic, here you must mean 'on an engineering time scale', inasmuch as over geologic time scales all of these features are 'dynamic' as fractures grow and interact. But even on the engineering time scale how can you be sure that, for example, connectivity and length are static? And there is good evidence that in some reservoirs apertures are not particularly dynamic. So, while the static versus dynamic aspect is useful to think about, it seems like a red herring here where you point is that many of these extended attributes are difficult or impossible to measure in the subsurface with the kinds of probes we have. Do you need the static versus dynamic distinction later? Maybe better to omit and make the table about attributes you desire to measure.

64-66 This account of what can and cannot be measured could be more nuanced and do a better job of setting up what your paper contributes. Some aspects of fractures can be measured on the meso scale, like strike and dip, aperture, some aspects of abundance, and if the wells are deviated 1D spatial

arrangement.  That's not the same as 'cannot be measured'. The elements that can't be measured are length, height, and connectivity.

68 'fractures are not always…'

86 'fracture state'

90-109 This commentary on DFNs seems out of place here. Maybe it belongs in the Discussion. DFN's are something you build once you have information about the fractures (however incomplete) and so topically it seems out of place in a lead up to ways to improve characterization.

99 Is a fixed height/length ratio realistic? Doubtful.

111 Here and elsewhere where you mention 'in detail'; note that this is a vague usage. Omit or mention a scale range.

116-118 This is vague. I'm not sure what you mean. What are 'traditional, direct…' surveys? Outcrop fracture trace maps via surveying instruments and film have been acquired since at least the 1980s and although those methods are certainly slower than DOMs they may not be less accurate. Are you trying to say that previous studies of fracture statistics from outcrop that don't use your method are unreliable? If this is your point, then the comments belong in the Discussion after you have demonstrated this.

114 For comparing outcrop and subsurface, I don't think you want to say they 'underwent the same geologic history' since outcrops and subsurface targets by definition have different geologic histories, and the differences could have a material effect on what fractures are there. Uplift, contraction, weathering and a bunch of other processes must differ from the target to the outcrop. See English, 2012, Engelder, 1985, and Eppes et al. 2019 for discussions of various aspects of these differences. Peacock 2016 and Elliott et al. 2015 have text that describes how the inevitable differences can be accounted for. The reality is that outcrops and targets will always vary in important ways and one of the steps in a 'rigorous workflow' needs to be collecting data that will allow the nature of these differences to be identified. That way, the applicability of the analog can be judged, and, in some cases, the outcrop patterns can be adjusted to match the subsurface situation (as in the example in Forstner and Laubach, 2022, a study you cite).

116 I don't know what you mean by the phrase 'unavoidable as it is limiting' and how does the 'traditional direct geological and structural field survey' differ materially from flying the outcrop with a drone? In any case, 'relative chronology' if you mean crossing and abutting relations can be estimated from images and 'mineralization/filling' is best done in the laboratory with a thin section. And if by 'geometrical datasets with traditional techniques' you mean fracture trace maps, this was accomplished in the past for large outcrops without drones or digital outcrop models (see Barton from the late 1980s) although no doubt current technology makes collecting such information easier. So these sentences need adjustment so as not to be misleading.

Either way, access to advanced image collection methods do not solve two big limitations the use of outcrop fracture tracer mapping: the finite size of most outcrops and the potential for fractures in outcrops to be unrelated to the subsurface. These caveats ought to be mentioned.

Provide a definition of what you mean by 'traditional direct geological and structural field survey'.

130 The actual scope of the paper needs to be clarified here. And mention what parameters you are leaving out.

132 Define the 'traditional direct field…'

135 Is it really correct that factors such as connectivity are not incorporated in DFNs? Some DFNs can incorporated aperture size variation. See papers by Sweeney et al, for example, 2023.

137 Awkward wording: 'minimizes assumptions at each step'.

138-146 This would be more compelling if you could state your specifics claims: 'here we show that…'

163, 425, 638 Describing and classifying height patterns should be a step in outcrop network description. There is a useful height classification in Hooker et al. (2013. J. Struct. Geol.)

175 What do you mean by 'distinctly younger'? This makes it seem like you can detect a gradation in age, but all you have is an abutting relation but no information about how much younger. Geomechanical modeling shows that such relations can arise in a single deformation sequence or can reflect much longer times. Check the text and remove such unjustified qualifiers.

181 Forstner et al. 2025 GSL Energy Geoscience Conference Series, v. 1 explicitly investigates the effects of such patches on length distribution statistics. A good workflow needs to establish and describe how continuity across these features is treated.

183 'drowned by artificial fractures' is causal and vague. Can you restate this?

249 I think the text here could be confusing. You mean the best parameter to extract information from your image, but the way you put it ('most important parameter for applications in structural geology') it sounds like a geologic or fracture parameter and the reader asks The most important parameter for what? Most important probably depends. Is this qualifier even needed here? The remark seems more appropriate for the Discussion, where it can be defended.

This sentence for definition of 'surface density' could be clearer. I take it that this is an imaging parameter but it could be read as a fracture abundance measure. If what you mean by this is 'where the most fractures are' at least in terms of fluid flow there are examples in the literature where production data shows that numerous fractures does not correlated with, for example, high fluid flow (Wang et al. 2023, Marine & Petroleum Geology).

270 Add graphic bar scales to the outcrop images.

424 This sounds like a serious problem.

503 'trace connectivity' and their possible effects on 'fluid flow' are two different things and ought to be more carefully separated in your description. Fully connected traces may not imply any enhancement in flow (for example, if the traces are faults or sealed fractures); disconnected open fractures can enhance flow if the host rock is permeable. Please be more careful in how these parameters are portrayed.

510 Include contingent nodes in nodes list. These kinds of nodes go back to at least Barton and Hsieh.

515. I think you mean 'hard to recognize'. Since many fracture arrays grow by linkage such connections my be common; and in fact, most opening mode fracture traces have evidence at a range of scale that

they are made up of end-to-end links. See papers by Olson and Pollard and Lamarche et al. If you increase the image resolution, single traces bounded by I-nodes may reveal numerous low angle Y nodes (see Forstner et al. 2025, their figure 6c.

516 'C-nodes', which you mention later, ought to be listed here. They certainly reflect the same level of abstraction as these other node types.

519 By 'external processes' do you mean 'the size and shape of the outcrop'? Why not just say that? It's less obscure.

540 This assumes that all the fractures are open. It's one thing to talk about traces and how connected thy may be, but it's quite a jump to assume 'percolation'.

594 Omit 'very' as vague. This interpretation of relative ages by abutting and crossing is confusing. Crossing relations and abutting relations amongst have the same implication for relative timing: the abutted fracture is older and the crossed fracture is older.

605 Where do these patterns fall on a height classification scheme?

641 'rely'?

641-642 Hel the reader by succinctly defining what these distinctions mean.

562-564 This section of text could use some clarification.

650-655 The data sets are still constrained by the size of the outcrops. Some early studies captured complete fracture inventories within large and clean outcrops (e.g., Barton, others). So these claims about 'massive' data sets seem like they are missing key elements of the problem.

660-663 But this does not solve the conceptual problems of measuring lengths as laid out for example in Ortega and Marrett 2000.

666 These size/resolution issues can affect length distributions. This is obvious when you collect fracture information at different scales, and it follows from the segmented character of most fractures and the tendency for fracture length to grow by linkage. See the example in Forstner et al. 2025 where drone, hand held LiDar, and scanline datasets cover the same fracture array.

In any case, saying that you know where your data are truncated is not the same as being able to claim that it can be safely assumed that truncation bias is not affecting the dataset.

672 It might be random censoring. But it's useful to wonder whether where you have continuous outcrop and where not is likely to be random, given that in many environments plants (and cover) may be localized in fractures (or in one case I know of, where the wide fractures are). It's a geomorphology and vegetation issue that should not be lost sight of in fracture trace collection; see the comments in Eppes et al.

680 See the Discussion in Forstner et al. 2025 of this problem. If what you are doing is just connecting traces that look coplanar across outcrop gaps, that is problematic.

737 Here and elsewhere you can omit 'very'; it is vague and not needed.

835 'implies'

849 The true scope of the workflow needs to be stated here and in the Introduction. What you are describing is really only part of a comprehensive workflow for describing fractures.

855-858 So how did you accomplish this filtering? This is a general problem, and not just restricted to outcrops in quarries. Spurious near surface fractures can be in regular sets. See the protocols in Eppes et al. and in the older literature.

865 The claim 'complete characterization…' of 'fracture network parameters' seems overstated and vague. What parameters? Do you mean 'heights and correlated lengths' and associated connectivity patterns? In sedimentary rocks the bed-normal patterns of heights are commonly quite different from length patterns. Heights may or may not be correlated with stratigraphic features that (depending on depositional environment, etc) may be on a meter or less scale, whereas sedimentary boundaries have been shown to affect length patterns but on a longer scale (again depending on the scale of the sedimentary features). An outcrop trace pattern example of the later is in Geol. Mag. 2016, v 108., p. 135, their fig. 2.  This is one reason why classifying height *patterns* (i.e., height relative to stratigraphic features) should precede drawing conclusions about height/length relations.

866 'only one of these two data sets' is confusing. It's not clear what 'datasets' in the cited references that you mean. Do you mean 'that rely on data collected on either bed-parallel outcrops or in cross section'?

859 'very high quality' is vague; omit 'very'; and 'perfectly exposed' seems to contradict your Conclusion, where you note that there are gas in the outcrop. Even the best outcrops have covered areas and exposure gaps, although there are some that are quite large and clean.

872 Is this typically done? I don't think this is the protocol in Healy et al.

874 Awkward. '…huge areal extension…' > 'xxxx m$^2$ extent' (provide an area rather than the vague 'huge' and it's 'extent' not 'extension'

866 On the other hand, some of these techniques provide information on key parameters that you do not address. Ortega et al. for example, provide aperture data over wide size ranges, but you apparently do not collect any width information so how can your technique be portrayed as 'a complete characterization of fracture network parameters'. Your contribution can stand on its own without overselling it.

870 But you ought to note the limitations; for example, your DOM is all at one scale. What happens if you collect fracture data over a range of scales and resolutions (e,g,. Forstner et al. 2025; Elliott et al. 2025).

911 The criterial is scale and diagenetic considerations. The latter point relates to whether the fractures are open.

906 It's not clear what your point is here. The text needs to be thought through a bit more. The contrast you draw between Sanderson and Nixon and Forstner and Laubach I think is ill posed. The problems with traces in 3D were already discussed by Ortega and Marrett 2000. The problem of *open* fracture continuity for opening mode fractures identified by Forstner and Laubach would be equally valid if you

could somehow see the 3d shape of fractures; this is not an artifact of looking at a 2D surface. The issues you need to address are on the most general level, that fractures intersecting a 2D surface (or fractures in 3D) may or may not be open. In other words, the fracture maps (and all the parameters derived from them) may be unrelated to fluid flow (or strength). Consequently, the only way to go from the outcrop analog fracture map to any kind of fluid flow estimation needs to account for whether the fractures are open. This can be done by simply assuming they are all open, as is usually done. Such an approach can lead to insights, but it contradicts geologic observations to say that the maps and derived parameters are all you need to estimate flow. Other geologic evidence can and should be brough to bear. The geologic evidence shows that what fractures (or parts of fractures) are open depends on scale (fracture width) so some fractures (or parts of fractures) may be sealed, so this information can be used to make trace maps more informative for flow parameters. Other geologic elements could be in play. For example, the 'network' might be a mixture of opening-mode fractures and faults.

943 On the other hand, Forstner et al. 2025 make the case that outcrop trace maps, however collected, always overestimate length.

962 Awkward. 'allowing to define' > 'defining'

966 The text starting "For these reasons a step behind…" is confusing. Clarify what you mean.

981 'every relevant parameter' is overstating this case, as you have not characterized the distributed aperture distribution or where the fractures are open versus not open.

984-988 Are these major conclusions? The first bullet item is no news to anyone who has measured fractures in outcrop and if this is a conclusion, the Introduction does not do a good job of setting up or prefiguring this point. Also, if this is an issue why not discuss the solution for describing and dealing with incomplete outcrop patches suggested by Forstner et al. 2025?

986-988 This statement does not seem to be posed as a Conclusion.

991 Where was 'tectonic related fracture' sets introduced? Why not stick to descriptive terms? How do you know that any of the sets are related to tectonics? I don't see anything in the MS that demonstrates when and why these fracture sets formed (and is it even relevant to a characterization workflow?).

1255/1260 Why are the article titles formatted differently? Make these uniform.

1265 Check the author's name.

References mentioned

Eppes, M.C., Rinehart, A., Aldred, J., Berberich, S., Dahlquist, M.P., Evans, S.G., Keanini, R., Laubach, S.E., Moser, F., Morovati, M., Porson, S., Rasmussen, M., Shaanan, U., 2024. Introducing standardized field methods for fracture-focused surface processes research. Earth Surface Dynamics 12, 35-66. https://doi.org/10.5194/esurf-12-35-2024

Elliott, S.J., Forstner, S.R., Wang, Q., Corrêa, R., Shakiba, M., Fulcher, S.A., Hebel, N.J., Lee, B.T., Tirmizi, S.T., Hooker, J.N., Fall, A., Olson, J.E., Laubach, S.E., 2025, Diagenesis is key to unlocking outcrop fracture data suitable for quantitative extrapolation to geothermal targets. Frontiers in Earth Science 13, 1545052.

Hooker, J.N., Laubach, S.E., and Marrett, R., 2013, Fracture-aperture size–frequency, spatial distribution, and growth processes in strata-bounded and non-strata-bounded fractures, Cambrian Mesón Group, NW Argentina. Journal of Structural Geology, 54, 54-71. doi.org/10.1016/j.jsg.2013.06.011

Peacock, D.C.P., Sanderson, D.J., Leiss, B. 2022. Use of analogue exposures of fractured rock for Enhanced Geothermal Systems. Geosciences 12(9), 318.

Sweeney, M.R., Hyman, J.D., O'Malley, D., Santos, J.E., Carey, J.W., Stauffer, P.H., & Viswanathan, H.S., 2023. Characterizing the impacts of multi-scale heterogeneity on solute transport in fracture networks. Geophysical Research Letters 50(21), e2023GL104958.

---

## Author Comment (AC1)

We thank prof. Stephen Laubach for his thoughtful and constructive feedback. His comments and the suggested references have been of great help in improving the quality of the paper. To address the general comments of prof. Laubach we modified the introduction and conclusions paragraphs, and we propose a new title, to better highlight the scope of the work. In the following, we will address all the comments and explain how we intend to modify the manuscript following the reviewer's suggestions. Reviewer's comments are reported in black and the replies in blue, line numbers refer to the submitted version of the manuscript.

**Title.** To better describe the scope of the paper, first of all we would like to propose a new title

Original title: **Quantitative parametrization of fracture networks in Digital Outcrop Models: an optimized workflow**

Revised title: **An Integrated Workflow for Parametrization of Fracture Network Geometry in Digital Outcrop Models**

29 The Introduction. Think about creating an opening line that points the reader to the focus of your paper and organizing the elements of the Introduction from general to specific. Line 36 is the most general and ought to be first: 'Fractures exert a fundamental control on mechanical and hydraulic properties of rock' and 'knowledge of fracture attributes has application to many societally important engineering operations' (cite a few references, or put all the references from lines 37-45 in a table. This long text list of applications goes off topic and some of it is repetitive.) Then cover what is important about fractures: 'The effects of fractures on strength and fluid flow depend on several factors, including mode (fault versus opening-mode), mineral fill, orientation, size, and spatial arrangement (some references).' Then focus in on your topic: how fractures are arranged in space. You might want to consider how topic is covered in the 2018 J. Struct. Geol. review of spatial arrangements of faults and opening-mode fractures: 'The arrangement of fractures in space and in relation to one another into networks...' note that size and spatial arrangement are challenging or impossible to document using well data and then go to something specific about what your paper provides. Much of lines 68 to 109 seem distracting for an Introduction. These thoughts may be more effectively explored in the Discussion

**Done**. We thank the reviewer for the helpful comment. As suggested, we have switched the first and second paragraphs to improve the logical flow. Additionally, we have revised and

condensed the paragraph discussing the applications of fracture networks, in order to reduce repetition and enhance clarity:

Original (lines 30-47)

Fracture networks are complex geological objects composed by all the fractures in a rock mass, were "fracture" is used here as a collective term for all the different types of discontinuities that affect rocks, including both primary features (e.g. bedding and foliation) and secondary discontinuities such as faults, shear fractures, joints, veins, stylolites and other dissolution features, deformation and compaction bands, dikes, etc. (Schultz, 2019). Fractures can be classified in sets, i.e. populations of cogenetic discontinuities related to the same deformation phases, kinematics (e.g. joint, normal fault), filling (e.g. quartz vein) and orientation, within statistical variability (Twiss and Moores, 2006; Davis et al., 2012).

Fractures exert a fundamental control on the mechanical and hydraulic properties of rock masses, and their relevance extends to multiple applications, including reservoirs of every kind of geofluid (e.g. Immenhauser et al., 2004; Pringle et al., 2006; Hodgetts, 2013; Wang et al., 2023), nuclear waste repositories (e.g. Follin et al., 2014; Hadgu et al., 2017), geothermal energy (e.g. Kosović et al., 2024), geo-hazard (e.g. Eberhardt et al., 2004; Agliardi et al., 2013; Riva et al., 2018), engineering geology (e.g. Franzosi et al., 2023a, b), seismic swarms migration (e.g. Cox, 2016), hydrothermal mineralization (e.g. Micklethwaite, 2009; Townend et al., 2017), and induced seismicity due to underground fluid injection (e.g. Karvounis and Wiemer, 2022). In recent years, the interest in fractured reservoirs has increased due to the growing number of projects related to decarbonization and the energy transition, such as CUS (Carbon Underground Sequestration; e.g. March et al., 2017, 2018), underground hydrogen storage (e.g. Wallace et al., 2021; Zamehrian and Sedaee, 2022), fractured aquifers and medium/high enthalpy geothermal fields (e.g. Genter et al., 2010), and underground energy storage (e.g. Menéndez et al., 2019). In all these applications, fracture patterns hold great importance as they influence the direction, magnitude, and heterogeneity of fluid flow, and the storage volume of reservoirs.

Revised:

Fractures exert a fundamental control on the mechanical and hydraulic properties of rock masses, and their relevance extends to multiple applications, including reservoirs of every kind of geofluid (March et al., 2017; Wallace et al., 2021; Wang et al., 2022; Forstner et al., 2025), nuclear waste repositories (Follin et al., 2014; Hadgu et al., 2017), geology engineering (Eberhardt et al., 2004; Agliardi et al., 2017; Franzosi et al., 2023) and contaminant transport (Cherubini, 2008; Medici et al., 2024). In all these applications,

fracture patterns hold great importance as they influence the direction, magnitude, and heterogeneity of fluid flow, and the storage volume of reservoirs.

Fracture networks are complex geological objects composed by all the fractures in a rock mass, were "fracture" is used here as a collective term for all the different types of discontinuities that affect rocks, including both primary features (e.g. bedding and foliation) and secondary discontinuities such as faults, shear fractures, joints, veins, stylolites and other dissolution features, deformation and compaction bands, dikes, etc. (Schultz, 2019). Fractures can be classified in sets, i.e. populations of cogenetic discontinuities related to the same deformation phases, kinematics (e.g. joint, normal fault), filling (e.g. quartz vein) and orientation, within statistical variability (Twiss and Moores, 2006; Davis et al., 2012).

30 This definition of 'fracture' seems overly broad. Useful definitions to some extent depend on the application and how results are going to be used. The Schultz 2019 definitions might be useful in rock mechanics/excavation/engineering setting, but seem to me to be the wrong place to start if your objective is using the outcrop as an analog for the subsurface in the applications that you list next. And there are other uses for outcrop fracture characterization that might use different definitions. For example, for geomorphologic work—which uses many of the same analytic tools of topology, spatial arrangement, and aperture measurement—alternate categories are useful (see for example, the workflow paper by Eppes et al.).

So I suggest you adopt a more structural geologic definition (faults and opening-mode fractures) and mention that for other applications workers may need other terms or definitions.

Not all bedding constitutes discontinuities. And why do you say foliations are a primary feature and fractures secondary? Foliations are certainly in the same 'secondary' category as foliations. Later in the text you do not consider bedding to be 'fractures' so you don't seem to be following your own definition.

**Done**. We thank the reviewer for this comment. We initially thought that a broader definition for the term "fracture" would have been more useful, as the techniques presented are not necessarily related to a fluid flow field of application. It is also true that this workflow was conceptualized to define the input parameters for stochastic DFN models, which are ultimately used to run flow simulations. We agree to remove the distinction between primary and secondary features, that was ill posed. Here is a revised version of the paragraph, trying to account for both the scenarios (fluid flow and other applications):

Original (lines 30-35):

Fracture networks are complex geological objects composed by all the fractures in a rock mass, where "fracture" is used here as a collective term for all the different types of discontinuities that affect rocks, including both primary features (e.g. bedding and foliation) and secondary discontinuities such as faults, shear fractures, joints, veins, stylolites and other dissolution features, deformation and compaction bands, dikes, etc. (Schultz, 2019). Fractures can be classified in sets, i.e. populations of cogenetic discontinuities related to the same deformation phases, kinematics (e.g. joint, normal fault), filling (e.g. quartz vein) and orientation, within statistical variability (Twiss and Moores, 2006; Davis et al., 2012).

Revised:

Fracture networks are complex geological objects composed of all the fractures in a rock mass.

Here, the term "fracture" will be used as a general term including both opening-mode or shear fractures (joints, faults, etc.), filled or not (veins, joints, etc.). Broadening the meaning of "fracture" by including other kind of discontinuities, such as deformation/compaction bands, foliations, bedding, pressure solution seams and stylolites, etc., may be useful in some research field or application, such as engineering rock mechanics, geomorphology or hydrogeology (Schultz, 2019; Eppes et al., 2024).

30 'where'?

**Done**. Thank you, we fixed the typo.

35 for the 'filling' aspect these references seem inadequate and moreover, it has already been shown that using mineral deposits alone to help define sets is quite misleading for several reasons. I suggest you call out the Reviews of Geophysics 2019 paper here, which is already in your reference list (line 1185). As for the other citations, I think citing textbooks is not favored. And you already have the Hancock 1985 review in your reference list with its classic summary of fracture set rules.

**Done**. We fixed the references as the reviewer suggested.

Original (lines 33-35):

Fractures can be classified in sets, i.e. populations of cogenetic discontinuities related to the same deformation phases, kinematics (e.g. joint, normal fault), filling (e.g. quartz vein) and orientation, within statistical variability (Twiss and Moores, 2006; Davis et al., 2012).

Revised:

Fractures can be classified in sets, i.e. populations of cogenetic discontinuities related to the same deformation phases, kinematics (e.g. joint, normal fault), filling (e.g. quartz vein) and orientation, within statistical variability (Hancock, 1985; Laubach et al., 2019).

46 And there are some papers that are examples of extracting fracture information from outcrop specifically for these applications. How does your work differ from or advance from these other studies?

**Done**. We added two references to complete the paragraph.

Original (lines 46-47):

In all these applications, fracture patterns hold great importance as they influence the direction, magnitude, and heterogeneity of fluid flow, and the storage volume of reservoirs.

Revised:

In all these applications, fracture patterns hold great importance as they influence the direction, magnitude, and heterogeneity of fluid flow, and the storage volume of reservoirs (Davy et al., 2013; Wang et al., 2022).

47 'and rock strength.'

**Done**. We corrected the typo.

31, 35, 55, 130 A complete description of a fracture network and a method for extrapolation from outcrop to subsurface ought to have a step in it where the diagenetic state of the outcrop host rock and fractures is documented. What is the diagenetic state of the host rock and what kinds, if any, mineral deposits are in the fractures? This is not a hard step to add. There are a number of examples in the literature that describe how to do it. It's just a matter of describing (or even reporting) host rock and fracture properties; and if for some reason this cannot be done (not even one thin section?) then at least the 'complete and rigorous workflow' could mark this as a gap. The paper need not be made much longer by mentioning the need for such a step. The MS already cites one paper that makes this point (Forstner and Laubach 2022) so this is not a matter of adding a citation.

And the case has been made in the literature, and I think it is hard to dispute at this point, that key parameters such as connectivity and length are modified in essential ways by cement deposits. This information, if possible, should be included in the basic network description and the topological formulations. For example, there is a big difference between a network of two orthogonal joints sets, where all fractures are open, and a geometrically identical arrangement where the first set is filled (veins, for example) and the second is open, and a case where all the fractures of both sets are sealed. All of these

cases have been found in outcrop. It's not helpful to the modeling or analog user community to report that all with connectivity indices based solely on the trace patterns.

As a brittle structural community and creators and users of fracture outcrop analogs we can't be satisfied with methods that ignore mineral deposits when core data shows that such deposits are a fundamental attribute of most of subsurface fractures that are of interest to practical applications. Moreover, the mineral deposits in fractures are one of the few features that can reliably be measured in both outcrop and subsurface. Such observations can be a useful part of relating outcrops to specific subsurface targets (Ukar et al., 2019; Elliott et al., 2025). Host rock composition and diagenesis is also something that you need to know in comparing mechanical and fracture stratigraphy from analog to target.

**Done**. Regarding the diagenetic state of the host rock and description (if any) of mineral deposits, we cited previous works in the same quarry focused on microstructural and petrophysical analysis. In this outcrop, veins are almost absent, as well as fibers on small faults. In any case, we feel that this simplification allows concentrating on meso-scale geometry and topology, which are the main purpose of the study. To improve clarity, we highlighted the lack of mineral fillings in the fractures in section 2 -**Selecting an outcrop: the Altamura Limestone at Pontrelli quarry** .

Revised (from line 177):

...Aside from the geometrical characteristics, veins are absent in all of the three fracture sets, as well as fibers on small faults (Set 1 and Set 2).

50 Table and 55-60 Text

This list of fracture properties needed is incomplete.

The table footnotes should do a better job of explaining what you mean by 'static/dynamic'. Overall the table does not seem very clear, and some aspects are questionable. For example, by starring 'network' for 'topology' but not 'sets' you imply that topology can (or should) only be documented for 'networks' but this way of thinking of the issue is limiting. For example, if there is only one set, you could still define the topology (it would be the topology defined by fractures in that set). Core data suggest that 'one set' is common in several basins (see papers by Laubach and by J. Lorenz from the 1990s). If you leave this circumstance out, you may be missing just what the analog is supposed to capture. And in many cases, the network in outcrop in not what you want to describe the subsurface. Instead, it would be better to isolate part of the outcrop network for topology analysis. The literature has plenty of examples of outcrop fracture studies where the first step is figuring out what weathering and other 'near surface' fractures ought to be omitted (or at least

accounted for separately in topology analysis). There is a clear example by Lorenz. The studies of fractures in central Texas related to the SSC cite are another. In both of these examples, extracting the meaningful fractures to analyze has major practical implications for the usefulness of the analog study.

On the static versus dynamic, here you must mean 'on an engineering time scale', inasmuch as over geologic time scales all of these features are 'dynamic' as fractures grow and interact. But even on the engineering time scale how can you be sure that, for example, connectivity and length are static? And there is good evidence that in some reservoirs apertures are not particularly dynamic. So, while the static versus dynamic aspect is useful to think about, it seems like a red herring here where you point is that many of these extended attributes are difficult or impossible to measure in the subsurface with the kinds of probes we have. Do you need the static versus dynamic distinction later? Maybe better to omit and make the table about attributes you desire to measure.

**Done**. Here we propose to modify Table 1 to better highlight all the parameters needed to fully characterize a fracture network, what parameters can be obtained from DOMs (facets and/or traces) and what parameters we are focusing on. We agree with the reviewer that the Static/Dynamic column can be misleading and it's not functional for the aims of the study, and we decided to remove it. We removed Static/Dynamic from even from the text.

Original (line 50):

| Parameter | Fracture network | Fracture set | Static/Dynamic |
|---|---|---|---|
| Number of sets | * | | Static |
| Orientation | | * | Static |
| Topology | * | | Static |
| Size (length/height) | | * | Static |
| H/L ratio | | * | Static |
| Density/Intensity (1) | * | * | Static |
| Aperture | | * | Dynamic |
| Spatial organization | * | * | Static |
| Representative Elementary Volume (2) | * | * | Static |

Revised:

| Parameter | Fracture network | Fracture set | DOM - Facets | DOM - Traces |
|---|---|---|---|---|
| Number of sets | * | | * | * |
| Orientation | | * | * | |
| Topology | * | * | | |
| Size (length, height) | | * | | * |
| H/L ratio | | * | | * |
| Density, Intensity (1) | * | * | | * |
| Aperture | | * | | |
| Spatial organization | * | * | | * |
| Representative Elementary Volume, Area (2) | * | * | | * |

| | | |
|---|---|---|
| Roughness | * | * |
| Kinematics | * | |
| Deformation Mechanism | * | |
| Filling | * | |

Original (line 55-61):

The quantitative characterization of fracture networks requires the determination of several geometrical and topological attributes of fractures and their statistical distributions (Table 1). Some of these attributes apply to the individual fracture set (e.g., orientation, length/height distribution) others to the whole fracture network (e.g., topology). Fracture properties can be static, meaning that they do not change in response to boundary stress field and fluid pressure variations (e.g., number and orientation of fracture sets), or dynamic, meaning that they can change due to variations of mechanical conditions, as for instance does fracture aperture when fluid pressure changes in response to injection of fluids in a reservoir or in the seismic cycle (Gleeson and Ingebritsen, 2012).

Revised:

The quantitative characterization of fracture networks requires the determination of several geometrical and topological attributes of fractures and their statistical distributions (Table 1). Some of these attributes apply to the individual fracture set (e.g., orientation, length/height distribution) others to the whole fracture network (e.g., topology).

64-66 This account of what can and cannot be measured could be more nuanced and do a better job of setting up what your paper contributes. Some aspects of fractures can be measured on the meso scale, like strike and dip, aperture, some aspects of abundance, and if the wells are deviated 1D spatial arrangement.  That's not the same as 'cannot be measured'. The elements that can't be measured are length, height, and connectivity.

**Done**. We changed the text according to the reviewer's suggestions.

Original (lines 64-67):

Fractures in the subsurface (e.g. in reservoirs) cannot be characterized at the mesoscale (meters to tens of meters) using direct techniques. Borehole data (cores and image logs) only provide 1D, very local and sparse information (limited to1D traces in a 3D volume), do not constrain the size of discontinuities, and are affected by important orientation biases (Baecher, 1983).

Revised:

Fractures in the subsurface (e.g. in reservoirs) can only be partially characterized at the mesoscale (meters to tens of meters) using direct techniques. Boreholes provide local information (limited to1D traces in a 3D volume) about the orientation distribution, aperture, fracture abundance (P10, Dershowitz and Herda, 1992) and, if the borehole is properly oriented with respect to the average orientation of a fracture set, 1D spatial arrangement. In contrast, length and height distributions, connectivity and the REA cannot be measured.

68 'fractures are not always...'
86 'fracture state'

**Done**. We fixed the typos

90-109 This commentary on DFNs seems out of place here. Maybe it belongs in the Discussion. DFN's are something you build once you have information about the fractures (however incomplete) and so topically it seems out of place in a lead up to ways to improve characterization.

**Done**. We thank the reviewer for the comment. The idea behind the addition of this part of the introduction was to highlight the fact that parameters calculation in this contribution is tuned for the specific purpose of stochastic modelling. We agree that this part of the introduction may be a bit long and too specific. We propose a summarized version of the paragraphs:

Original (lines 91-109):

The impossibility to directly map or image fractures in the subsurface suggested using continuum representations based on some form of upscaling or homogenization, such as the dual porosity model (Warren and Root, 1963). Alternatively, the Discrete Fracture Network (DFN) approach allows generating stochastic simulations where fractures are simplified as planar polygons in 3D or segments in 2D. Generating DFNs requires fracture parameters and a stopping criterion to end the simulation. Standard DFN software (e.g. Move – https://www.petex.com/pe-geology/move-suite/, Petrel – 95 https://www.slb.com/products-and-services/delivering-digital-at-scale/software/petrel-subsurface-software/petrel, FracMan – https://www.wsp.com/en-gl/services/fracman, DFNworks – https://dfnworks.lanl.gov/) are based on a Poisson point process that generates fractures with a random spatial distribution. The geometrical properties of each fracture are drawn from parametric length and orientation distributions, and fracture height is generally controlled by a fixed height/length ratio. The simulator generates fractures until a target fracture intensity $P32$ (Dershowitz and Herda, 1992) is reached in the simulation 100 volume. Connectivity or any other form of spatial organization cannot be taken into

account in these models due to limitations of the Poisson distribution, that is specifically based on the assumption of spatial independence between fractures (e.g. Davis, 2002). Modern approaches have been developed in the last years to try and solve this fundamental limitation, for instance controlling clustering of fractures by means of the Ripley's K function (Shakiba et al., 2024), or including attractive vs. repulsive spatial and directional processes controlled by statistical and/or pseudo-mechanical parametrizations (Bonneau 105 et al., 2013; Davy et al., 2013; Bonneau et al., 2016), but a satisfactory solution has yet to be found, especially in 3D. Sometimes also "deterministic" DFNs are used, but the possibility of creating such models is limited to structures that can be imaged in 3D seismics, i.e. meso-scale faults larger than some hundred meters and characterized by an offset that results in a contrast in seismic impedance.

Revised:

The impossibility to directly map or image fractures in the subsurface lead to using continuum representations based on some form of upscaling or homogenization, such as the dual porosity model (Warren and Root, 1963). Alternatively, the Discrete Fracture Network (DFN) approach allows generating stochastic simulations where fractures are simplified as planar polygons in 3D or segments in 2D. In the standard and most widespread approach to stochastic 3D DFNs, the geometrical properties of each fracture are drawn from parametric length and orientation distributions, and fracture height is generally controlled by a fixed height/length ratio. The simulator generates fractures until a target fracture intensity $P32$ (Dershowitz and Herda, 1992) is reached in the simulation volume (e.g. Move – https://www.petex.com/pe-geology/move-suite/, Petrel – https://www.slb.com/products-and-services/delivering-digital-at-scale/software/petrel-subsurface-software/petrel, FracMan – https://www.wsp.com/en-gl/services/fracman, DFNworks – https://dfnworks.lanl.gov/). Fractures are randomly distributed in the simulation volume according to a Poisson point process, therefore connectivity or any other form of spatial organization cannot be reproduced in these models. More sophisticated approaches have been developed in the last years to try and solve this fundamental limitation (Shakiba et al., 2024, Bonneau et al., 2013; Davy et al., 2013; Bonneau et al., 2016), but a satisfactory solution has yet to be found, especially in 3D.

99 Is a fixed height/length ratio realistic? Doubtful.

We agree that a fixed H/L ratio is not realistic, but it's a requirement in most of the stochastic 3D DFN models. This is the reason why the following discussion has been developed.

111 Here and elsewhere where you mention 'in detail'; note that this is a vague usage. Omit or mention a scale range.

**Done**. We thank the reviewer for this correction. We followed your suggestion and removed "in detail" where present.

114 For comparing outcrop and subsurface, I don't think you want to say they 'underwent the same geologic history' since outcrops and subsurface targets by definition have different geologic histories, and the differences could have a material effect on what fractures are there. Uplift, contraction, weathering and a bunch of other processes must differ from the target to the outcrop. See English, 2012, Engelder, 1985, and Eppes et al. 2019 for discussions of various aspects of these differences. Peacock 2016 and Elliott et al. 2015 have text that describes how the inevitable differences can be accounted for. The reality is that outcrops and targets will always vary in important ways and one of the steps in a 'rigorous workflow' needs to be collecting data that will allow the nature of these differences to be identified. That way, the applicability of the analog can be judged, and, in some cases, the outcrop patterns can be adjusted to match the subsurface situation (as in the example in Forstner and Laubach, 2022, a study you cite).

**Done**. We thank the reviewer for this suggestion; we changed the text as follows:

Original (lines 113-114):

that underwent the same geological history (Bertrand et al., 2015; Bistacchi et al., 2015; Jacquemyn et al., 2015; Martinelli et al., 2020).

Revised:

that underwent a geological and tectonic evolution that is at least partly comparable. The applicability of an outcrop as an analogue should be evaluated carefully, and some assumptions should be eventually made (Forstner and Laubach, 2022)

116-118 This is vague. I'm not sure what you mean. What are 'traditional, direct...' surveys? Outcrop fracture trace maps via surveying instruments and film have been acquired since at least the 1980s and although those methods are certainly slower than DOMs they may not be less accurate. Are you trying to say that previous studies of fracture statistics from outcrop that don't use your method are unreliable? If this is your point, then the comments belong in the Discussion after you have demonstrated this.

116 I don't know what you mean by the phrase 'unavoidable as it is limiting' and how does the 'traditional direct geological and structural field survey' differ materially from flying the outcrop with a drone? In any case, relative chronology' if you mean crossing and abutting relations can be estimated from images and 'mineralization/filling' is best done in the

laboratory with a thin section. And if by 'geometrical datasets with traditional techniques' you mean fracture trace maps, this was accomplished in the past for large outcrops without drones or digital outcrop models (see Barton from the late 1980s) although no doubt current technology makes collecting such information easier. So these sentences need adjustment so as not to be misleading.

Either way, access to advanced image collection methods do not solve two big limitations the use of outcrop fracture tracer mapping: the finite size of most outcrops and the potential for fractures in outcrops to be unrelated to the subsurface. These caveats ought to be mentioned.

Provide a definition of what you mean by 'traditional direct geological and structural field survey'.

132 Define the 'traditional direct field…'

**Done**. These three comments are addressed together given that they are about the same topic. We agree with the reviewer that "traditional" might deliver a misleading significance, and we decided to remove the term. At the same time, we want to clarify that in the paper it is nowhere stated that other methods are considered unreliable. We agree that fracture network properties such as mineralization and filling are best characterized in laboratory. Even if we agree that trace mapping and data collection in general can be accomplished without the support of digital outcrop models, we think that this statement is limited to horizontal outcrops. Vertical walls or gorges that are more than 50 m in height, common occurrence in the Alps for example, cannot be characterized without a digital model, given safety and accessibility restrictions. Certainly, DOMs don't inherently solve certain problems like the cited finite size of the outcrop or potential fractures unrelated to subsurface, but these limitations appliy to any data acquisition method. DOMs are a mean for data collection, which does not imply any kind of interpretation. In this paper we present a statistical approach, based on data collected from DOMs, to address the finite size of an outcrop, at least for the length/height distribution calculation, while detecting fractures not related to the subsurface is out of the scope of this contribution.

Original (lines 115-119):

In this regard, the traditional direct geological and structural field survey is as unavoidable as it is limiting, in the sense that parameters such as kinematics, roughness, relative chronology and mineralization/filling of structures can only be gathered during fieldwork (e.g. Hancock, 1985), but on the other hand limited accessibility and logistical limitations prevent the collection of extensive geometrical datasets with traditional techniques (e.g. McCaffrey et al., 2005)

Revised:

In this regard, field survey, intended as physically inspecting and collecting data from outcrops, is a fundamental step in the process of fracture network characterization, because features such as kinematics, roughness, relative chronology and mineralization/filling can only be gathered during fieldwork. At the same time, even if it is possible, manually collecting massive amounts of data is time consuming on horizontal outcrops, and very difficult in vertical outcrops, where the accessibility is limited (data can only be collected in the portion of the outcrop reachable by the geologist) and depending on the conditions, safety is not guaranteed (e.g. rocks falling from the top of the cliff).

130 The actual scope of the paper needs to be clarified here. And mention what parameters you are leaving out.

**Done**. Here is a revised version of the last two paragraphs:

Original (lines 130-144):

In this paper we present a workflow that combines new and existing methodologies to quantitatively characterize all the parameters of a fracture network, with a particular focus on obtaining robust statistical distributions to be used as input in stochastic DFN models. The analysis is based on both traditional direct field observations and photogrammetric DOM analysis. The integrated workflow is aimed at maximizing the structural information that can be obtained from different types of DOMs, including orientation distributions, topological relationships, length and height distributions and fracture intensity. Some of these parameters (i.e. topology) are not direct inputs to DFN models, yet they represent a fundamental  control on the quality of the generated model itself.

The workflow, rooted in a rigorous statistical background, attempts at minimizing the assumptions made at every step, for example during the choice of the orientation distribution, or the length and height distribution. The methodologies proposed to estimate each parameter can be applied independently, subject to the type and quality of the outcrop. The complete workflow (Figure 1), combining both facet and trace data, includes two separate processing pipelines: (i) semi-automated fracture orientation analysis carried out on point cloud DOMs (Sect.4.4); and (ii) fracture trace analysis carried out on orthomosaics, allowing to measure topological relationships, length and/or height distributions, $P21$, and to estimate (subject to assumptions) the H/L ratio distribution (Sect. 5 to 8). The two pipelines are integrated to achieve a complete 3D parametrization of the fracture network (Sect. 9 and following).

Revised:

The scope of this paper is to present a workflow based on statistically rigorous methodologies to characterize a fracture network from the geometrical point of view. The result of such workflow provides a suite of parametrical distributions to be used as input in current stochastic 3D DFN models. The parameters considered here are: The orientation distribution, the length/height distributions, the topological parameters, the fracture areal intensity (P21) and the H/L ratio. We aim at integrating 2D and 3D data sources (point clouds, orthomosaics, DEMs), vertical and horizontal outcrops and facets and traces data to achieve a 3D geometrical parametrization of the fracture network (Sect. 9 and following). The methodologies proposed to estimate each parameter can be applied independently, subject to the type and quality of the outcrop.

The first part of the paper is dedicated to best practices about data acquisition (both ground-based and UAV-based), pre-processing, reconstruction and quality assessment of a photogrammetric model (Sect. 3). Then two separate processing pipelines are presented, depending on the DOM type: (i) semi-automated fracture orientation analysis carried out on point cloud DOMs (Sect. 4.4); and (ii) fracture trace analysis carried out on orthomosaics, allowing to measure topological relationships, length and/or height distributions, $P21$, and to estimate (subject to assumptions) the H/L ratio distribution (Sect. 5 to 8).

135 Is it really correct that factors such as connectivity are not incorporated in DFNs? Some DFNs can incorporated aperture size variation. See papers by Sweeney et al, for example, 2023.

Yes, topology is never taken into account as an explicit parameter in DFN models, for example including a P20 specific for X nodes or Y nodes. The starting point is the standard stochastic 3D DFN model, based on Poisson point process, where the connectivity is achieved randomly and only by means of X nodes (Y nodes are high unlikely). Advancements have been made by different approaches (Davy et al., 2013; Bonneau et al., 2016), where a higher degree of connectivity is achieved by integrating geo mechanical principles in the stochastic generation of fractures. Recently, FRACMAN software integrated a new approach in which fracture set are generated in different steps depending on the relative chronology, and the connectivity is managed by a "termination chance" parameter. This is the reason why we wrote that connectivity represents a fundamental quality check.

137 Awkward wording: 'minimizes assumptions at each step'.

**Done**. Thank you for the correction. This phrase was deleted in the revised version of the introduction.

138-146 This would be more compelling if you could state your specifics claims: 'here we show that...'

This comment falls in the revised version of the paper proposed in a comment above (line 130)

163, 425, 638 Describing and classifying height patterns should be a step in outcrop network description. There is a useful height classification in Hooker et al. (2013. J. Struct. Geol.)

This comment, and the others related to height patterns will be answered at line 605

175 What do you mean by 'distinctly younger'? This makes it seem like you can detect a gradation in age, but all you have is an abutting relation but no information about how much younger. Geomechanical modeling shows that such relations can arise in a single deformation sequence or can reflect much longer times. Check the text and remove such unjustified qualifiers.

**Done**. We agree to remove "distinctly", the time gap could be very small or very large actually, but abutting relationships are systematic and point to two different events, with the exception of ca. 4% of D1 fractures that might have been reactivated in a later stage.

Original (lines 174-175):

However, structures belonging to Set 2 always abut on those belonging to Set 1, showing that Set 2 is distinctly younger (Table 2).

Revised:

However, structures belonging to Set 2 always abut on those belonging to Set 1, showing that Set 2 is younger (Table 2).

181 Forstner et al. 2025 GSL Energy Geoscience Conference Series, v. 1 explicitly investigates the effects of such patches on length distribution statistics. A good workflow needs to establish and describe how continuity across these features is treated.

This comment will be addressed at line 680

183 'drowned by artificial fractures' is causal and vague. Can you restate this?

**Done**.

Original (lines 181-183):

Other zones distributed across the pavement are partially affected by non-natural, quarrying-related fractures, but with a careful analysis it is still possible to detect Set 1,

while Set 2 and especially Set 3, being characterized by smaller fractures, are drowned by artificial fractures.

Revised:

Other zones distributed across the pavement are partially affected by non-natural, quarrying-related fractures, but with a careful analysis it is still possible to detect Set 1, while Set 2 and especially Set 3, being composed of smaller fractures, are more difficult to interpret and separate from the ones related to quarrying.

249 I think the text here could be confusing. You mean the best parameter to extract information from your image, but the way you put it ('most important parameter for applications in structural geology') it sounds like a geologic or fracture parameter and the reader asks The most important parameter for what? Most important probably depends. Is this qualifier even needed here? The remark seems more appropriate for the Discussion, where it can be defended.

This sentence for definition of 'surface density' could be clearer. I take it that this is an imaging parameter but it could be read as a fracture abundance measure. If what you mean by this is 'where the most fractures are' at least in terms of fluid flow there are examples in the literature where production data shows that numerous fractures does not correlated with, for example, high fluid flow (Wang et al. 2023, Marine & Petroleum Geology).

**Done**. We thank the reviewer for the comment. The surface density parameter in this paragraph doesn't refer to the areal fracture density as defined by Dershowitz and Herda, 1992, but it is a parameter specific to point clouds and refer to the number of points inside a sphere of arbitrary radius. The higher the point surface density, the easier it will be to detect geological features on the cloud surface. Following the suggestions of the reviewer here is a new version of the paragraph:

Original (line 249):

We believe that the most important parameter for applications in structural geology is surface density.

Revised:

We believe that the most important parameter to evaluate the quality of a photogrammetric model for applications in structural geology is the point cloud surface density (SD).

270 Add graphic bar scales to the outcrop images.

**Done**, we modified the figure as suggested.

[Figure]

503 'trace connectivity' and their possible effects on 'fluid flow' are two different things and ought to be more carefully separated in your description. Fully connected traces may not imply any enhancement in flow (for example, if the traces are faults or sealed fractures); disconnected open fractures can enhance flow if the host rock is permeable. Please be more careful in how these parameters are portrayed.

**Done**. We thank the reviewer for this correction. We acknowledge that connectivity and fluid flow are two different things, and we decided to remove "fluid flow" from the sentence.

Original (lines 501-503):

Even under this limitation, topology is a fundamental component of fracture network analysis because it is directly related to connectivity and fluid flow, as demonstrated by Sanderson and Nixon (2015).

Revised:

Even under this limitation, topology is a fundamental component of fracture network analysis because it is directly related to connectivity, as demonstrated by Sanderson and Nixon (2015).

510 Include contingent nodes in nodes list. These kinds of nodes go back to at least Barton and Hsieh.

**Done**. We added C nodes to the list as the reviewer suggested. We strongly believe that the detailed analysis presented in the suggested paper is a key step in implementing representative fluid flow models. Using C nodes implies that the geologist must take a decision about their nature, since "activated" C nodes become V nodes, and "not activated" C nodes become I nodes. This decision heavily impacts all the statistical parameters calculated downstream (length and height parameters, H/L ratio). We therefore believe that making a decision about the nature of C nodes should be undertaken before standard topological analysis, and it is beyond the scope of this contribution. At the same time, the non-uniqueness of the definition criterion makes the implementation in a standardized library complex.

Original (lines 508-516):

According to Benedetti et al. (2025), four main nodes categories can be found in a fracture network (Figure 8):

- I nodes: fracture trace true tip points;
- Y nodes: abutting relationship;
- X nodes: crosscutting relationship;
- V nodes: perfect coincidence of two tip points belonging to two different fractures - these are theoretically possible, but extremely unlikely;
- B nodes: boundary nodes, where a fracture trace terminates at the interpretation boundary.

Revised:

Six main nodes categories can be found in a fracture network (Benedetti et al., 2024; Forstner and Laubach, 2022; Nyberg et al., 2018, Figure 8):

- I nodes: fracture trace true tip points;
- Y nodes: abutting relationship;
- X nodes: crosscutting relationship;
- V nodes: perfect coincidence of two tip points belonging to two different fractures - these are theoretically possible, but extremely unlikely;
- B nodes: boundary nodes, where a fracture trace terminates at the interpretation boundary.
- C nodes: Contingent nodes that can be enabled or not, generating different fracture network configurations, depending on configuration rules defined according to the study objectives and sometimes micro-scale observations (Forstner & Laubach, 2022).

515. I think you mean 'hard to recognize'. Since many fracture arrays grow by linkage such connections my be common; and in fact, most opening mode fracture traces have evidence at a range of scale that they are made up of end-to-end links. See papers by Olson and Pollard and Lamarche et al. If you increase the image resolution, single traces bounded by I-nodes may reveal numerous low angle Y nodes (see Forstner et al. 2025, their figure 6c.

**Done**. We fixed the phrase in accordance with the reviewer's suggestions.

Original (lines 514-515):

V nodes: perfect coincidence of two tip points belonging to two different fractures - these are theoretically possible, but extremely unlikely;

Revised:

V nodes: perfect coincidence of two tip points belonging to two different fractures - these are theoretically possible, but hard to recognize at the interpretation scale.

516 'C-nodes', which you mention later, ought to be listed here. They certainly reflect the same level of abstraction as these other node types.

**Done**.

Original (lines 517-519):

The nature of I, Y, X and V nodes is related to the processes that generate the fractures in the first place, but an additional consideration pertains to B nodes (Nyberg et al., 2018, Benedetti et al., 2025), which result from the interaction between the fracture network and external processes.

Revised:

The nature of I, Y, X, V and C nodes is related to the processes that generate the fractures in the first place, but an additional consideration pertains to B nodes (Nyberg et al., 2018, Benedetti et al., 2025), which result from the interaction between the fracture network and external processes.

Original (lines 522-525):

Nodes classification is based on their topological value (Sanderson et al., 2019), representing the number of branches connected to each node. Specifically, I nodes have a topological value of 1, V nodes have a value of 2, Y nodes have a value of 3, and X nodes have a value of 4. B nodes can be categorized as nodes with a topological value of 3, but one branch originates from the fracture trace while the others come from the interpretation boundary.

Revised:

Nodes classification is based on their topological value (Sanderson et al., 2019), representing the number of branches connected to each node. Specifically, I nodes have a topological value of 1, V nodes have a value of 2, Y nodes have a value of 3, and X nodes have a value of 4. B nodes can be categorized as nodes with a topological value of 3, but one branch originates from the fracture trace while the others come from the interpretation boundary. C nodes assume a different topological value depending on the chosen configuration. If they are enabled, the topological value will be equal to 2 (V node), if they are not enabled, topological value will be equal to 1, and one C node generates two I nodes. This choice heavily impacts length and height distribution calculation as it is connected to topological analysis. Therefore, the decision about connecting or not fractures through C nodes should be made before running the topological classification.

519 By 'external processes' do you mean 'the size and shape of the outcrop'? Why not just say that? It's less obscure.

**Done**. We fixed the phrase in accordance with the reviewer's suggestions.

Original (lines 517-519):

The nature of I, Y, X and V nodes is related to the processes that generate the fractures in the first place, but an additional consideration pertains to B nodes (Nyberg et al., 2018, Benedetti et al., 2025), which result from the interaction between the fracture network and external processes.

Revised:

The nature of I, Y, X and V nodes is related to the processes that generate the fractures in the first place, but an additional consideration pertains to B nodes (Nyberg et al., 2018, Benedetti et al., 2025), which result from the interaction between the fracture network and the size and shape of the outcrop.

540 This assumes that all the fractures are open. It's one thing to talk about traces and how connected thy may be, but it's quite a jump to assume 'percolation'.

**Done**. We agree with the reviewer, in fact, the cited paper is based on synthetic models and considers all the fractures as open. We will specify this assumption in the text.

Original (540-543):

Backbone extraction solves this problem, highlighting the more numerous connected cluster, and also represents a graphical solution to the percolation threshold problem, since if the backbone touches two opposite sides of the interpretation boundary, this means that a giant connected component exists, and a thoroughgoing flow can be established (Haridy et al., 2020). As shown in Figure 9, the backbone is marked by a significant increase in the CI value.

Revised:

Backbone extraction addresses this problem by highlighting the most extensively connected cluster. Under the assumption that all fractures are open, it also provides a graphical solution to the percolation threshold problem. Specifically, if the backbone spans two opposite sides of the interpretation boundary, it indicates the presence of a giant connected component, allowing for the establishment of a continuous flow (Haridy et al., 2020). As illustrated in Figure 9, the backbone is characterized by a notable increase in the CI value.

594 Omit 'very' as vague. This interpretation of relative ages by abutting and crossing is confusing. Crossing relations and abutting relations amongst have the same implication for relative timing: the abutted fracture is older and the crossed fracture is older.

**Done**. Thanks for the correction, we omitted "very" as suggested. It is true that both crosscutting and abutting relationships have the same implications when it comes to relative timing. At a certain point in time a fracture of a hypothetical "set 2" crosscut a fracture of another hypothetical "set 1", meaning that "set 2 is younger". The same line of thought can be applied to abutting relationships. In absence of distinctive differences between the two fracture sets, the systematically abutting set is interpreted as "younger". At the same time defining relative ages between two mutually crosscutting sets is more difficult.

Original (lines 594-595):

100% of abutting nodes is an asymptotic value, very difficult to reach in a natural context, but nonetheless revealing a tendency in this direction would be interesting.

Revised:

100% of abutting nodes is an asymptotic value, difficult to reach in a natural context, but nonetheless revealing a tendency in this direction would be interesting.

605 Where do these patterns fall on a height classification scheme?

**Done**. We thank the reviewer for providing this reference. The height classification scheme defined by Hooker et al. 2013 provides a visual way to classify the height patterns on a vertical outcrop. The application of FBI index quantifies the percentage of fractures that abut on the bedding but does not provide an interpretation or a binding with aperture measurement as Hooker et al. 2013 does. We believe that the two methods are complementary, and we decided to associate the proposed height classification scheme with the FBI in the results.

Revised (from lines 818 and 835):

818 In relation to classical height pattern classification schemes (Hooker et al., 2013), Set 1 falls between the perfectly bed bounded and the top bounded class, given that even if the major part of the fracture about on the bedding, some fractures (33% of the complete fractures) end between two bedding surfaces.

835 In particular, almost all the Set 2 fractures abut on the bedding surfaces, identifying it a perfectly bed bound fracture set in the height classification scheme (Hooker et al., 2013).

641 'rely'?

**Corrected**

641-642 Hel the reader by succinctly defining what these distinctions mean

**Done**. We modified the text as reported below.

Original (639-641):

Defining an unbiased trace length distribution has always been one of the main challenges in rock mass and fracture network characterization. When calculating or estimating trace length parameters it is possible to distinguish between distribution-dependent and distribution-free methods (Mauldon, 1998).

Revised:

Defining an unbiased trace length distribution has always been one of the main challenges in rock mass and fracture network characterization. When calculating or estimating trace length parameters it is possible to distinguish between distribution-dependent (assume a specific probability distribution) and distribution-free methods (population parameters not linked to any specific probability distribution, Mauldon, 1998).

562-564 This section of text could use some clarification.

**Done**. Here is a revised version of the paragraph:

Original (lines 561-565):

This kind of information can be obtained with what we call "directional topology", where every node not only includes information about its type (I, Y, X and B), but also about its "origin" or "direction", i.e. from which sets a Y node is generated and if it is the first fracture set that abuts on the second or vice versa. The same line of thought can be applied to X and I nodes but to a shallower level, given that in a crosscutting relationship it is not possible to define, just with topological information, which fracture is older or younger, and for I nodes it is only possible to identify the origin set.

Revised:

This kind of information can be obtained using what we call "directional topology." In standard topological analysis, nodes store only the topological value. In contrast, in directional topology nodes also contain information about the fracture set (in the case of I-nodes) or sets (in the case of Y- and X-nodes) from which the connected branches originate. This enables a more detailed topological characterization: I-nodes are classified by set, X-nodes are described by the intersecting sets (e.g., an X-node between Set 1 and Set 2), and for Y-nodes, it is possible to determine whether they are generated by Set 1 abutting on Set 2 or vice versa, by counting the number of branches (Fig. 10).

650-655 The data sets are still constrained by the size of the outcrops. Some early studies captured complete fracture inventories within large and clean outcrops (e.g., Barton, others). So these claims about 'massive' data sets seem like they are missing key elements of the problem.

**Done**. We were not able to find the cited paper and therefore making a proper confrontation between the size of the cited outcrop and our case study. A significant amount of data is fundamental in this kind of analysis, and DOMs are an aid in this regard. Nonetheless we agree that the phrase can be made clearer, in particular we would like to specify that computational power is another key element that the previous author did not

have at their disposal. For example, Baecher, (1980), was only able to correct for censoring the exponential distribution, thanks to its closed form solution.

Original (lines 652-655):

Digital outcrops make it possible to overcome with huge datasets some problems that previous authors could only consider theoretically from a mathematical and stereological point of view. With these new techniques it is possible to acquire massive datasets on very large sampling windows and successfully tackle the different biases that can be present on an outcrop.

Revised:

Digital outcrops and the increasing computational power make it possible to overcome some problems that previous authors could only consider theoretically from a mathematical and stereological point of view. On one hand, these new techniques facilitate the acquisition of massive datasets on very large sampling windows and successfully tackle the different biases that can be present on an outcrop. On the other hand, the increased computational power makes it possible to calculate the solution to mathematical problems that previously could not be solved due to the lack of a closed form solution (Baecher, 1980).

660-663 But this does not solve the conceptual problems of measuring lengths as laid out for example in Ortega and Marrett 2000.

**Done**. We thank the reviewer for this comment. It is true that we didn't consider the problem from this point of view. Adding to the problem highlighted in the cited paper, a potential fracture set parallel to the outcrop mean plane would be undetected. We think that this problem can be partially solved by associating the vertical and horizontal sides of the outcrop. The vertical side acts as a window on the 3D geometry of the network in which it is possible to check the presence/absence of a fracture set parallel to the horizontal side surface and if fractures are under-sampled on the bedding interface. If present, how to address this underestimation is beyond the scope of this paper and should be evaluated on a case-by-case basis. Regarding our case study, we observed that the number of Set 1 fractures on the vertical outcrop roughly matches the number of fractures in the adjacent part of the horizontal outcrop. For Set 2 fractures, as written in the discussion, we can measure them on the vertical side but are drowned by artificial fractures related to quarrying activities on the horizontal side. The doubt remains for Set 3, that is sub parallel to the vertical outcrop and therefore we cannot measure fracture traces. We propose to modify the introduction, adding this bias in the paragraph starting at line 75, and modifying the current paragraph reporting what was mentioned in this reply:

Original (lines 82-83):

82 The size bias states that larger fractures (i.e. fracture surfaces with a larger area) have a greater probability to intersect the outcrop surface and to be sampled.

Revised:

The size bias states that larger fractures (i.e. fracture surfaces with a larger area) have a greater probability to intersect the outcrop surface and to be sampled. Another bias, related to layered media, is the under-sampling of fractures shorter than the bed thickness (Ortega and Marrett, 2000). This bias changes the shape of the length distribution, given that only the fracture high enough to about or crosscut the bedding interface can be systematically sampled.

Original (lines 660-663):

The size bias applies to 1D sampling methodologies (scanlines) where longer fractures have a higher probability of being sampled, but this bias does not apply to areal sampling strategies where everything inside the interpretation boundary is sampled. Even fractures much longer than the interpretation boundary are sampled and classified as censored fractures (see below).

Revised:

The size bias applies to 1D sampling methodologies (scanlines) where longer fractures have a higher probability of being sampled, but this bias does not apply to areal sampling strategies where everything inside the interpretation boundary is sampled. Even fractures much longer than the interpretation boundary are sampled and classified as censored fractures (see below). Areal sampling alone, however, does not account for the possibility of fractures parallel to the outcrop mean plane, and for the under-sampling of fractures shorter than the bed thickness (Ortega and Marrett, 2000). The association between the vertical and horizontal side of the outcrop can partially solve this bias. On the vertical side it is possible to check the presence/absence of a fracture set parallel to the horizontal outcrop surface and the relationship between fractures and the bedding interface. The problem remains for fracture sets parallel to the vertical outcrop mean plane, as the orientation bias hinders the trace mapping. Regarding our case study, we observed that the number of Set 1 fractures on the vertical outcrop roughly matches the number of fractures in the adjacent part of the horizontal outcrop. For Set 2 fractures, we can measure them on the vertical side but they are hidden by artificial fractures related to quarrying activities on the horizontal side. Set 3 is almost parallel to the vertical outcrop configuring the situation in which this bias cannot be evaluated.

666 These size/resolution issues can affect length distributions. This is obvious when you collect fracture information at different scales, and it follows from the segmented character of most fractures and the tendency for fracture length to grow by linkage. See the example in Forstner et al. 2025 where drone, hand held LiDar, and scanline datasets cover the same fracture array.

In any case, saying that you know where your data are truncated is not the same as being able to claim that it can be safely assumed that truncation bias is not affecting the dataset.

**Done**. We thank the reviewer for this specification. It is true that at a fixed scale, fracture smaller than the DOM resolution are lost. It is also true that it is always possible to miss some fractures during the digitalization and among them there may be a fracture smaller than the one we have identified as the limit for truncation bias. We modified the text to account for this suggestion.

Original (lines 664-666):

Working with DOMs, the truncation bias applies to small fractures that can be truncated by limited DOM resolution. In our case, the resolution of the TS-DOM is around 4 mm/pixel and the smallest digitized fracture is 57 cm. This means that there is an order of magnitude between the two and it can be safely assumed that truncation bias is not affecting the dataset.

Revised:

Working with DOMs, the truncation bias applies to small fractures that can be truncated by limited DOM resolution. In our case, the resolution of the TS-DOM is around 4 mm/pixel and the smallest digitized fracture is 57 cm. Although the possibility remains that some fractures were missed during the digitization process — including potentially fractures smaller than the identified truncation threshold — the order of magnitude difference between the resolution of the DOM and the smallest recognized fracture is expected to mitigate truncation bias at a fixed scale.

672 It might be random censoring. But it's useful to wonder whether where you have continuous outcrop and where not is likely to be random, given that in many environments plants (and cover) may be localized in fractures (or in one case I know of, where the wide fractures are). It's a geomorphology and vegetation issue that should not be lost sight of in fracture trace collection; see the comments in Eppes et al.

The assumption of independence implies that the mechanism responsible for fracture generation operates independently from the censoring process; that is, the processes

leading to fracture formation and those causing censoring are distinct and unrelated (Benedetti et al., 2025). For example, tectonic activity that induces fracturing is independent from post-fracture processes such as vegetation growth, which may obscure or censor the observable record.

680 See the Discussion in Forstner et al. 2025 of this problem. If what you are doing is just connecting traces that look coplanar across outcrop gaps, that is problematic.

**Done**. We have better clarified how no-data zones are treated in the caption of figure 12:

Original (lines 678-680):

Figure 12 Different cases of censoring in a natural outcrop. The presence of information gaps affects trace length measurements. Double censored fractures are considered a single censored fractures with one of the end nodes coinciding with the interpretation boundary.

Revised:

Figure 12 Different cases of censoring in a natural outcrop. The presence of information gaps affects trace length measurements. Double censored fractures are considered as a single censored fracture with one of the end nodes coinciding with the interpretation boundary. Fractures that look coplanar across an information gap are considered two separate censored fractures.

737 Here and elsewhere you can omit 'very'; it is vague and not needed.

**Done**. Thanks for the correction, also following the comment above we will omit 'very'.

835 'implies'

**Done**. We fixed the typo.

849 The true scope of the workflow needs to be stated here and in the Introduction. What you are describing is really only part of a comprehensive workflow for describing fractures.

**Done**. We modified this paragraph following the revised version of the introduction:

Original (lines 849-851):

This contribution proposes a workflow with quantitative methodologies to address the parametrization of fracture networks aimed, for instance, to the generation of stochastic models that are more realistic under the geological and structural point of view.

Revised:

This contribution is focused on the geometrical characterization of fracture networks and in particular on the input parameters necessary to generate stochastic DFN models. The main goal of the paper is to provide quantitative methodologies that limit the user choices as much as possible through the implementation of statistical tests (e.g. orientation distribution). If statistical tests are not viable due to the violation of the underlying assumptions, other statistical parameters (P21 REA) or statistical distances from a non-parametrical estimator (length and height distribution) are provided. The presented methodologies are based on data collected from DOMs, both point clouds and orthomosaics. In the context of upscaling geometrical parameters, DOMs are a convenient framework when it comes to collecting data on wide outcrops, decreasing the time for the acquisition process, allowing data collection in areas inaccessible due to practical or safety reasons, and opening to the possibility of implementing automatic feature extraction methods or automatic classification methods (topology). For a complete characterization of the fracture network, especially when targeted to fluid flow simulations, the geometrical parameters included in this contribution have to be integrated with further analysis, to characterize filling, mineralization and other characteristics of the network (e.g., microscale connectivity) that can be assessed with other type of techniques and at a smaller scale (Forstner et al., 2025).

855-858 So how did you accomplish this filtering? This is a general problem, and not just restricted to outcrops in quarries. Spurious near surface fractures can be in regular sets. See the protocols in Eppes et al. and in the older literature.

**Done**. Regarding this comment, we used the word "filtering" but probably is not appropriate for what we did in this paper. We didn't write it explicitly (it is explained in the result section), but in areas affected by noise related to quarrying activities we digitalized only Set 1 fractures that were clearly recognizable due to orientation, length, and presence of centimetric displacement. We know that the other fracture sets are present, because we can measure them on the adjacent wall, but we were not able to digitalize them on the pavement. We modified the text to better explain the digitalization procedure.

Original (855-858):

Filtering noise and associating a genetic signature to each fracture set is an obvious concern in a quarry, where some fractures were generated during quarrying operations, but is of the outmost importance also in an outcrop analogue modelling perspective, if for instance we want to filter out fractures generated during exhumation, that might not be present in a reservoir still buried at depth.

Revised:

In quarries some fractures are generated during excavation. It is thus of the utmost importance, for both genetic reconstructions and analogue modelling, to exclude fractures that are related to anthropogenic surface processes. In our case study, the measured fracture sets are in accordance with the existing literature on the area (Sec. 2). In the outcrop pavement are present no data zones, characterized by debris accumulations, where no fracture set can be detected. Other parts of the pavement are affected by quarrying activities, resulting in zones "saturated" by fractures with random orientations or distributed following a radial pattern (related to explosions). In these areas only Set 1 is clearly detectable, given the constant spacing and orientation, an average length higher than the other fractures and the centimetrical displacement. Set 2 and Set 3 are drowned by these artificial fractures and even if present it is difficult to reliably isolate and digitalize them.

865 The claim 'complete characterization…' of 'fracture network parameters' seems overstated and vague. What parameters? Do you mean 'heights and correlated lengths' and associated connectivity patterns? In sedimentary rocks the bed-normal patterns of heights are commonly quite different from length patterns. Heights may or may not be correlated with stratigraphic features that (depending on depositional environment, etc) may be on a meter or less scale, whereas sedimentary boundaries have been shown to affect length patterns but on a longer scale (again depending on the scale of the sedimentary features). An outcrop trace pattern example of the later is in Geol. Mag. 2016, v 108., p. 135, their fig. 2. This is one reason why classifying height patterns (i.e., height relative to stratigraphic features) should precede drawing conclusions about height/length relations.

**Done**. We thank the reviewer for the comment. Following the suggestions provided in other comments we have modified the sentence to make it clearer:

Original (lines 865-866):

The integration of facets and traces (collected both on horizontal and vertical outcrops) allows a complete characterization of fracture network parameters…

Revised:

The integration of facets and traces (collected both on horizontal and vertical outcrops) allows a complete characterization of the parameters listed in Table 1…

866 'only one of these two data sets' is confusing. It's not clear what 'datasets' in the cited references that you mean. Do you mean 'that rely on data collected on either bed-parallel outcrops or in cross section'?

**Done**. We have changed the phrase to make it clearer:

Original (lines 866-867):

unlike other approaches that rely on the analysis of only one of these two datasets (e.g. Ortega et al., 2006; Boro et al., 2014; Martinelli et al., 2020; Smeraglia et al., 2021).

Revised:

unlike other approaches that rely on facets or traces only (e.g. Ortega et al., 2006; Boro et al., 2014; Martinelli et al., 2020; Smeraglia et al., 2021).

859 'very high quality' is vague; omit 'very'; and 'perfectly exposed' seems to contradict your Conclusion, where you note that there are gas in the outcrop. Even the best outcrops have covered areas and exposure gaps, although there are some that are quite large and clean.

**Done**. We modified the text as follows:

Original (lines 859-863):

The very high quality of our outcrop, with perfectly exposed horizontal and vertical surfaces, was instrumental in testing techniques that represent, in our opinion, a step forward in collecting rich quantitative datasets and developing rigorous statistical treatments for many parameters of a fracture network (Table 1). On the other hand, we must also recall that for some parameters there are still limitations in data collection and analysis. Both these points are discussed in the following sub-sections.

Revised:

The high quality of our outcrop, with adjacent horizontal and vertical surfaces, was instrumental in testing techniques that represent, in our opinion, a step forward in collecting rich quantitative datasets and developing rigorous statistical treatments for many parameters of a fracture network (Table 1). On the other hand, we must also recall that for some parameters there are still limitations in data collection and analysis. Both these points are discussed in the following sub-sections.

872 Is this typically done? I don't think this is the protocol in Healy et al.

Yes, we confirm that Fracpaq considers B nodes as if they are I nodes.

874 Awkward. '…huge areal extension…' > 'xxxx m2 extent' (provide an area rather than the vague 'huge' and it's 'extent' not 'extension'

**Done**. We modified the text as suggested:

Original (lines 874-875):

*P*21 is calculated on the pavement TS-DOM where the huge areal extension enables to define a sufficient number of scan areas to detect the REA lower threshold.

Revised:

*P*21 is calculated on the pavement TS-DOM where the huge areal extension ($\approx$ 18.000 m$^2$) enables to define a sufficient number of scan areas to detect the REA lower threshold.

866 On the other hand, some of these techniques provide information on key parameters that you do not address. Ortega et al. for example, provide aperture data over wide size ranges, but you apparently do not collect any width information so how can your technique be portrayed as 'a complete characterization of fracture network parameters'. Your contribution can stand on its own without overselling it

**Done**. We agree with the reviewer. We adjusted the scope of the paper in the comment above by removing "complete characterization" and limiting it to the parameter addressed in this contribution. (lines 865-866)

870 But you ought to note the limitations; for example, your DOM is all at one scale. What happens if you collect fracture data over a range of scales and resolutions (e,g,. Forstner et al. 2025; Elliott et al. 2025).

**Done**. We thank the reviewer for the comment. We brought an example where data has been collected at a fixed scale. The presented methodologies for analyzing fracture traces can be applied to data collected at different scales. It is more difficult to collect facets data at the thin section or at satellite scale but nonetheless clustering and goodness-of-fit tests can be applied to every dip/dip direction datasets regardless of the scale or technique used to collect them.

Original (lines 870-873):

TS-DOMs allow the digitalization of fracture traces and of the interpretation boundary both on horizontal and vertical outcrops. The integration between fracture traces and the interpretation boundary is fundamental to avoid underestimating the connectivity index by misinterpreting B-nodes as I-nodes and provides a fundamental input to identify censored fractures.

Revised:

TS-DOMs allow the digitalization of fracture traces and of the interpretation boundary both on horizontal and vertical outcrops. TS-DOM allows the digitalization of fracture traces at a fixed scale, corresponding to the resolution at which they were collected. Nonetheless, the proposed methodologies can be applied to different scales, from thin sections to satellite

images, provided that data are organized as digitized fracture traces combined with the interpretation boundary.  The integration between fracture traces and the interpretation boundary is fundamental to avoid underestimating the connectivity index by misinterpreting B-nodes as I-nodes and provides a fundamental input to identify censored fractures.

911 The criterial is scale and diagenetic considerations. The latter point relates to whether the fractures are open.

**Done**. We want to apologize for the misinterpretation. We fixed the text:

Original (lines 909-911):

From an almost opposite perspective, other authors (Forstner and Laubach, 2022) suggest considering also contingent nodes (C nodes) that would allow merging individual small branches to form larger traces, based on genetic or hydraulic considerations.

Revised:

From an almost opposite perspective, other authors (Forstner and Laubach, 2022) suggest considering also contingent nodes (C nodes) that would allow merging individual small branches to form larger traces, based on the considered scale and/or diagenetic consideration.

906 It's not clear what your point is here. The text needs to be thought through a bit more. The contrast you draw between Sanderson and Nixon and Forstner and Laubach I think is ill posed. The problem of open fracture continuity for opening mode fractures identified by Forstner and Laubach would be equally valid if you could somehow see the 3d shape of fractures; this is not an artifact of looking at a 2D surface. The issues you need to address are on the most general level, that fractures intersecting a 2D surface (or fractures in 3D) may or may not be open. In other words, the fracture maps (and all the parameters derived from them) may be unrelated to fluid flow (or strength). Consequently, the only way to go from the outcrop analog fracture map to any kind of fluid flow estimation needs to account for whether the fractures are open. This can be done by simply assuming they are all open, as is usually done. Such an approach can lead to insights, but it contradicts geologic observations to say that the maps and derived parameters are all you need to estimate flow. Other geologic evidence can and should be brough to bear. The geologic evidence shows that what fractures (or parts of fractures) are open depends on scale (fracture width) so some fractures (or parts of fractures) may be sealed, so this information can be used to make trace maps more informative for flow parameters. Other geologic elements could be in play. For example, the 'network' might be a mixture of opening-mode fractures and faults.

**Done**. We thank the reviewer for the comment. We want to apologize as the comparison between the two approaches is indeed ill posed. We removed "From an almost opposite perspective" from the revised version. Fractures that should or should not be considered in topological analysis, and more generally in the orientation, length and height parameters calculation, depend on the scope of the work. How length, orientation and topological relationship are measured is independent of the type of fracture or discontinuity we are considering (open fracture, veins, stylolites etc.).

Original (lines 909-911):

From an almost opposite perspective, other authors (Forstner and Laubach, 2022) suggest considering also contingent nodes (C nodes) that would allow merging individual small branches to form larger traces, based on genetic or hydraulic considerations.

Revised:

Other authors (Forstner and Laubach, 2022) suggest considering also contingent nodes (C nodes) that would allow merging individual small branches to form larger traces, based on genetic or hydraulic considerations.

943 On the other hand, Forstner et al. 2025 make the case that outcrop trace maps, however collected, always overestimate length.

We think that the digitalization scale and the underestimation of fracture mean trace length are two separate problems. Underestimation of fracture length comes from the presence of censored fractures, while overestimation depends on the scale and additional considerations as well explained in Forstner and Laubach 2025. For example, consider a one-meter-long censored fracture (one of the two terminations touches the interpretation boundary). If further analysis reveals that this fracture actually consists of ten individual fractures, each approximately 10 cm in length, the one intersecting the interpretation boundary remains censored, resulting in an underestimation of the length distribution parameters.

962 Awkward. 'allowing to define' > 'defining'

**Done**. Thank you for the correction.

966 The text starting "For these reasons a step behind..." is confusing. Clarify what you mean.

**Done**. We modified the text accordingly.

Original (lines 966-969):

For these reasons a step behind has been taken with respect to the application of formal statistical tests in the definition of REA (Martinelli et al., 2020), adopting a more qualitative approach which, however, does not require such stringent assumptions as the tests used by (Martinelli et al., 2020).

Revised:

For these reasons a quantitative approach based on formal statistical tests cannot be safely applied. Adopting a more qualitative approach will result in a less significant result, which partially depends on the interpreter choice, however, it does not require such stringent assumptions as the tests used by (Martinelli et al., 2020).

981 'every relevant parameter' is overstating this case, as you have not characterized the distributed aperture distribution or where the fractures are open versus not open.

984-988 Are these major conclusions? The first bullet item is no news to anyone who has measured fractures in outcrop and if this is a conclusion, the Introduction does not do a good job of setting up or prefiguring this point. Also, if this is an issue why not discuss the solution for describing and dealing with incomplete outcrop patches suggested by Forstner et al. 2025?

986-988 This statement does not seem to be posed as a Conclusion.

991 Where was 'tectonic related fracture' sets introduced? Why not stick to descriptive terms? How do you know that any of the sets are related to tectonics? I don't see anything in the MS that demonstrates when and why these fracture sets formed (and is it even relevant to a characterization workflow?).

**Done**. We have decided to revise the conclusions with a stronger emphasis on the methodologies presented, while leaving out the geological interpretations regarding the nature of the fracture sets.

Revised:
In conclusion, this paper presented a series of quantitative methodologies to characterize fracture network geometry from Digital Outcrop Models (DOMs). Among all the parameters required to fully characterize a fracture network, we focused on those required to generate 3D stochastic DFN models, that are: orientation parameters, topological relationships, length and height distribution parameters, H/L ratio and P21:

- Orientation data are collected through a semi-automatic workflow, clustered with k-medoids, and tested for the goodness-of-fit to a Fisher distribution. Alternatively,

the Kent distribution parameters are also provided. This procedure allows subjectivity to be removed from the assignment of dip/dip direction data to a specific fracture set and supports the choice of meaningful orientation parameters through the implementation of statistical tests.

- Topological relationships are calculated including the interpretation boundary, this allows to: (i) to define B nodes and exclude them from the connectivity index (CI) calculation (ii) to identify censored fractures in an automatic way. Backbone extraction highlights the presence of large, connected clusters in the network. Crosscutting and abutting relationships between different fracture sets are quantified through directional topology.
- The approach developed to deal with censoring bias provides as a result a set of fully specified distributions (all parameters are explicit) corrected for censoring. The best model among the initial selection is defined through a graphical approach and a series of statistical distances.
- We demonstrate that estimating H/L is not possible without introducing some assumption, even for the best exposed set and in the presence of both horizontal and vertical exposures. Therefore, we opted to make our assumption as transparent as possible, and we tested it with regression analysis.
- P21REA is calculated with a qualitative approach to avoid violating the underlying assumption of more formal statistical tests.

1255/1260 Why are the article titles formatted differently? Make these uniform.

**Done**. Thank you for the correction, we did not notice the difference in format.

1265 Check the author's name.

**Done**. Thank you for the correction.

---

## Author Comment (AC2)

We thank the reviewer for the useful comments and the time spent reading the paper. As for the first reviewer, we will address all the comments and explain how we intend to modify the manuscript following the reviewer's suggestions. Reviewer's comments are reported in black and the replies in blue, line numbers refer to the submitted version of the manuscript.

Introduction: it is clear from the reading that the presented workflow is specifically intended to provide useful parameters to be considered in DFNs. For this reason, much space is dedicated to the limitations of these and the benefits that could be drawn from them. However, only one paragraph is dedicated to the true problems addressed in the paper, that is, to give statistical robustness to some of the major bias (four identified by the Authors) concerning the determination of fracture set/network in DOMs. This section should be expanded by considering more potential bias (for example subjective bias in the data collection/analysis or bias due to inefficient algorithms for the automatic extraction of planes) and an improved consideration of the relevant literature. Relevant papers to be mentioned are, among the others; Baecher, 1983, Zeeb et al., 2013; Zhang, 2016; Watkins, 2018, Andrews et al., 2019, Eppes, 2024.

**Done**. We thank the reviewer for the useful comment and references. We agree that we didn't consider subjective biases and biases related to, for example automatic feature extraction method. We added a phrase to include them in the introduction: Revised (from line 85):

In addition to objective biases related to outcrop geometry or sampling methods, subjective biases introduced by the interpreter should also be considered (Andrews et al., 2019). In the specific context of automatic feature extraction, it is also important to account for biases inherent to the algorithms themselves, including the potential for extracting artifact features.

Figure 2: the correspondence of colours between pavement/wall and steroplots is somewhat lacking; for example, red bars in E are missing (or are they placed together with blue one?). Colours of Set 1 and 2 are too similar, please use more contrasted colours. Field-data in D are shown without set differentiation but, as you also state in the text, fieldwork is the best way to distinguish fracture sets on the basis of their geological characteristics. In addition to the raw data, the cluster division proposed by the fieldwork should also be provided. In fig Fig 2C, the distinction between set 1 and 2 on the 2D orthoimage of the wall seems to be impossible because of lack of strike information. Later (lines 776-795), the Authors present a statistical correlation between height and length based on the assumption that traces mapped on the pavement and on the wall can be associated in ordered pairs from the shortest to the longest. Maybe in this particluarly

simple geological setting this assumption could be plausible, however I think it is necessary to manually check the corresponding sets by mapping the 3D fractures traces and fit the 3D planes. I understand that, due to the flatness of the wall, it could be hard to accomplish for mostly visible traces but it can be do at least for some of them. This procedure can provide a significant validation of your workflow.

**Done**. We thank the reviewer for the comment, which gives us the opportunity to clarify an aspect we mistakenly took for granted. Starting with Figure 2, we agree that the colors chosen for Set 1 and Set 2 on the vertical wall are not sufficiently contrasted. We have now added the medoid orientation for each cluster in the filed data stereoplot. Each medoid is colorized with the color of the set according to the legend.

[Figure]

The fractures were digitized on the vertical orthomosaic while taking the point cloud into account. Each vertical trace was associated with a plane characterized by dip and dip direction extracted from the point cloud. Some fractures on the vertical wall could not be reliably associated with a fracture plane and were therefore excluded from digitization. Additionally, on the vertical wall, we were unable to distinguish fractures with centimeter-scale displacement from those without. Consequently, we grouped features such as joints and faults into broader categories—e.g., combining both into a more general Set 1. We added a paragraph in section 5 to clarify this aspect:

Revised (from line 486):

The digitization of fractures on the vertical TS-DOM is supported by the corresponding PC-DOM. By integrating TS-DOM and PC-DOM data, each digitized fracture trace can be associated with a best-fit plane derived from the point cloud. This approach enables the assignment of fracture traces to specific fracture sets. Fractures on the vertical wall that could not be reliably linked to a fracture plane were excluded from the digitization process.

Line 212: You miss to specify the resolution of the terrestrial photogrammetric acquisition. Based on figure 3A the mean point spacing is around 0.5 pts/mm, which corresponds to ca. 2 mm of resolution (GSD). Therefore, the wall and pavement datasets are different in resolution (4 mm vs. 2 mm) possibly influencing the fracture mapping. I think the Authors should consider this in the methodology section.

**Done**. We thank the reviewer for this correction. The resolution of the vertical orthomosaics is 2 mm/pixel (specifically, 2.06 mm/pixel). We have now added this information at the end of the paragraph on line 220.

We do not consider this slight difference in resolution significant enough to affect the digitization process, as the length of the smallest detected features is on the order of some decimeters, approximately an order of magnitude larger than the orthomosaic resolution. However, we agree that if the orthomosaics were used to digitalize structures at the millimeter scale, such a resolution difference could indeed have a noticeable impact.

Revised (from line 220):

The resulting photogrammetric model has a resolution of approximately 2 mm/pixel.

Line 255-261: it seems that the Authors inverted the reference to the figure panel 3C and 3D.

**Done**. Thank you for the correction. We fixed the reference

Line 257-261: Why the Authors propose a comparison between a low-quality photo acquisition from commercial drone DJI Mini 3 Pro when they used a professional DJI Mavic 3E for the pavement acquisition, that can provide higher quality picture? Moreover, the comparison between the two datasets is unclear because the Authors never specified how the drone dataset was acquired (outcrop camera distance, scheme of acquisition, orientation of the camera, etc). The resolution of DOM mostly depends from the drone camera quality and distance of acquisition, therefore using professional drones and flying close to the outcrop it's possible to obtain high quality DOM, comparable to the terrestrial acquisition. Moreover, as visible in the figure 3C and 3D, the drone dataset suffers less of occlusions (i.e. shadow areas) making it more appropriate for 3D tracing and mapping. This is particularly true for large and high rock walls where the terrestrial acquisition cannot provide proper data. In conclusion, the presented comparison between the two datasets is weak and doesn't provide any significant contribution in the workflow.

**Done**. We thank the reviewer for the comment and the opportunity to clarify this point. The purpose of the comparison was not to compare ground-based and UAV photogrammetry per se, but rather to highlight the impact of using a high-quality camera and a proper acquisition scheme versus a lower-performing camera and a simplified acquisition geometry. This was illustrated through differences in point cloud surface density.

We fully agree that a UAV equipped with a professional-grade camera and a proper acquisition scheme could achieve results comparable to the ground-based survey. Likewise, a similar contrast could have been demonstrated using a ground-based survey performed with an entry-level camera. At the time of acquisition, the DJI Mini 3 represented the least capable instrument available to us, which is why it was selected for this illustrative comparison.

We believe the comparison remains meaningful, as the three-order-of-magnitude difference in point cloud density between the two models strongly supports the need for high-quality equipment and acquisition geometry—whether ground-based or UAV—for reliable data collection.

We have slightly modified the text to avoid highlighting the comparison between terrestrial survey and UAV.

Original (lines 255-259):

As an example, in Figure 3, two PC-DOMs of the same vertical outcrop are compared, collected in two different ways to 255 obtain a different $SD$. The PC-DOM in Figure 3D is reconstructed from more than 400 photos collected as discussed above (terrestrial survey with fans scheme, with high end Nikon Z7 mirrorless). On the other hand, the PC-DOM in

Figure 3C is collected with a smaller dataset (150 photos) collected with the lower quality camera of a small commercial drone (DJI Mini 3 Pro).

Revised:

As an example, in Figure 3, two PC-DOMs of the same vertical outcrop are compared, collected in two different ways to obtain a different $SD$. The PC-DOM in Figure 3C is reconstructed from more than 400 photos collected as discussed above (fans scheme, with high end camera, Nikon Z7). On the other hand, the PC-DOM in Figure 3D is collected with a smaller dataset (150 photos) collected with a lower quality camera (DJI Mini 3 Pro).

Cap 4 dealing with the semi-automated analysis of fracture orientation from point clouds. My major concerns are about the novelty of this workflow. It seems it has some similaritiy with the DSE of Riquelme et al., 2014. Please, provide a comparision. I also have doubts about its applicability if addressed to geologically complex setting. I guess that when rock masses are affected by multiple foliations, fracture sets, folding etc, it is virtually impossible to detect reliable pre-defined clusters, invalidating all the procedure. In similar cases, it is well known that semi-automatic methods for plane extraction have multiple bias affecting the quality of the data collection. I would like to invite the Authors to consider and highlight these limitations of the method and provide the warning that it can successfully used in areas with low complexity. In the other cases, manual mapping is still the more effective way to derive robust data, even taking into consideration the subjectivity bias (Andrews et al., 2019).

We thank the reviewer for this comment. Regarding the comparison with the method proposed by Riquelme et al., we believe that such a comparison would be ill-posed. Riquelme and co-authors propose a method for the automatic extraction of 2D features directly from point clouds. In contrast, our contribution focuses on a supervised calibration procedure intended to enhance the performance of such automatic methods.

A more appropriate comparison would be between the method developed by Riquelme et al. and FACETS (Dewez et al., 2016), the automatic feature extraction algorithm implemented in CloudCompare. We chose to work with FACETS because it is already integrated into CloudCompare; however, our calibration procedure is generalizable and could also be applied to the method proposed by Riquelme and co-authors.

Concerning the applicability of our approach to more geologically complex case studies, we argue that, from a methodological point of view rather than a geological one, the complexity is not primarily related to the number of fracture sets or the presence of foliations. Instead, it depends on the size of the point cloud patches that can be identified as planar features. Smaller patches require a higher octree level, which increases the risk

of fitting 2D planes to noise. In this respect, the selected case study is particularly challenging, as many fracture planes are not continuous along their trace, and orientation measurements are based on small, isolated point clusters.

To further support the effectiveness of our calibration procedure, we include in this reply a stereoplot resulting from the application of the automatic feature extraction algorithm to the entire outcrop without any prior segmentation of the point cloud.

[Figure]

Cap 8, about fracture intensity. The Authors propose to use the Representative Elementary Area (REA) that is the area above which the value becomes independent from the position and scan area size. They use a lower threshold of REA defined as the minimum hexagon area where no significant difference is detected between the mean and standard deviation of $P21$ obtained at that area and at the next step. Due to the unequal sample size of different scan areas, the Authors propose a qualitative approach based on the difference between the interquartile range (deltaIQR) of two subsequential $P21$ samples, where the REA is reached when deltaIQR stabilizes around 0, displaying a plateau in the diagram of its variation. Given that in scan windows with size close to that of the outcrop the representativity is compromised by the too small sample size, it is also well-known that windows smaller of the average fracture dimension cannot correctly capture the geometrical features. The new proposed calculation gives a statistical confirmation of the prevoius approaches. These suggest that  the size of the area required for a representative quantification of fractures depends on both fracture average length and number density (Rohrbaugh et al., 2002; Zeeb et al., 2013; Zhang, 2016; Eppes, 2024). I'm not entirely convinced by the plateau window indicated by the Authors. It could be widened or

narrowed a bit without significantly affecting the statistical variations. In anycase, a range of 5-12 m is very large and fall in the interval derived from the standard approach.  In general, the case study seems very homogeneous, so it is easy to see how the average P21 is stable for almost all samples. It would be more interesting to see the calculation on more heterogeneous outcrops. The question is whether it is useful to use a statistically averaged REA in heterogeneous outcrops, rather than adapting the window to geological variations?

**Done**. We understand that the reviewer has some reservations about the choice of the interval of sizes where P21 is stable. It is the downside of qualitative methods where the choice is left to the user. At the same time, we think that a qualitative method is still better than a quantitative one where the underlying assumptions are violated (sample size, normality, homogeneity of variance). About the application to more heterogeneous outcrops, it is beyond the scope of this paper, and we are currently working on a future paper about this topic. We added a sentence in the discussions to highlight this problem:

Revised (from line 969):

We also recognize that adopting a more qualitative approach may introduce subjectivity in the selection of window size, but still having an order of magnitude for the REA (and hence for REV) is important in modelling studies.

Line 763: Authors write "$P21$ REA can be safely calculated only for Set 1 fractures, because, as highlighted in Section 2, in some areas only Set 1 fractures can be digitized, while Set 2 and Set 3 are drowned by the quarrying related fractures". If this is true, which is the usefulness of this analysis? Authors declare that they only may safely map one set of fracture, whereas the rest of dataset is somehow masked, hidden, in any case not representative. So, what is the validity of the P21 dataset of only part of the factures)? Same question is valid for the others parameters (topology, H/L ratio, …)?

Some parameters are inherently influenced by the conditions of the outcrop. The P21 analysis, for example, is valid for Set 1 fractures. However, as correctly noted, the datasets for Set 2 and Set 3—derived from the pavement—are incomplete, making it unreliable to calculate a representative elementary area (REA) for P21. This is not a limitation of the method itself, but rather a limitation of the specific case study. Even the use of alternative methods would yield similarly unreliable results due to the incomplete dataset.

This limitation does not apply to all parameters. For the length distribution, we were able to gather a substantial amount of data for all fracture sets, as shown in Tables 5, 6, and 7. The same holds for the height distribution, with the exception of Set 3.

Regarding the topological analysis and backbone extraction, it is explicitly stated in the Discussion section that the current backbone geometry is likely to change if the complete

fracture network was available. This uncertainty is acknowledged and discussed in the context of the limitations imposed by partial exposure (starting at line 927).

Line 859: Authors claim for a "very high quality of our outcrop, with perfectly exposed horizontal and vertical surfaces", but even in this case of exceptional outcrop, they calculate very few statistically robust parameters. A conclusion should therefore be to honestly say that an approach inclined towards statistical calculation, like the one proposed, has very little chance of being applicable in outcrops.

**Done**. In accordance with Reviewer 1's suggestions, we have removed "very" and similar subjective terms. We acknowledge that the outcrop is not perfect. However, the parameters we did not calculate are not related to the applicability of our methodology, but rather to the limitations of the available data.

Specifically, the absence of Set 3 fracture traces on the vertical wall makes it impossible to calculate height distribution parameters for Set 3 and H/L ratio—regardless of the method used. Similarly, the limited presence of Set 2 and Set 3 fracture traces on the pavement prevents a reliable calculation of representative P21 values. While it would have been technically possible to distribute grids to the areas where Set 2 and Set 3 are present, the restricted areal extent would not have allowed us to investigate sufficiently large windows. This remains true even though we could have used windows larger than the mean trace length, following the criteria proposed by the reviewer in other comments.

Therefore, we are not in a position to assess whether the P21 values obtained for these sets would be representative, or whether the representative elementary area (REA) has been reached.

In any case all measured parameters are listed in Table 1, and we believe that they are the majority.

Line 865: Authors write: "The integration of facets and traces (collected both on horizontal and vertical outcrops) allows a complete characterization of fracture network parameters, unlike other approaches that rely on the analysis of only one of these two datasets (e.g. Ortega et al., 2006; Boro et al., 2014; Martinelli et al., 2020; Smeraglia et al., 2021)." However, Authors must consider they contradict what has just been said, that is they have failed to provide many, if not most, of the fracture parameters due to outcrop conditions. I also find it unfair to attribute incompleteness (which subtly suggests poor quality) to previous works that did not use a method similar to the one described by the Authors. Each of the cited works provides description of their approach, placing it in the existing literature and highlighting limitations. The Authors, rather than discrediting previous works, should focus on emphasise merits and limitations of their own research.

**Done**. This sentence has been partially modified following a comment by reviewer 1. We would like to make it clear that we do not intend to discredit other authors, and we apologize if we have given this impression through a poorly formulated sentence. Here is a revised version of the sentence:

Original (lines 865-867):

The integration of facets and traces (collected both on horizontal and vertical outcrops) allows a complete characterization of fracture network parameters, unlike other approaches that rely on the analysis of only one of these two datasets (e.g. Ortega et al., 2006; Boro et al., 2014; Martinelli et al., 2020; Smeraglia et al., 2021).

Revised:

The integration of facets and traces (collected both on horizontal and vertical outcrops) allows a complete characterization of the parameters listed in Table 1, while other approaches rely on the analysis of facets or traces only (e.g. Ortega et al., 2006; Boro et al., 2014; Martinelli et al., 2020; Smeraglia et al., 2021).

Cap 11. I suggest the Authors to avoid the continuous use of terms such as "robust" or "rigorous analysis" in contrast to what done in the past. It seems to read that the Authors discover now how to manage DOM fracture data. This is not the case. The paper has some merits that I recognize and that should be rightly highlighted. However, it also has many limitations, as highlighted by the Authors themselves. I therefore ask to review the way in which this discussion is presented.

We acknowledge the reviewer's observation and have revised the discussion accordingly, taking into account the comments provided by both Reviewer 1 and Reviewer 2. In the revised version, we have avoided overly assertive terms such as 'robust' or 'rigorous' when referring to our approach.

Line 886: Why "unfortunately"? I suggest to avoid moralisms. Moreover, why these papers among the others? Practically the entire community takes the same assumptions. I guess that this is simply an aspect not considered in much of the previous research. I suggest to the Authors to highlight the novelty of their statistical approach. So, these sentences need adjustment so as not to be misleading.

**Done**. We apologize for the poor wording of the sentence. We have removed the term "unfortunately" which is indeed unnecessary. On the other hand, we would like to highlight that carrying out a "rigorous analysis" includes being self-critical, considering both pros and cons of the method. In any case, although we originally cited a few recent papers as

examples, we agree that this practice is common and does not require specific references. Here is the revised version of the bullet point:

Original (lines 885-887):

Testing the fitted orientation distributions with goodness-of-fit tests, instead of assuming circular symmetry and a Fisher distribution without a proper statistical test as (unfortunately) is a common practice in structural geology (e.g. Bisdom et al., 2014; Smeraglia et al., 2021; Menegoni et al., 2024; Panara et al., 2024, just to cite some recent papers).

Revised:

Rather than assuming circular symmetry and fitting a Fisher distribution without prior statistical verification, our approach explicitly tests the fitted orientation distributions using goodness-of-fit tests. This provides a more robust and statistically grounded assessment of fracture set orientation parameters.

Line 928-936: here the Authors seem to make explicit the main problem of the presented approach based on a statistical validation of each parameter. The robustness of the statistical analysis is effective only if with truly complete fracture mapping along the entire outcrop. The presence of even small hole in the dataset (i.e. debris patches or not perfectly exposed walls) can invalidate the entire results. The Authors need to highlight these limitation and the repercussions on the applicability of the method in other settings.

We thank the reviewer for this comment. However, we respectfully disagree with the statement that topological analysis is invalid due to the presence of no-data zones. If this were the case, it would imply that most published studies involving fracture network topology—many of which are based on incompletely exposed outcrops—would also be considered invalid. The limited extent of the outcrop itself often represents a more significant constraint. We agree that if the no data zones cover most of the outcrop surface or their size match the outcrop scale, probably the analysis would be compromised. But this is an outcrop selection problem.

In our study, we explicitly addressed the impact of no-data zones in several ways. For length and height distributions, we identified censored fractures using B-nodes and applied a survival analysis approach to correct for censoring bias. From a topological perspective, we accounted for B-nodes by identifying and removing them to avoid underestimating the connectivity index. These steps were taken specifically to mitigate the limitations associated with partial data and ensure that the results remain as representative as possible.

Line 960-961: why "arbitrary" and why "without defining a proper representative sampling area"? Previous studies constrain the size of the window area on the fracture average length and the number density as described by, among the others, Zeeb et al., 2013, Zhang, 2016, etc The use of the REA is a novelty in the DOM analysis but it's not the unique reliable method. Previously defined standards for the scan window definition are based on more empirical data rather than purely statistical approach, i.e. performing multiple tests in different geological contexts, with changing operators, outcrop conditions, varying fracture intensity homogeneity, which often doesn't match the ideal one described in this paper.

**Done**. We apologize for the misleading statement regarding the methodologies cited. It is correct that a correlation between mean fracture trace length and the minimum scan area required for a representative fracture intensity calculation was provided in the referenced works. Nevertheless, defining the mean trace length without defining the statistical distribution is meaningless (as demonstrated by Benedetti et al, 2025). For instance, the REA would not exist at all for a strictly fractal (power-law) length distribution. In any case we have revised the paragraph, removing the misleading statement and placing greater emphasis on the specifics and contributions of our own method.

Original (lines 960-963):

Areal fracture intensity $P21$ is quite often calculated using scan areas of arbitrary size, without defining a proper representative sampling area (e.g. Bisdom et al., 2014; Menegoni et al., 2024; Panara et al., 2024). To our knowledge, only Martinelli et al. (2020) presented an analysis allowing to define the minimum REA where fracture intensity can be mediated to ensure a proper continuum-equivalent description (Bear, 1975).

Revised:

Areal fracture intensity is often estimation using methods based on scan lines, scan areas or circular scan line (Rohrbaugh Jr. et al., 2002; Zeeb et al., 2013). These methods provide a minimum scan area size for a representative estimation of P21 based on the mean fracture trace length. In this contribution we proposed a different approach, based on the concept of Representative Elementary Area to try to quantify the range of scan area size in which fracture intensity can be mediated to ensure a proper continuum-equivalent description (Bear, 1975).

Conclusions don't fit the Introduction themes, and they present main results in a too local way. I suggest to better discuss the improvements of the new statistical approach that, tahnks to the related algorithms, allows to improve the determination of the fracture parameters. On the other hand, the important limits of applicability of the methodology should be highlighted even in cases of top-quality outcrops.

**Done**. The conclusions have been revised in accordance with the suggestions of both Reviewer 1 and 2. We attach the modified version of the conclusions for reference:

Revised:

In conclusion, this paper presented a series of quantitative methodologies to characterize fracture network geometry from Digital Outcrop Models (DOMs). Among all the parameters required to fully characterize a fracture network we focused on those required to generate 3d stochastic DFN models, that are: Orientation parameters, Topological relationships, length and height distribution parameters, H/L ratio and P21:

- Orientation data are collected through a semi-automatic workflow, divided into cluster via a clustering algorithm (k-medoid) and tested for the goodness-of-fit to a Fisher distribution. Alternatively, the Kent distribution parameters are also provided. This procedure allows subjectivity to be removed from the assignment of dip/dip direction data to a specific fracture set and supports the choice of meaningful orientation parameters through the implementation of statistical tests.
- Topological relationships are calculated including the interpretation boundary, this allows to: (i) to define B nodes and exclude them from the connectivity index (CI) calculation (ii) to identify censored fractures in an automatic way. Backbone extraction highlights the presence of large, connected clusters in the network. Crosscutting and abutting relationships between different fracture sets are quantified through directional topology.
- The approach developed to deal with censoring bias provides as a result a set of fully specified distributions (all parameters are explicit) corrected for censoring. The best model among the initial selection is defined through a graphical approach and a series of statistical distances.
- Estimating H/L was not possible without introducing some assumption, even for the best exposed set and in presence of both horizontal and vertical exposures. Therefore, we opted to make our assumption as transparent and possible, and testing it with regression analysis.
- P21REA is calculated with a qualitative approach, to avoid violating the underlying assumption of more formal statistical tests.

---

## Author Comment (AC3)

We thank Giacomo Medici for the time spent reviewing the manuscript, enriching the discussion with this insightful community comment. We will address all the comments and explain how we intend to modify the manuscript following the commenter's suggestions. The commenter's comments are reported in black and the replies in blue, line numbers refer to the submitted version of the manuscript.

Lines 36-42. Add the fundamental control of fractures on contaminant transport, see references below:

- Medici, G., Munn, J.D., Parker, B.L. 2024. Delineating aquitard characteristics within a Silurian dolostone aquifer using high-density hydraulic head and fracture datasets. Hydrogeology Journal, 32(6), pp.1663-1691.

- Cherubini, C., 2008. A modeling approach for the study of contamination in a fractured aquifer. Geotechnical and geological engineering, 26, pp.519-533.

**Done**. Thank you for the references. We indeed missed this part of literature about fracture networks. We added "contaminant transport" at line 42:

Revised (from line 42):

contaminant transport (Medici et al., 2024; Cherubini, 2008)

Line 145. Mention the Apula Platform?

Apulian platform is already mentioned in the case study section

Line 145. Age of the limestones?

**Done**. Thanks for the comment. We added the age of the formation at line 166:

Original (lines 166-167):

The quarry is carved into the shallow marine intertidal limestones of the Calcare di Altamura Formation.

Revised:

The quarry is carved into the shallow marine intertidal limestones of the Calcare di Altamura Formation (Coniacian to Early Campanian, Panza et al., 2016).

Line 146. You need to clearly state the specific objectives of your DFN research by using numbers (e.g., i, ii, and iii).

The introduction has been revised in accordance with the reviewers' comments. We believe that the revised version more clearly conveys the scope and objectives of the paper.

Line 304. This sentence is not clear: "This means that different....". You need to specify what you mean by "this" to clarify.

**Done**. We thank the reviewer for the comment. Here is the revised version of the sentence:

Original (line 304):

This means that different conventions can be used for the sense of normal vectors...

Revised:

This symmetry implies that different conventions can be adopted for the sense of normal vectors...

Lines 849-864. You should add references. This is a discussion.

We thank the reviewer for the comment. Specific references are provided in the following sub sections of the discussion.

Line 1038. Insert the relevant papers suggested above on fractured aquifers.

**Done.**

Figure 2. How many rock discontinuities in the 3 stereonets? You need to add the number close to each rose diagram.

**Done**. Thank you, we added the number of data as suggested.

[Figure]

**A**

Pannonian Basin

Alps

Dinarides

Adriatic Sea

Apennines

Tirrenian Sea

Jonian Sea

Atlas

250 km

SW

0 m          50

**B**

- Set 1 - Joint
- Set 1 - Fault
- Set 2 - Joint
- Set 2 - Fault
- Set 3a,b
- Interpretation Boundary
- Major Structures

0 m     25     50     N

**C**

0 m    2    NE

- Set 1
- Set 2
- ND
- Int. Boundary
- S
- Set 1
- Set 2
- Set 3a,b
- Major Structures
- S

**D**    n. data = 85

**E**    n. data = 4790

**F**    n. data = 810

Figures 3c and d. Do you need a spatial scale?

**Done**. Thank you, we added the spatial scale in figure 3c and 3d

Figure 5. Also here number of fracture readings close to the stereonets.

**Done**. We added the number of data collected for every set in the caption (Steroplot B). Regarding stereoplot E the number of data is already specified in the result tables.

Original (lines 380-383):

Figure 5 Scheme of the semi-automatic workflow presented in Section 4. Point cloud colored based on dip direction with a HSV 380 colour scale. (A) Manual data collection on PC-DOM. (B) Manually collected orientation data during the preliminary orientation analysis. (C) Manual segmentation of the PC-DOM. (D) automatic feature detection with FACETS plugin. (E) Final result of the semi-automatic extraction workflow. Each fracture set is individually shown with contour lines.

Revised:

Figure 5 Scheme of the semi-automatic workflow presented in Section 4. Point cloud colored based on dip direction with a HSV 380 colour scale. (A) Manual data collection on PC-DOM. (B) Manually collected orientation data during the preliminary orientation analysis. Number of data: Set 1 = 351, Set 1 = 256, S = 87, Set 3a = 74, Set 3b = 42 (C) Manual segmentation of the PC-DOM. (D) automatic feature detection with FACETS plugin. (E) Final result of the semi-automatic extraction workflow. Each fracture set is individually shown with contour lines.

Figure 14. Increase the size of the numbers on vertical and horizontal axes.

**Done**.

[Figure]

Figure 15a. Specify the number of points which are present in the graph.

Thank you for the comment. The number of points is already specified in the caption.

---

## Author Response (AR2)

The authors thank Stephen Laubach and the anonymous referee for taking the time to review the responses provided during the first round of reviews. The comments are appreciated and have been considered.

Detailed replies to each of the reviewer's points are provided below, following the color scheme proposed in the previous replies:

- **Red** indicates lines from the revised manuscript (first round).
- **Black** corresponds to reviewer or community comments as well as the unchanged portions of the revised manuscript (first round).
- **Blue** highlights the revised sections and newly added lines in the updated version of the manuscript.

**REVIEWER #1 – STEPHAN LAUBACH**

We fixed the highlighted typos:

**Original (line 75):**

... that only the fracture high enough to about or crosscut the bedding interface can be systematically sampled.

**Revised (line 75):**

... that only the fracture high enough to abut or crosscut the bedding interface can be systematically sampled.

**Original (lines 1131-1132):**

Fisher, N. I. and Best, D. J.: GOODNESS-OF-FIT TESTS FOR FISHER'S DISTRIBUTION ON THE SPHERE, Aust. J. Stat., 26, 142–150, https://doi.org/10.1111/j.1467-842X.1984.tb01228.x, 1984.

**Revised (lines 1040-1041):**

Fisher, N. I. and Best, D. J.: Goodness-of-Fit Tests For Fisher's Distribution On The Sphere, Aust. J. Stat., 26, 142–150, https://doi.org/10.1111/j.1467-842X.1984.tb01228.x, 1984.

**REVIEWER #2**

Cap 4 –Authors argue that "from a methodological point of view rather than a geological one, the complexity is not primarily related to the number of fracture sets or the presence of foliations". Beyond the problem of recognizing small, discontinuous patches, the real challenge is assigning them to discontinuities that make sense geologically. Therefore, I

disagree that the case in question is particularly complex. In truth, almost every rock mass presents fracture planes that are "not continuous along their trace, and orientation measurements that are based on small, isolated point clusters". This is a basic problem common to all case studies. The persisting in the cases with complex geology (in terms of discontinuity network) are the reliability of the cluster predefinition and the possible forcing into classes defined either on a small amount of field data or manually mapped onto the DOM. I suggest the authors exercise caution on this point and highlight limitation of the method.

**Done.** We thank the reviewer for this comment; this is indeed a point of discussion that we had not considered. We have decided to add a sentence in the discussions (sec. 11.2) to highlight the limitations of the method applicability.

**Revised (from line 922):**

The applicability, and thus the quality of the results produced by the automatic feature extraction algorithm, strongly depend on the ability to distinguish and characterize each fracture set within the network. In this study, reliable results were obtained by clearly distinguishing fracture sets through the integration of field data, DOM-derived data, and clustering analysis. In more geologically complex settings, where fracture sets are less well defined, caution is advised both when applying the clustering algorithm—since the number of sets must be specified a priori—and when using the automatic feature extraction algorithm.

Conclusions: I suggest introducing a statement describing the outcrop conditions where the methodology can be applied with more complete and effective results.

**Done**. We agree with the reviewer's comment and have added a sentence in the Conclusions (sec. 12) section to clarify the ideal conditions for applying the proposed methodology.

**Revised (from line 1018):**

The ideal conditions for applying our methodologies involve an outcrop that enables the collection of a statistically significant and complete dataset (depending on the scope of the work). This requires favorable orientation of the outcrop faces relative to the fracture set orientation, overall surface cleanliness (minimal debris, vegetation, or damaged zones), sufficient size to ensure adequate sampling, and the presence of at least two perpendicular exposures (horizontal and vertical). Although such conditions are challenging to achieve in natural settings, they should serve as guidelines for selecting a suitable outcrop.